

# Transfer matrix spectrum for cyclic representations of the 6-vertex reflection algebra II

**Jean Michel Maillet[1*], Giuliano Niccoli[1†] and Baptiste Pezelier[1◇]**

**1** Univ Lyon, Ens de Lyon, Univ Claude Bernard Lyon 1, CNRS,
Laboratoire de Physique, UMR 5672, F-69342 Lyon, France

* maillet@ens-lyon.fr
† giuliano.niccoli@ens-lyon.fr
◇ baptiste.pezelier@ens-lyon.fr

## Abstract

This article is a direct continuation of [1] where we begun the study of the transfer matrix spectral problem for the cyclic representations of the trigonometric 6-vertex reflection algebra associated to the Bazhanov-Stroganov Lax operator. There we addressed this problem for the case where one of the $K$-matrices describing the boundary conditions is triangular. In the present article we consider the most general integrable boundary conditions, namely the most general boundary $K$-matrices satisfying the reflection equation. The spectral analysis is developed by implementing the method of Separation of Variables (SoV). We first design a suitable gauge transformation that enable us to put into correspondence the spectral problem for the most general boundary conditions with another one having one boundary $K$-matrix in a triangular form. In these settings the SoV resolution can be obtained along an extension of the method described in [1]. The transfer matrix spectrum is then completely characterized in terms of the set of solutions to a discrete system of polynomial equations in a given class of functions and equivalently as the set of solutions to an analogue of Baxter's T-Q functional equation. We further describe scalar product properties of the separate states including eigenstates of the transfer matrix.

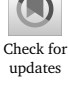
# 1  Introduction

In the recent years, the out-of-equilibrium behavior of close and open physical systems has attracted a lot of interest motivated in particular by new experimental results, see e.g. [2–10]. Microscopic models able to describe such situations are thus in general not only described through their bulk Hamiltonian but also by specifying appropriate boundary conditions. It leads eventually to rather complicated dynamical properties with possible deformations of the bulk symmetries. In the context of strongly coupled systems, integrable models in low dimension, with boundary conditions preserving integrability properties, can be used to gain insights into the non-perturbative behavior of such out-of-equilibrium dynamics, see e.g. [11] and references therein. In particular, they can also describe classical stochastic relaxation processes, like ASEP [12–17] or transport properties in one dimensional quantum systems, see e.g. [18, 19].

The algebraic description of quantum integrable models with non-trivial boundary conditions (namely going beyond periodic boundary conditions) goes back to Cherednik [20] and Sklyanin [21]. Such models have already a long history, that started with spin chains and Bethe ansatz [22–31, 34–36], and continued using its modern developments, see e.g. [21, 37–96]. The key point of the algebraic approaches is an extension of the standard Quantum Inverse Scattering method, see e.g. [97–113], and its associated Yang-Baxter algebra; it takes the form of the so-called reflection equations [20, 21] satisfied by the boundary version of the quantum monodromy matrix. The integrable structure of the model with boundaries can be described in terms of the corresponding bulk quantities supplemented with boundary conditions encoded in some $K$-matrices. To preserve integrability properties, these $K$-matrices should satisfy reflection equations driven by the $R$-matrix of the model in the bulk which solves the usual Yang-Baxter equation. As shown by Cherednik in [20] these reflection equations are just consequences of the factorization property of the scattering of particles on a segment having reflecting ends described by the boundary $K$-matrices. It leads to compatibility properties be-

tween the scattering in the bulk described by the *R*-matrix and the reflection properties of the ends encoded in the *K*-matrices. These are such that there still exists full series of commuting conserved quantities for the model with boundaries generated by the boundary transfer matrix [21]. Its expression is quadratic in the bulk monodromy matrix entries and depends on the right and left boundary *K*-matrices. Then, as for the periodic case, the local Hamiltonian for the boundary model can be obtained from this boundary transfer matrix. This is the standard framework to then address the resolution of the common spectral problem for the transfer matrix and its associated local Hamiltonian. There have been quite a number of works devoted to boundary integrable models using, as in the periodic situation, various versions of the Bethe ansatz [37–96]. It appeared however, that while for special models and boundary *K*-matrices a method very similar to the standard algebraic Bethe ansatz (ABA), here based on the reflection equations, can be applied, the case of the most general boundary conditions (and associated *K*-matrices) preserving integrability turns out of be out of the reach of these methods. This motivated the use of different approaches like in particular the use of *q*-Onsager algebras, see e.g. [56–59], modifications of the Bethe ansatz [54,55,60–64,71] and the implementation for this case of the separation of variable (SoV) method [114–141]. For an extensive discussion and comparison of these various methods in the case of boundary integrable models, we refer to the general discussion given in the introduction of our first article [1] and to the references therein.

In our article [1] we started the implementation of the SoV method to cyclic representation of the 6-vertex reflection equation associated to the most general Bazhanov-Stroganov quantum Lax operator [142–147]. Let us recall that the periodic boundary conditions case (spectrum and form factors) was considered in previous works [123–131], generalizing in particular [148,149]. The interest in such a problem is due to the fact that special cases include the Sine-Gordon lattice model at roots of unity and the Chiral Potts model [150–160, 160–164]. In [1] we started the analysis considering the special case where one of the boundary *K*-matrices has triangular form (which is equivalent to one constraint on the boundary parameters). For that situation we have been able to apply successfully the SoV method by identifying the separate basis as the eigenstate basis of a special diagonalizable *B*-operator with simple spectrum which can be constructed from the boundary monodromy matrix entries. Then using this separate basis, the spectrum (eigenvalues and eigenstates) for the boundary transfer matrix was completely characterized in terms of the set of solutions to a discrete system of polynomial equations in a given class of functions.

The purpose of the present article is to address this spectral problem for the most general boundary conditions preserving integrability, namely for the most general *K*-matrices solution of the reflection equation. The method to reach this goal is to design a gauge transformation that enable us to put this general situation into correspondence with the previous one, namely with a model having one triangular *K*-matrix. For that purpose, the standard idea of Baxter's gauge transformations, see e.g. [98, 111] and references therein, has to be adapted in a way similar to [80, 82] and generalized to these cyclic representations of the 6-vertex Yang-Baxter algebra. Then using this correspondence, the method and tools obtained in our first paper [1] can be used, leading to the complete characterization of the spectrum (again eigenvalues and eigenstates) of the general boundary transfer matrix. We also give determinant formula for the scalar products of the separate states. Further, we show that the spectrum characterization admits a representation in terms of functional equations of Baxter T-Q equation type. Let us note that an analogous inhomogeneous Baxter's like equation has been already proposed in [72] for this model on the basis of pure functional arguments on the fusion of transfer matrices. Thanks to our SoV construction, we prove in the present article that our inhomogeneous Baxter's like equation does characterize the full transfer matrix spectrum. Let us further remark that the inhomogeneous term in the proposal of [72] is presented in terms of the averages of the entries

of the monodromy matrix. For general cyclic representations these quantities are defined only through (unresolved) recursion formula in [72]. Hence in [72] this inhomogeneous term is in fact not given explicitly which makes the comparison with our explicit functional equation not directly possible. Moreover, we would like to stress that in our formulation the Q-function are Laurent polynomial of a smaller degree compared to the one in [72]. This is due to the fact that the inhomogeneous term being computed explicitly in our SoV derivation it allows us to remove $4p$ ($p$ being an integer characteristic value defining the cyclic representation, see (10) in section 2) irrelevant zeros from the T-Q functional equation as they can be factored out from each term of this equation (see section 5).

This article is organized as follows. In section 2 we just recall the basics of the cyclic representations associated to the Bazhanov-Stroganov quantum Lax operator. In section 3 we define the gauged transformed reflection algebra that put into correspondence the most general boundary condition $K$-matrix with a triangular one. It enables us to adapt the SoV method that we already described in our first article [1] to this more general context, leading in section 4 to the transfer matrix spectrum complete characterization in this SoV basis. There we also present the scalar product formulae for the so-called separate states containing the transfer matrix eigenstates. In section 5 we show that the spectrum characterization admit a representation in terms of functional equations of Baxter T-Q equation type. Details about the construction of the gauge transformation are given in Appendices A and B together with determinant identities used in the spectrum characterization in Appendix C.

## 2 Cyclic representations of 6-vertex reflection algebra.

Following Sklyanin's paper [21], we consider the most general cyclic solutions of the 6-vertex reflection equation associated to the Bazhanov-Stroganov Lax operator [143]:

$$R_{12}(\lambda/\mu)\,\mathcal{U}_{1,-}(\lambda)R_{21}(\lambda\mu/q)\,\mathcal{U}_{2,-}(\mu) = \mathcal{U}_{2,-}(\mu)R_{21}(\lambda\mu/q)\,\mathcal{U}_{1,-}(\lambda)R_{12}(\lambda/\mu), \quad (1)$$

where the two sides of the equation belong to $\mathrm{End}(V_1\otimes V_2\otimes\mathcal{H})$ and are defined by the following boundary monodromy matrices,

$$\mathcal{U}_{a,-}(\lambda) = M_a(\lambda)K_{a,-}(\lambda)\hat{M}_a(\lambda) = \begin{pmatrix} \mathcal{A}_-(\lambda) & \mathcal{B}_-(\lambda) \\ \mathcal{C}_-(\lambda) & \mathcal{D}_-(\lambda) \end{pmatrix}_a \in \mathrm{End}(V_a\otimes\mathcal{H}), \quad (2)$$

where

$$\hat{M}_a(\lambda) = (-1)^{\mathsf{N}}\,\sigma_a^y\,M_a^{t_a}(1/\lambda)\,\sigma_a^y, \quad (3)$$

and $V_a\simeq\mathbb{C}^2$ is the so-called auxiliary space. Here,

$$M_a(\lambda) = \begin{pmatrix} A(\lambda) & B(\lambda) \\ C(\lambda) & D(\lambda) \end{pmatrix}_a \equiv L_{a,\mathsf{N}}(\lambda q^{-1/2})\cdots L_{a,1}(\lambda q^{-1/2}) \in \mathrm{End}(V_a\otimes\mathcal{H}) \quad (4)$$

is the cyclic solution of the 6-vertex Yang-Baxter equation:

$$R_{12}(\lambda/\mu)M_1(\lambda)M_2(\mu) = M_2(\mu)M_1(\lambda)R_{12}(\lambda/\mu) \in \mathrm{End}(V_1\otimes V_2\otimes\mathcal{H}), \quad (5)$$

associated to the R-matrix

$$R_{ab}(\lambda) = \begin{pmatrix} q\lambda-q^{-1}\lambda^{-1} & 0 & 0 & 0 \\ 0 & \lambda-\lambda^{-1} & q-q^{-1} & 0 \\ 0 & q-q^{-1} & \lambda-\lambda^{-1} & 0 \\ 0 & 0 & 0 & q\lambda-q^{-1}\lambda^{-1} \end{pmatrix} \in \mathrm{End}(V_a\otimes V_b), \quad (6)$$

and defined in terms of the Bazhanov-Stroganov's Lax operators [143]:

$$L_{a,n}(\lambda) \equiv \begin{pmatrix} \lambda\alpha_n v_n - \beta_n\lambda^{-1}v_n^{-1} & u_n\left(q^{-1/2}a_n v_n + q^{1/2}b_n v_n^{-1}\right) \\ u_n^{-1}\left(q^{1/2}c_n v_n + q^{-1/2}d_n v_n^{-1}\right) & \gamma_n v_n/\lambda - \delta_n\lambda/v_n \end{pmatrix}_a \in \text{End}(V_a \otimes \mathscr{R}_n),$$

(7)

where

$$\gamma_n = a_n c_n/\alpha_n, \qquad \delta_n = b_n d_n/\beta_n.$$

(8)

The $u_n \in \text{End}(\mathscr{R}_n)$ and $v_m \in \text{End}(\mathscr{R}_m)$ are unitary Weyl algebra generators,

$$u_n v_m = q^{\delta_{n,m}} v_m u_n \quad \text{with} \quad u_n^p = v_m^p = 1 \ \forall n, m \in \{1, ..., \mathsf{N}\},$$

(9)

and

$$q = e^{-i\pi\beta^2}, \beta^2 = p'/p \text{ with } p' \text{ even and } p = 2l + 1 \text{ odd}, l \in \mathbb{N}.$$

(10)

The local quantum spaces $\mathscr{R}_n$ are $p$-dimensional Hilbert spaces and the full representation space of the cyclic Yang-Baxter and reflection algebra is defined by the tensor product of the local quantum spaces, i.e. $\mathscr{H} = \otimes_{n=1}^{\mathsf{N}}\mathscr{R}_n$. Moreover, we consider here the most general boundary matrices defined as

$$K_{a,\pm}(\lambda) = \begin{pmatrix} a_\pm(\lambda) & b_\pm(\lambda) \\ c_\pm(\lambda) & d_\pm(\lambda) \end{pmatrix}_a \equiv K_a(\lambda q^{(1\pm 1)/2}; \zeta_\pm, \kappa_\pm, \tau_\pm),$$

(11)

where

$$K_a(\lambda; \zeta, \kappa, \tau) = \frac{1}{\zeta - \frac{1}{\zeta}} \begin{pmatrix} \frac{\lambda\zeta}{q^{1/2}} - \frac{q^{1/2}}{\lambda\zeta} & \kappa e^{\tau}\left(\frac{\lambda^2}{q} - \frac{q}{\lambda^2}\right) \\ \kappa e^{-\tau}\left(\frac{\lambda^2}{q} - \frac{q}{\lambda^2}\right) & \frac{q^{1/2}\zeta}{\lambda} - \frac{\lambda}{\zeta q^{1/2}} \end{pmatrix}_a \in \text{End}(V_a).$$

(12)

We introduce the functions

$$\mathsf{A}_-(\lambda) \equiv g_-(\lambda)a(\lambda q^{-1/2})d(1/(q^{1/2}\lambda)), \quad \mathsf{D}_-(\lambda) = k(\lambda)\mathsf{A}_-(q/\lambda),$$

(13)

where

$$a(\lambda) \equiv a_0 \prod_{n=1}^{\mathsf{N}}(\frac{\beta_n}{\lambda} + q^{-1}\frac{b_n\alpha_n}{a_n}\lambda), \quad k(\lambda) = \frac{(\lambda^2 - 1/\lambda^2)}{(\lambda^2/q^2 - q^2/\lambda^2)},$$

(14)

$$d(\lambda) \equiv \frac{(-1)^{\mathsf{N}}}{a_0} \prod_{n=1}^{\mathsf{N}} \frac{a_n c_n}{\alpha_n}(\frac{1}{\lambda} + q\frac{d_n\alpha_n}{c_n\beta_n}\lambda),$$

(15)

where $a_0$ is a free nonzero parameter and

$$g_\epsilon(\lambda) \equiv \frac{(\lambda\alpha_\epsilon/q^{1/2} - q^{1/2}/(\lambda\alpha_\epsilon))(\lambda\beta_\epsilon^{-\epsilon}/q^{1/2} + q^{1/2}/(\lambda\beta_\epsilon^{-\epsilon}))}{(\alpha_\epsilon - 1/\alpha_\epsilon)(\beta_\epsilon + 1/\beta_\epsilon)},$$

(16)

where $\epsilon = \pm 1$ and we have defined

$$(\alpha_\epsilon - 1/\alpha_\epsilon)(\beta_\epsilon + 1/\beta_\epsilon) \equiv \frac{\zeta_\epsilon - 1/\zeta_\epsilon}{\kappa_\epsilon}, \quad (\alpha_\epsilon + 1/\alpha_\epsilon)(\beta_\epsilon - 1/\beta_\epsilon) \equiv \frac{\zeta_\epsilon + 1/\zeta_\epsilon}{\kappa_\epsilon}.$$

(17)

Moreover, later we will use

$$\mu_{n,h} \equiv \begin{cases} iq^{1/2}(a_n\beta_n/\alpha_n b_n)^{1/2} & h = +, \\ iq^{1/2}(c_n\beta_n/\alpha_n d_n)^{1/2} & h = -. \end{cases}$$

(18)

Following the Sklyanin's paper [21] the next proposition holds:

**Proposition 2.1.** *The most general boundary transfer matrix associated to the Bazhanov-Stroganov Lax operator in the cyclic representations of the reflection algebra is defined by*

$$\mathscr{T}(\lambda) \equiv tr_a\{K_{a,+}(\lambda)\mathscr{U}_{a,-}(\lambda)\} \tag{19}$$

$$= a_+(\lambda)\mathscr{A}_-(\lambda) + d_+(\lambda)\mathscr{D}_-(\lambda) + b_+(\lambda)\mathscr{C}_-(\lambda) + c_+(\lambda)\mathscr{B}_-(\lambda). \tag{20}$$

*It is a one parameter family of commuting operators satisfying the following symmetries proprieties:*

$$\mathscr{T}(\lambda) = \mathscr{T}(1/\lambda), \quad \mathscr{T}(-\lambda) = \mathscr{T}(\lambda). \tag{21}$$

*The boundary quantum determinant*

$$det_q\ \mathscr{U}_{a,-}(\lambda) \equiv ((\lambda/q)^2 - (q/\lambda)^2)[\mathscr{A}_-(\lambda q^{1/2})\mathscr{A}_-(q^{1/2}/\lambda) + \mathscr{B}_-(\lambda q^{1/2})\mathscr{C}_-(q^{1/2}/\lambda)] \tag{22}$$

$$= ((\lambda/q)^2 - (q/\lambda)^2)[\mathscr{D}_-(\lambda q^{1/2})\mathscr{D}_-(q^{1/2}/\lambda) + \mathscr{C}_-(\lambda q^{1/2})\mathscr{B}_-(q^{1/2}/\lambda)] \tag{23}$$

*is a central element in the reflection algebra, i.e.*

$$[det_q\ \mathscr{U}_{a,-}(\lambda), \mathscr{U}_{a,-}(\mu)] = 0, \tag{24}$$

*and its explicit expression reads:*

$$det_q\ \mathscr{U}_{a,-}(\lambda) = (\lambda^2/q^2 - q^2/\lambda^2)A_-(\lambda q^{1/2})A_-(q^{1/2}/\lambda). \tag{25}$$

# 3 Gauged cyclic reflection algebra and SoV representations

In our previous paper we solved the spectral problem associated to the transfer matrix of the cyclic representations under the requirement that one of the boundary matrices is triangular, i.e. $b_+(\lambda) \equiv 0$. In this paper we want to solve the same type of spectral problem but for the most general boundary conditions. In order to do so we can follow the same approach used in the case of the transfer matrix associated to the spin-1/2 reflection algebra [80]. That is, we introduce the following linear combinations of the original reflection algebra generators

$$\mathscr{A}_-(\lambda|\beta) = \frac{\left[-\left(\lambda q^{3/2}/\beta\right)\mathscr{A}_-(\lambda) - \alpha q\mathscr{B}_-(\lambda) + \mathscr{C}_-(\lambda)/(\alpha q) + \beta\mathscr{D}_-(\lambda)/\left(\lambda q^{3/2}\right)\right]}{(\beta/q^2 - q^2/\beta)}, \tag{26}$$

$$\mathscr{B}_-(\lambda|\beta) = \frac{\left[-\left(\lambda\beta/q^{1/2}\right)\mathscr{A}_-(\lambda) - \alpha q\mathscr{B}_-(\lambda) + \left(\beta^2/\alpha q\right)\mathscr{C}_-(\lambda) + \left(\beta q^{1/2}/\lambda\right)\mathscr{D}_-(\lambda)\right]}{(\beta - 1/\beta)}, \tag{27}$$

$$\mathscr{C}_-(\lambda|\beta) = \frac{\left[\left(\lambda q^{3/2}/\beta\right)\mathscr{A}_-(\lambda) + \alpha q\mathscr{B}_-(\lambda) - \left(q^3/\alpha\beta^2\right)\mathscr{C}_-(\lambda) - \left(q^{5/2}/\lambda\beta\right)\mathscr{D}_-(\lambda)\right]}{(\beta/q^2 - q^2/\beta)}, \tag{28}$$

$$\mathscr{D}_-(\lambda|\beta) = \frac{\left[\left(\lambda\beta/q^{1/2}\right)\mathscr{A}_-(\lambda) + \alpha q\mathscr{B}_-(\lambda) - \mathscr{C}_-(\lambda)/\alpha q - \left(q^{1/2}/\lambda\beta\right)\mathscr{D}_-(\lambda)\right]}{(\beta - 1/\beta)}, \tag{29}$$

where $\beta \neq \pm 1, \pm q^2$ and $\alpha$ are arbitrary complex values; to simplify the notation, we won't explicit the dependance in $\alpha$. As it is discussed in the appendices A and B, these operators families still satisfy a set of commutation relations which are gauged versions of the reflection algebra commutation relations. In the following we will refer to these families as the gauge transformed reflection algebra generators. In the same appendices, we prove the following theorem, characterizing the representation of these generators:

**Theorem 3.1.** *For almost all the values of the boundary-bulk-gauge parameters there exit a left* $\langle \Omega_\beta |$ *and a right* $|\Omega_\beta\rangle$ *pseudo-eigenstate of* $\mathscr{B}_-(\lambda|\beta)$,

$$\langle \Omega_\beta | \mathscr{B}_-(\lambda|\beta) = {}_{\rm B_0}(\lambda|\beta) \langle \Omega_{\beta/q^2} |, \quad \mathscr{B}_-(\lambda|\beta) | \Omega_\beta\rangle = |\Omega_{q^2\beta}\rangle {}_{\rm B_0}(\lambda|\beta), \tag{30}$$

*where*

$$_{\rm B_{\boldsymbol{h}}}(\lambda|\beta) = {}_{\rm B_-}(\beta)\left(\frac{\lambda^2}{q} - \frac{q}{\lambda^2}\right) \prod_{a=1}^{\rm N} \left( \frac{\lambda}{{\rm B}_{-,a}(\beta)q^{h_a}} - \frac{{\rm B}_{-,a}(\beta)q^{h_a}}{\lambda} \right) \left( \lambda q^{h_a} {\rm B}_{-,a}(\beta) - \frac{1}{\lambda q^{h_a} {\rm B}_{-,a}(\beta)} \right), \tag{31}$$

*for* $\boldsymbol{h} = (h_1, ..., h_{\rm N}) \in \{0, ..., p-1\}^{\rm N}$ *with*

$$\begin{align}
{\rm B}^2_{-,n}(\beta) &\neq q^{1-2h}\alpha_-^{2\epsilon}, \quad {\rm B}^2_{-,n}(\beta) \neq -q^{1-2h}\beta_-^{2\epsilon}, \tag{32}\\
{\rm B}^2_{-,n}(\beta) &\neq q^{1-2h}\mu_{m,+}^{2\epsilon}, \quad {\rm B}^2_{-,n}(\beta) \neq q^{1-2h}\mu_{m,-}^{2\epsilon}, \tag{33}
\end{align}$$

*for any* $\epsilon = \pm 1$, $n, m \in \{1, ..., {\rm N}\}$ *and* $h \in \{1, ..., p-1\}$. *Then, the following set of states:*

$$\langle \beta, h_1, ..., h_{\rm N} | = \frac{1}{{\rm N}_\beta} \langle \Omega_\beta | \prod_{n=1}^{\rm N} \prod_{k_n=1}^{h_n} \frac{\mathscr{A}_-(q^{1-k_n}/{\rm B}_{-,n}(\beta)|\beta q^2)}{{\rm A}_-(q^{1-k_n}/{\rm B}_{-,n}(\beta))}, \tag{34}$$

$$|\beta, h_1, ..., h_{\rm N}\rangle \equiv \frac{1}{{\rm N}_{\beta/q^2}} \prod_{n=1}^{\rm N} \prod_{k_n=1}^{h_n} \frac{\mathscr{D}_-(q^{1-k_n}/{\rm B}_{-,n}(\beta)|\beta)}{{\rm D}_-(q^{1-k_n}/{\rm B}_{-,n}(\beta))} |\Omega_\beta\rangle, \tag{35}$$

*form a left and a right basis of the representation space defining the following decomposition of the identity*

$$\mathbb{I} \equiv \sum_{h_1, ..., h_{\rm N}=0}^{p-1} \prod_{1 \leq b < a \leq {\rm N}} (X_a^{(h_a)} - X_a^{(h_a)}) |\beta q^2, h_1, ..., h_{\rm N}\rangle \langle \beta, h_1, ..., h_{\rm N}|, \tag{36}$$

*with*

$$\langle \beta, h_1, ..., h_{\rm N} | \beta q^2, k_1, ..., k_{\rm N}\rangle = \prod_{1 \leq a \leq {\rm N}} \delta_{h_a, k_a} \prod_{1 \leq b < a \leq {\rm N}} \frac{1}{X_a^{(h_a)} - X_b^{(h_b)}}, \tag{37}$$

*where*

$$X_b^{(h_b)} = ({\rm B}_{-,b}(\beta)q^{h_b})^2 + 1/({\rm B}_{-,b}(\beta)q^{h_b})^2, \tag{38}$$

*for the non-zero normalization fixed by*

$$\rm N_\beta = \left( \prod_{1 \leq b < a \leq {\rm N}} \left( X_a^{(p-1)} - X_b^{(p-1)} \right) \langle \Omega_\beta | \Omega_{\beta q^2}\rangle \right)^{1/2}. \tag{39}$$

*In this basis the operator family* $\mathscr{B}_-(\lambda|\beta)$ *is pseudo-diagonalized,*

$$\begin{align}
\langle \beta, h_1, ..., h_{\rm N} | \mathscr{B}_-(\lambda|\beta) &= {}_{\rm B_{\boldsymbol{h}}}(\lambda|\beta) \langle \beta/q^2, h_1, ..., h_{\rm N}|, \tag{40}\\
\mathscr{B}_-(\lambda|\beta) | \beta, h_1, ..., h_{\rm N}\rangle &= |q^2\beta, h_1, ..., h_{\rm N}\rangle {}_{\rm B_{\boldsymbol{h}}}(\lambda|\beta), \tag{41}
\end{align}$$

*with simple pseudo-spectrum*

$${\rm B}^{2p}_{-,n}(\beta) \neq \pm 1, \quad {\rm B}^p_{-,m}(\beta) \neq {\rm B}^p_{-,n}(\beta), \quad \forall n \neq m \in \{1, ..., {\rm N}\}, \tag{42}$$

*and the operator families* $\mathscr{A}_-(\lambda|\beta)$ *and* $\mathscr{D}_-(\lambda|\beta)$ *in the zeros* $\zeta_a^{(h_a)}$ *of* $\mathscr{B}_-(\lambda|\beta)$ *act as simple shift operators:*

$$\langle \beta, h_1, ..., h_{\rm N} | \mathscr{A}_-(\zeta_a^{(h_a)}|\beta q^2) = {\rm A}_-(\zeta_a^{(h_a)}) \langle \beta, h_1, ..., h_{\rm N} | T_a^{-\varphi_a}, \tag{43}$$

$$\mathscr{D}_-(\zeta_a^{(h_a)}|\beta) | \beta, h_1, ..., h_{\rm N}\rangle = T_a^{-\varphi_a} | \beta, h_1, ..., h_{\rm N}\rangle {\rm D}_-(\zeta_a^{(h_a)}), \tag{44}$$

*where*

$$\langle \beta, h_1, ..., h_a, ..., h_N | T_a^{\pm} = \langle \beta, h_1, ..., h_a \pm 1, ..., h_N |, \tag{45}$$

$$T_a^{\pm} | \beta, h_1, ..., h_a, ..., h_N \rangle = | \beta, h_1, ..., h_a \pm 1, ..., h_N \rangle, \tag{46}$$

*and*

$$\zeta_n^{(h)} = \left( B_{-,n}(\beta) q^h \right)^{\varphi_n} \ \text{for} \ h \in \{0, ..., p-1\} \ \text{and} \ \forall n \in \{1, ..., 2N\}, \tag{47}$$

$$\varphi_a = 1 - 2\theta(a - N) \ \text{with} \ \theta(x) = \{0 \text{ for } x \le 0, \ 1 \text{ for } x > 0\}. \tag{48}$$

Let us comment that the existence of the states $\langle \Omega_{\beta} |$ and $| \Omega_{\beta} \rangle$ can be proven by a general argument which we present in Appendix B. For general representations, the pseudo-spectrum of $\mathscr{B}_-(\lambda | \beta)$, i.e. the values of $B_{-,n}(\beta)$ and $B_-(\beta)$, must be computed by recursion on the number of sites. However, in Appendix B we present the explicit expression for $B_{-,n}(\beta)$ and $B_-(\beta)$ in some particular representations.

The interest in these gauge transformed boundary generators is due to the possibility to use them to rewrite the transfer matrix associated to the most general cyclic 6-vertex reflection algebra representations in a simple form, as presented in the following proposition:

**Proposition 3.1.** *The quantum determinant can be written in terms of the gauge transformed boundary generators as*

$$\frac{det_q \ \mathscr{U}_-(\lambda)}{(\lambda^2/q^2 - q^2/\lambda^2)} = \mathscr{A}_-(q^{1/2}\lambda^{\epsilon} | \beta q^2) \mathscr{A}_-(q^{1/2}/\lambda^{\epsilon} | \beta q^2) + \mathscr{B}_-(q^{1/2}\lambda^{\epsilon} | \beta) \mathscr{C}_-(q^{1/2}/\lambda^{\epsilon} | \beta q^2) \tag{49}$$

$$= \mathscr{D}_-(q^{1/2}\lambda^{\epsilon} | \beta) \mathscr{D}_-(q^{1/2}/\lambda^{\epsilon} | \beta) + \mathscr{C}_-(q^{1/2}\lambda^{\epsilon} | \beta q^2) \mathscr{B}_-(q^{1/2}/\lambda^{\epsilon} | \beta), \tag{50}$$

$\epsilon = \pm 1$. *Moreover, if we set the gauge parameter $\alpha$ to*

$$\alpha = -\beta \beta_+ / q^2 \alpha_+ e^{\tau_+}, \tag{51}$$

*($\alpha_+$ and $\beta_+$ are defined in (17), they are linked to the boundary parameters $\zeta_+, \kappa_+$ and $\tau_+$, see (11)-(12)) then the transfer matrix can be written as*

$$\mathscr{T}(\lambda) = a_+(\lambda) \mathscr{A}_-(\lambda | \beta) + a_+(1/\lambda) \mathscr{A}_-(1/\lambda | \beta) + q c_+(\lambda | \beta) \mathscr{B}_-(\lambda | \beta/q^2) \tag{52}$$

$$\mathscr{T}(\lambda) = d_+(\lambda) \mathscr{D}_-(\lambda | \beta) + d_+(1/\lambda) \mathscr{D}_-(1/\lambda | \beta) + c_+(\lambda | \beta) \mathscr{B}_-(\lambda | \beta)/q, \tag{53}$$

*where we have defined*

$$a_+(\lambda) = -\frac{\lambda^2 q - 1/q\lambda^2}{\lambda^2 - 1/\lambda^2} g_+(\lambda), \quad d_+(\lambda) = \frac{\lambda^2 q - 1/q\lambda^2}{\lambda^2 - 1/\lambda^2} g_+(q/\lambda), \tag{54}$$

$$c_+(\lambda | \beta) = \frac{-q(\lambda^2 q - 1/q\lambda^2)(\beta \beta_+/q\alpha_+ - q\alpha_+/\beta \beta_+)}{\beta (\alpha_+ - 1/\alpha_+)(\beta_+ + 1/\beta_+)}. \tag{55}$$

*Proof.* The proof of this statement coincides with the one given in [80] for the XXZ spin 1/2 quantum chain with general integrable boundaries; in fact, this statement is representation independent. The only difference is that here we have used a Laurent polynomial form while in the XXZ case it was a trigonometric form. $\square$

The simple representations (52)-(53) of the transfer matrix in terms of the gauge transformed boundary generators and the known actions (43)-(44) of these operators imply that the transfer matrix spectral problem is separated in the pseudo-eigenbasis of $\mathscr{B}_-(\lambda | \beta)$.

# 4 $\mathscr{T}$-spectrum characterization in SoV basis and scalar products

In this section we present the complete characterization of the spectrum of the transfer matrix $\mathscr{T}(\lambda)$ associated to the cyclic representations of the 6-vertex reflection algebra. We first present some preliminary properties satisfied by all the eigenvalue functions of the transfer matrix $\mathscr{T}(\lambda)$:

**Lemma 4.1.** *Denote by $\Sigma_{\mathscr{T}}$ the transfer matrix spectrum, then any $\tau(\lambda) \in \Sigma_{\mathscr{T}}$ is an even function of $\lambda$ symmetrical under the transformation $\lambda \to 1/\lambda$ which admits the following interpolation formula*

$$
\tau(\lambda) = \sum_{a=1}^{\mathsf{N}} \frac{\Lambda^2 - X^2}{(X_a^{(0)})^2 - X^2} \prod_{\substack{b=1 \\ b \neq a}}^{\mathsf{N}} \frac{\Lambda - X_b^{(0)}}{X_a^{(0)} - X_b^{(0)}} \tau(\zeta_a^{(0)}) + (-1)^{\mathsf{N}} \frac{(\Lambda + X)}{2} \prod_{b=1}^{\mathsf{N}} \frac{\Lambda - X_b^{(0)}}{X - X_b^{(0)}} det_q M(1)
$$
$$
- (-1)^{\mathsf{N}} \frac{(\Lambda - X)}{2} \prod_{b=1}^{\mathsf{N}} \frac{\Lambda - X_b^{(0)}}{X + X_b^{(0)}} \frac{(\zeta_+ + 1/\zeta_+)}{(\zeta_+ - 1/\zeta_+)} \frac{(\zeta_- + 1/\zeta_-)}{(\zeta_- - 1/\zeta_-)} det_q M(i)
$$
$$
+ (\Lambda^2 - X^2) \tau_\infty \prod_{b=1}^{\mathsf{N}} (\Lambda - X_b^{(0)}), \tag{56}
$$

*where*

$$
\Lambda \equiv (\lambda^2 + 1/\lambda^2) \quad and \quad X \equiv q + 1/q, \tag{57}
$$

*and*

$$
\tau_\infty \equiv \frac{\kappa_+ \kappa_- (e^{\tau_+ - \tau_-} \prod_{b=1}^{\mathsf{N}} \delta_b \gamma_b + e^{\tau_- - \tau_+} \prod_{b=1}^{\mathsf{N}} \alpha_b \beta_b)}{(\zeta_+ - 1/\zeta_+)(\zeta_- - 1/\zeta_-)}. \tag{58}
$$

*We recall that $\zeta_-, \kappa_-, \tau_-, \zeta_+, \kappa_+$ and $\tau_+$ are the boundary parameters, see (11)-(12), and $\alpha_n, \beta_n, \gamma_n$ and $\delta_n$ are the bulk parameters, see (7).*

*Proof.* This lemma coincides with Lemma 5.1 of our previous paper. □

We introduce the following one-parameter family $D_\tau(\lambda)$ of $p \times p$ matrices,

$$
D_\tau(\lambda) \equiv \begin{pmatrix}
\tau(\lambda) & -\mathrm{A}(1/\lambda) & 0 & \cdots & 0 & -\mathrm{A}(\lambda) \\
-\mathrm{A}(q\lambda) & \tau(q\lambda) & -\mathrm{A}(1/(q\lambda)) & 0 & \cdots & 0 \\
0 & & \ddots & & & \vdots \\
\vdots & & & \cdots & & \vdots \\
\vdots & & & & \cdots & \vdots \\
\vdots & & & & \ddots & 0 \\
0 & \cdots & 0 & -\mathrm{A}(q^{2l-1}\lambda) & \tau(q^{2l-1}\lambda) & -\mathrm{A}(1/(q^{2l-1}\lambda)) \\
-\mathrm{A}(1/(q^{2l}\lambda)) & 0 & \cdots & 0 & -\mathrm{A}(q^{2l}\lambda) & \tau(q^{2l}\lambda)
\end{pmatrix}, \tag{59}
$$

where for now $\tau(\lambda)$ is a generic function and we have defined

$$
\mathrm{A}(\lambda) = \mathsf{a}_+(\lambda)\mathrm{A}_-(\lambda), \tag{60}
$$

(from (13) and (54)) where the coefficient $\mathrm{A}(\lambda)$ satisfies the quantum determinant condition

$$
\mathrm{A}(\lambda q^{1/2})\mathrm{A}(q^{1/2}/\lambda) = \frac{\mathsf{a}_+(\lambda q^{1/2})\mathsf{a}_+(q^{1/2}/\lambda) det_q \mathscr{U}_-(\lambda)}{(\lambda/q)^2 - (q/\lambda)^2}. \tag{61}
$$

The separation of variables lead to the following discrete characterization of the transfer matrix spectrum.

**Theorem 4.1.** *For almost all the values of the boundary-bulk parameters $\mathcal{T}(\lambda)$ is diagonalizable and it has simple spectrum and $\Sigma_{\mathcal{T}}$ coincides with the set of polynomials $\tau(\lambda)$ of the form (56) which satisfy the following discrete system of equations:*

$$\det D_{\tau}(\zeta_a^{(0)}) = 0, \ \forall a \in \{1, ..., \mathsf{N}\}. \tag{62}$$

*I) The right $\mathcal{T}$-eigenstate corresponding to $\tau(\lambda) \in \Sigma_{\mathcal{T}}$ is defined by the following decomposition in the right SoV-basis:*

$$|\tau\rangle = \sum_{h_1,...,h_\mathsf{N}=0}^{p-1} \prod_{a=1}^{\mathsf{N}} q_{\tau,a}^{(h_a)} \prod_{1 \le b < a \le \mathsf{N}} (X_a^{(h_a)} - X_b^{(h_b)}) |\beta, h_1, ..., h_\mathsf{N}\rangle, \tag{63}$$

*where the gauge parameters $\alpha$ and $\beta$ satisfy the condition (51) and the $q_{\tau,a}^{(h_a)}$ are the unique nontrivial solutions up to normalization of the linear homogeneous system,*

$$D_{\tau}(\zeta_a^{(0)}) \begin{pmatrix} q_{\tau,a}^{(0)} \\ \vdots \\ q_{\tau,a}^{(p-1)} \end{pmatrix} = \begin{pmatrix} 0 \\ \vdots \\ 0 \end{pmatrix}. \tag{64}$$

*II) The left $\mathcal{T}$-eigenstate corresponding to $\tau(\lambda) \in \Sigma_{\mathcal{T}}$ is defined by the following decomposition in the left SoV-basis:*

$$\langle \tau | = \sum_{h_1,...,h_\mathsf{N}=0}^{p-1} \prod_{a=1}^{\mathsf{N}} \hat{q}_{\tau,a}^{(h_a)} \prod_{1 \le b < a \le \mathsf{N}} (X_a^{(h_a)} - X_b^{(h_b)}) \langle h_1, ..., h_\mathsf{N}, \beta/q^2 |, \tag{65}$$

*where the gauge parameters $\alpha$ and $\beta$ satisfy the condition (51) and the $\hat{q}_{\tau,a}^{(h_a)}$ are the unique nontrivial solutions up to normalization of the linear homogeneous system,*

$$\begin{pmatrix} \hat{q}_{\tau,a}^{(0)} & \cdots & \hat{q}_{\tau,a}^{(p-1)} \end{pmatrix} \left( \hat{D}_{\tau}(\zeta_a^{(0)}) \right)^{t_0} = \begin{pmatrix} 0 & \cdots & 0 \end{pmatrix}, \tag{66}$$

*and $\hat{D}_{\tau}(\lambda)$ is the family of $p \times p$ matrices defined substituting in $D_{\tau}(\lambda)$ the coefficient $\mathrm{A}(\lambda)$ with*

$$\mathrm{D}(\lambda) = \mathsf{d}_+(\lambda)\mathsf{D}_-(\lambda), \tag{67}$$

*defined from (13) and (54).*

*Proof.* The Theorem 3.1 implies that for almost all the values of the gauge-boundary-bulk parameters the conditions (32)-(33) hold. Here, we need to prove also that for almost all the values of the boundary-bulk parameters we have,

$$\mathrm{B}_{-,n}^2(\beta) \ne q^{1-2h}\alpha_+^{\pm 2}, \ \mathrm{B}_{-,n}^2(\beta) \ne -q^{1-2h}\beta_+^{\pm 2}, \ \forall h \in \{1, ..., p-1\}, n \in \{1, ..., \mathsf{N}\}, \tag{68}$$

once we set the ratio $\alpha/\beta$ as in (51). Let us first observe that $\mathcal{B}_-(\lambda|\beta)$ is a Laurent polynomial in $\alpha$, $\beta$, the inner boundary parameters and the bulk parameters. So that by (51), the one parameter family $\mathcal{B}_-(\lambda|\beta)$ becomes Laurent polynomial in the outer boundary parameters too. Consequently, to prove that (68) is satisfied for almost all the values of the boundary-bulk parameters it is enough to prove that we can find some values of these parameters for which (68) is satisfied. Indeed, we can chose arbitrary boundary-bulk parameters satisfying the following inequalities,

$$\mu_{+,n}^p \ne \alpha_+^{\pm p} \text{ and } \mu_{+,n}^p \ne -\beta_+^{\pm p}, \ \forall n \in \{1, ..., \mathsf{N}\}, \tag{69}$$

together with those in (323) and (324) and impose the N conditions (322). Under these conditions, Theorem 3.1 implies the pseudo-diagonalizability of $\mathcal{B}_-(\lambda|\beta)$ and fixes the spectrum of its zeros $\mathrm{B}_{-,n}(\beta)$ by (325); so that the inequality (68) is satisfied.

As we have proven that for almost all the values of the boundary-bulk parameters the inequalities (68), (323) and (324) hold, to prove this theorem we have just to follow the same proof given in the non-gauged case, i.e. the proof of Theorem 5.1 of our previous paper.

Let us comment that with respect to this last theorem here we are stating also the diagonalizability of the transfer matrix for almost any value of the parameters of the representation. This last statement can be proven as it follows. Let us consider the following special representation, where the bulk parameters satisfy

$$c_n = -b_n^* \; ; \; d_n = -a_n^* \text{ and } \alpha_n^* \beta_n = a_n^* b_n \,, \tag{70}$$

and where the boundary matrices are diagonal, $K_{a,-}(\lambda) = K_a(\lambda; \zeta_-, 0, 0)$ and $K_{a,+}(\lambda) = K_a(q\lambda; \zeta_+, 0, 0)$ (see (11)-(12)), with the associated boundary parameters satisfying moreover $|\zeta_-| = |\zeta_+| = 1$. The $*$ operation is the complex conjugation. A simple direct calculation made for example in [129] leads to the following Hermitian conjugate of the monodromy matrix (4):

$$M_a^\dagger(\lambda) = \sigma_a^y M_a(\lambda^*) \sigma_a^y, \tag{71}$$

where $\sigma_a^y$ denotes the Pauli matrix.

From this relation, and using the specific inner boundary matrix introduced, one can compute the Hermitian conjugate of the boundary monodromy matrix (2),

$$\mathcal{U}_{a,-}^\dagger(\lambda) = \mathcal{U}_{a,-}^{t_a}(1/\lambda^*). \tag{72}$$

Then, from the definition of the boundary transfer matrix, and for the special choice of representation here chosen, we can show

$$\mathcal{T}^\dagger(\lambda) = \mathcal{T}(1/\lambda^*). \tag{73}$$

Thus for this special representation the boundary transfer matrix is normal. Then it follows that the determinant of the $p^N \times p^N$ matrix of elements $\langle e_i | \tau_j \rangle$, where $\langle e_i |$ is the generic element of a given basis of covectors and $| \tau_j \rangle$ is the generic transfer matrix eigenvector, is non zero.

Noticing that this determinant is a ratio of algebraic functions of the bulk and boundary parameters, non zero for the special choice of the parameters above defined, it follows that it is non zero for almost every choice of the parameters. Which concludes the proof. □

It is also interesting to remark that we can obtain the coefficients of a left transfer matrix eigenstates in terms of those of the right one. The following lemma defines this characterization and can be proven as in the standard case [1]:

**Lemma 4.2.** *Let* $\tau(\lambda) \in \Sigma_{\mathcal{T}}$ *then it holds*

$$\frac{\hat{q}_{\tau,a}^{(h)}}{\hat{q}_{\tau,a}^{(h-1)}} = \frac{\mathrm{A}(1/\zeta_a^{(h-1)})}{\mathrm{D}(1/\zeta_a^{(h-1)})} \frac{q_{\tau,a}^{(h)}}{q_{\tau,a}^{(h-1)}}. \tag{74}$$

Let us introduce a class of left and right states, the so-called separate states, characterised by the following type of decompositions in the left and right separate basis:

$$\langle \alpha | \equiv \sum_{h_1,\dots,h_N=0}^{p-1} \prod_{a=1}^{N} \alpha_a^{(h_a)} \prod_{1 \le b < a \le N} (X_a^{(h_a)} - X_b^{(h_b)}) \langle \beta/q^2, h_1, \dots, h_N |, \tag{75}$$

$$| \beta \rangle = \sum_{h_1,\dots,h_N=0}^{p-1} \prod_{a=1}^{N} \beta_a^{(h_a)} \prod_{1 \le b < a \le N} (X_a^{(h_a)} - X_b^{(h_b)}) | \beta, h_1, \dots, h_N \rangle, \tag{76}$$

where the coefficients $\alpha_a^{(h_a)}$ and $\beta_a^{(h_a)}$ are arbitrary complex numbers, meaning that the coefficients of these separate states have a factorised form in these basis. ($X_a^{(h_a)}$ defined in (38)).

These separate states are interesting at least for two reasons: the eigenstates of the boundary transfer matrix are special separate states, and they admit a simple determinant scalar product, as it is stated in the next proposition:

**Proposition 4.1.** *Let us take an arbitrary separate left state $\langle\alpha|$ and an arbitrary separate right state $|\beta\rangle$. Then it holds*

$$\langle\alpha|\beta\rangle = det\ \mathcal{M}^{(\alpha,\beta)}, \tag{77}$$

*where the elements of the size* N *matrix* $\mathcal{M}^{(\alpha,\beta)}$ *are given by*

$$\forall(a,b)\in[1,N]^2,\ \ \mathcal{M}_{a,b}^{(\alpha,\beta)} \equiv \sum_{h=0}^{p-1}\alpha_a^{(h)}\beta_a^{(h)}(X_a^{(h)})^{(b-1)}. \tag{78}$$

The proof is quite straightforward, it is based on the fact that one can see a Vandermonde determinant when computing the scalar product. One of the main corollary is the orthogonality of two eigenstates $\langle\tau|$ and $|\tau'\rangle$ of the boundary transfer matrix associated to two different eigenvalues $\tau(\lambda)$ and $\tau'(\lambda)$,

$$\langle\tau|\tau'\rangle = 0. \tag{79}$$

The computation of such scalar products is the very first step towards the dynamics, several further steps being required to reach this characterization for the models associated to cyclic representations of the 6-vertex reflection algebra: the reconstruction of the local operators in separate variables, the identification of the ground state, the homogeneous and the thermodynamic limit. For example a rewriting of the determinant representations for the form factors obtained from separation of variable will be necessary to overcome the standard problems related to the homogeneous limit. This problem has been addressed and solved for the XXX spin 1/2 chain, linking the separation of variable type determinants with Izergin's, Slavnov's and Gaudin's type determinants [140,141].

# 5 Functional equation characterizing the $\mathcal{T}$-spectrum

The purpose of this section is to characterize the spectrum by functional relations analogous to Baxter's T-Q equation. To begin with, we first need the following property.

**Lemma 5.1.** *Let $\tau(\lambda)$ be a function of $\lambda$ invariant under the transformation $\lambda \to 1/\lambda$ and $\lambda \to -\lambda$ then $det_p D_\tau(\lambda)$ (from (59)) is a function of*

$$Z = \lambda^{2p} + \frac{1}{\lambda^{2p}}, \tag{80}$$

*i.e. it is a function of $\lambda^p$ invariant under the transformations $\lambda^p \to 1/\lambda^p$ and $\lambda \to -\lambda$. Moreover, if $\tau(\lambda)$ is a Laurent polynomial of degree $N+2$ in $\Lambda = \lambda^2 + \frac{1}{\lambda^2}$ then $det_p D_\tau(\lambda)$ is a Laurent polynomial of degree $N+2$ in $Z$.*

*Proof.* The first part of this lemma about the dependence w.r.t. $Z$ of $det_p D_\tau(\lambda)$ has been already proven in Lemma 5.2 of our previous paper [1] while the second part of this lemma can be proven following the proof given in Proposition 6.1 of the same paper. To adapt this proof here, let us observe that the matrix $D_\tau(i^a q^{h+1/2})$ for $a \in \{0,1\}$ and $h \in \{0,...,p-1\}$ contains one row with two divergent elements, i.e. $-A(\pm 1)$ and $-A(\pm i)$, respectively for $a = 0$ and

$a = 1$. Nevertheless the determinants $\det_p D_\tau(i^a q^{h+1/2})$ are all finites for any $a \in \{0,1\}$ and $h \in \{0,...,p-1\}$ if $\tau(i^b q^{k+1/2})$ are finite for any $b \in \{0,1\}$ and $k \in \{0,...,p-1\}$. Indeed, by the symmetries $\lambda^p \to 1/\lambda^p$ and $\lambda \to -\lambda$ all the determinants $\det_p D_\tau(q^{h+1/2})$ coincide as well as all the determinants $\det_p D_\tau(i q^{h+1/2})$. So that we have to prove our statement for one value of $q^{h+1/2}$ and one value of $i q^{h+1/2}$. Now, we can use the expansion of the determinant w.r.t. the central row,

$$
\det_p D_\tau(\lambda q^{1/2}) = \tau(\lambda)\det_{p-1} D_{\tau,(p+1)/2,(p+1)/2}(\lambda q^{1/2}) +
$$
$$
\frac{\text{x}(\lambda)\det_{p-1} D_{\tau,(p+1)/2,(p+1)/2-1}(\lambda q^{1/2})}{\lambda^2 - 1/\lambda^2} - \frac{\text{x}(1/\lambda)\det_{p-1} D_{\tau,(p+1)/2,(p+1)/2+1}(\lambda q^{1/2})}{\lambda^2 - 1/\lambda^2}, \quad (81)
$$

where

$$
\text{x}(\lambda) = \left(\lambda^2 - \frac{1}{\lambda^2}\right)\text{A}(\lambda), \quad (82)
$$

and $D_{\tau,i,j}(\lambda)$ denotes the $(p-1) \times (p-1)$ matrix obtained from $D_\tau(\lambda)$ removing the row $i$ and the column $j$. From the identity

$$
\det_{p-1} D_{\tau,(p+1)/2,(p+1)/2+1}(\lambda q^{1/2}) = \det_{p-1} D_{\tau,(p+1)/2,(p+1)/2-1}(q^{1/2}/\lambda), \quad (83)
$$

and the regularity of these two determinants for $\lambda \to \pm 1$ and $\lambda \to \pm i$, it follows that $\det_p D_\tau(i^a q^{1/2})$ are finites too for $a \in \{0,1\}$. Now, our statement about the Laurent polynomiality of degree $N+2$ of $\det_p D_\tau(\lambda)$ w.r.t. $Z$ follows from the symmetries and from the fact that $\tau(\lambda)$ and $\text{x}(\lambda)$ are Laurent polynomials in $\lambda$ of degree $2N+4$.

□

Let us introduce the following notations:

$$
\text{A}_\infty = \lim_{\lambda \to +\infty} \lambda^{-2(N+2)}\text{A}(\lambda) = \frac{(-1)^{N+1}\kappa_+\kappa_-\alpha_-\beta_-\alpha_+ \prod_{n=1}^N b_n c_n}{q^{3+N}\beta_+(\zeta_+ - 1/\zeta_+)(\zeta_- - 1/\zeta_-)}, \quad (84)
$$

$$
\text{A}_0 = \lim_{\lambda \to 0} \lambda^{2(N+2)}\text{A}(\lambda) = \frac{(-1)^{N+1}q^{3+N}\kappa_+\kappa_-\beta_+ \prod_{n=1}^N a_n d_n}{\alpha_-\beta_-\alpha_+(\zeta_+ - 1/\zeta_+)(\zeta_- - 1/\zeta_-)}, \quad (85)
$$

and

$$
F(\lambda) = \prod_{b=1}^{2N}\left(\frac{\lambda^p}{\left(\zeta_b^{(0)}\right)^p} - \frac{\left(\zeta_b^{(0)}\right)^p}{\lambda^p}\right), \quad (86)
$$

where we recall that $\zeta_-, \kappa_-, \tau_-, \alpha_-, \beta_-, \zeta_+, \kappa_+, \tau_+, \alpha_+$ and $\beta_+$ are the boundary parameters (see (11),(12) and (17)), while $a_n, b_n, c_n$ and $d_n$ are the bulk parameters, see (7). Then the following results hold:

**Proposition 5.1.** *For almost all the values of the boundary-bulk parameters, $\mathcal{T}(\lambda)$ has simple spectrum and $\tau(\lambda)$ of the form (56) is an element of $\Sigma_{\mathcal{T}}$ (the set of the eigenvalues of $\mathcal{T}(\lambda)$) if and only if $\det_p D_\tau(\lambda)$ is a Laurent polynomial of degree $N+2$ in the variable $Z$ (see (80)) which satisfies the following functional equation:*

$$
det_p D_\tau(\lambda) = F(\lambda)\left(\lambda^{2p} - \frac{1}{\lambda^{2p}}\right)^2 \prod_{k=0}^{p-1}(\tau_\infty - (q^k \text{A}_\infty + q^{-k}\text{A}_0)). \quad (87)
$$

*Proof.* The SoV characterization of the spectrum implies that $\tau(\lambda) \in \Sigma_{\mathcal{T}}$ if and only if it holds

$$
\det_p D_\tau(\zeta_a^{(0)}) = 0, \quad \forall a \in \{1,...,N\}, \quad (88)
$$

and $\tau(\lambda)$ has the form (56). In the previous lemma we have shown that $\det_p D_\tau(\lambda)$ is a Laurent polynomial of degree $N + 2$ in $Z$, here we show that from $\tau(\lambda)$ of form (56) it follows the identities

$$\lim_{\lambda \to \pm 1, \pm i} \det_p D_\tau(\lambda q^{1/2+h}) = 0 \quad \forall h \in \{0, ..., p-1\}. \tag{89}$$

For the symmetry it is enough to consider the above limit in the case $h = 0$. Let us denote with $\bar{D}_\tau(\lambda q^{1/2})$ the matrix whose first row is the sum of the first and the last row of $D_\tau(\lambda q^{1/2})$ divided for $(\lambda^2 - 1/\lambda^2)$ and whose row $(p+1)/2$ is the row $(p+1)/2$ of $D_\tau(\lambda q^{1/2})$ multiplied for $(\lambda^2 - 1/\lambda^2)$ while all the others rows of $\bar{D}_\tau(\lambda q^{1/2})$ and $D_\tau(\lambda q^{1/2})$ coincide. Clearly it holds

$$\det_p \bar{D}_\tau(\lambda q^{1/2}) = \det_p D_\tau(\lambda q^{1/2}), \tag{90}$$

so that we can compute the limits directly for $\det_p \bar{D}_\tau(\lambda q^{1/2})$. The interesting point is that now all the rows of the matrix $\bar{D}_\tau(\lambda q^{1/2})$ are finites in the limits $\lambda \to \pm 1, \pm i$, this is a consequence of the identities

$$\tau(\pm i^a q^{1/2}) = A(\pm i^a q^{1/2}), \quad A(\pm i^a q^{-1/2}) = 0, \quad \forall a \in \{0, 1\}. \tag{91}$$

Explicitly, we have that the nonzero elements of the rows $1$, $(p+1)/2$ and $p$ are

$$\left[\bar{D}_\tau(\pm i^a q^{1/2})\right]_{1,1} = r_{a,\pm}, \quad \left[\bar{D}_\tau(\pm i^a q^{1/2})\right]_{1,2} = s_{a,\pm}, \tag{92}$$

$$\left[\bar{D}_\tau(\pm i^a q^{1/2})\right]_{1,p-1} = -s_{a,\pm}, \quad \left[\bar{D}_\tau(\pm i^a q^{1/2})\right]_{1,p} = -r_{a,\pm}, \tag{93}$$

$$\left[\bar{D}_\tau(\pm i^a q^{1/2})\right]_{(p+1)/2,(p+1)/2-1} = -X(\pm i^a), \tag{94}$$

$$\left[\bar{D}_\tau(\pm i^a q^{1/2})\right]_{(p+1)/2,(p+1)/2+1} = X(\pm i^a), \tag{95}$$

$$\left[\bar{D}_\tau(\pm i^a q^{1/2})\right]_{p,1} = -\tau(\pm i^a q^{1/2}), \quad \left[\bar{D}_\tau(\pm i^a q^{1/2})\right]_{p,p} = \tau(\pm i^a q^{1/2}), \tag{96}$$

where we have defined

$$r_{a,\pm} = (-1)^a \lim_{\lambda \to \pm 1} \frac{\tau(i^a q^{1/2}\lambda) - A(i^a q^{1/2}/\lambda)}{\lambda^2 - 1/\lambda^2}, \quad s_{a,\pm} = (-1)^a \lim_{\lambda \to \pm 1} \frac{A(i^{-a} q^{-1/2}/\lambda)}{\lambda^2 - 1/\lambda^2}. \tag{97}$$

The remaining rows of $\bar{D}_\tau(\pm i^a q^{1/2})$ produce the tridiagonal part of this matrix. Then, it is possible to prove that this matrix has linear dependent rows; so that $\det_p \bar{D}_\tau(\pm i^a q^{1/2}) = 0$. Finally, we can compute the following asymptotic formulae:

$$\Delta_\infty \equiv \lim_{\lambda \to \infty} \lambda^{-2p(N+2)} \det_p D_\tau(\lambda) = \det_p \left[ \lim_{\lambda \to \infty} \lambda^{-2(N+2)} D_\tau(\lambda) \right] \tag{98}$$

$$= \lim_{\lambda \to 0} \lambda^{2p(N+2)} \det_p D_\tau(\lambda) = \det_p \left[ \lim_{\lambda \to 0} \lambda^{2(N+2)} D_\tau(\lambda) \right]^t \tag{99}$$

$$= \det_p \begin{pmatrix} \tau_\infty & -A_0 & 0 & \cdots & 0 & -A_\infty \\ -x A_\infty & x\tau_\infty & -x A_0 & 0 & \cdots & 0 \\ 0 & -x^2 A_\infty & x^2\tau_\infty & -x^2 A_0 & \ddots & \vdots \\ \vdots & & \ddots & \ddots & 0 & 0 \\ 0 & \cdots & 0 & -x^{2l-1} A_\infty & x^{2l-1}\tau_\infty & -x^{2l-1} A_0 \\ -x^{2l} A_0 & 0 & \cdots & 0 & -x^{2l} A_\infty & x^{2l}\tau_\infty \end{pmatrix}, \tag{100}$$

where we have denoted with $^t$ the transpose of the matrix and $x = q^{2(N+2)}$. We have that $\Delta_\infty$ is a degree $p$ polynomial in $\tau_\infty$ whose zeros are known from the identities

$$\Delta_\infty|_{\tau_\infty = q^k A_\infty + q^{-k} A_0} = 0 \quad \forall k \in \{0, ..., p-1\}, \tag{101}$$

so that we get

$$\Delta_\infty = \prod_{k=0}^{p-1} (\tau_\infty - (q^k A_\infty + q^{-k} A_0)). \tag{102}$$

This means that we have determined $\det_p D_\tau(\lambda)$ in $N+2$ different values of $Z$ together with the asymptotic for $Z \to \infty$. From which the characterization (87) trivially follows. $\qquad\square$

The discrete characterization of the spectrum given in Theorem 4.1 can be reformulated in terms of Baxter's type T-Q functional equations and the eigenstates admit an algebraic Bethe ansatz like reformulation, as we show in the next theorem. These type of reformulations of the spectrum holds for several models once they admit SoV description, see for example [81, 127–129, 137–139].

In the following we denote with $Q(\lambda)$ a polynomial in $\Lambda = \lambda^2 + \frac{1}{\lambda^2}$ of degree $N_Q$ of the form:

$$Q(\lambda) = \prod_{b=1}^{N_Q} (\Lambda - \Lambda_b). \tag{103}$$

**Theorem 5.1.** *For almost all the values of the boundary-bulk parameters such that*

$$\tau_\infty \neq q^{-k} A_\infty + q^k A_0 \quad \forall k \in \{0, ..., p-1\}, \tag{104}$$

$\tau(\lambda) \in \Sigma_{\mathcal{T}}$ *(the set of the eigenvalues of $\mathcal{T}(\lambda)$) if and only if $\tau(\lambda)$ is an entire function and there exists and is unique a polynomial $Q(\lambda)$ of the form (103) with $N_Q = (p-1)N$, satisfying the following functional equation,*

$$\tau(\lambda)Q(\lambda) = A(\lambda)Q(\lambda/q) + A(1/\lambda)Q(\lambda q) + \left[\tau_\infty - (q^{-N_Q} A_\infty + q^{N_Q} A_0)\right](\Lambda^2 - X^2)F(\lambda), \tag{105}$$

*and the conditions*

$$(Q(\zeta_a^{(0)}), ..., Q(\zeta_a^{(p-1)})) \neq (0, ..., 0) \quad \forall a \in \{1, ..., N\}. \tag{106}$$

*We recall that $A_\infty, A_0$ and $F(\lambda)$ are defined in (84)-(86) and that $X = q + 1/q$ (57).*

*Proof.* Let us prove first that if it exists a $Q(\lambda)$ of the form (103) with $N_Q = (p-1)N$ satisfying (106) and (105) with $\tau(\lambda)$ an entire function, then $\tau(\lambda) \in \Sigma_{\mathcal{T}}$. The r.h.s of the equation (105) is a Laurent polynomial in $\lambda$ as we have

$$A(\lambda)Q(\lambda/q) + A(1/\lambda)Q(\lambda q) = \frac{X(\lambda)Q(q/\lambda) - X(1/\lambda)Q(\lambda q)}{\lambda^2 - 1/\lambda^2}, \tag{107}$$

which is finite in the limits $\lambda \to \pm 1$, $\lambda \to \pm i$. So that the r.h.s. of (105) is a polynomial of degree $pN + 2$ in $\Lambda$, as it is invariant w.r.t. the transformations $\lambda \to -\lambda$ and $\lambda \to 1/\lambda$. Then, the assumption that $\tau(\lambda)$ is entire in $\lambda$ implies by the equation (105) that $\tau(\lambda)$ is a polynomial in $\Lambda$ of the form (56) and that it satisfies the equations

$$\det_p D_\tau(\zeta_a^{(0)}) = 0, \quad \forall a \in \{1, ..., N\}, \tag{108}$$

thanks to (105) and (106), so that we obtain by SoV characterization $\tau(\lambda) \in \Sigma_{\mathcal{T}}$.

Let us now prove the reverse statement, i.e. we assume $\tau(\lambda) \in \Sigma_{\mathcal{T}}$ and we prove that there exists $Q(\lambda)$ of the form (103) with degree $N_Q = (p-1)N$ satisfying (106) and (105). Let us consider the system of equations

$$D_\tau(\lambda) \begin{pmatrix} X_0(\lambda) \\ X_1(\lambda) \\ \vdots \\ \vdots \\ X_{p-1}(\lambda) \end{pmatrix}_{p \times 1} = \left[\tau_\infty - (q^{-N_Q} A_\infty + q^{N_Q} A_0)\right]F(\lambda) \begin{pmatrix} \Lambda_0^2 - X^2 \\ \Lambda_1^2 - X^2 \\ \vdots \\ \vdots \\ \Lambda_{p-1}^2 - X^2 \end{pmatrix}_{p \times 1}, \tag{109}$$

where we have used the notations

$$\Lambda_i = q^{2i}\lambda^2 + \frac{1}{q^{2i}\lambda^2}. \tag{110}$$

From the condition $\tau(\lambda) \in \Sigma_{\mathcal{T}}$ and the assumption of general values of the boundary-bulk parameters (104), we know that $\det_p D_\tau(\lambda)$ is a non-zero polynomial, so defining

$$Z_{det_p D_\tau} = \left\{ \pm i^a q^{h+1/2}, \pm \zeta_n^{(h)} \ \forall a \in \{0,1\}, n \in \{1,...,2\mathsf{N}\}, h \in \{0,...,p-1\} \right\}, \tag{111}$$

we can solve the previous system of equations for any value of $\lambda \in \mathbb{C} \backslash Z_{det_p D_\tau}$ by the Cramer's rule,

$$X_i(\lambda) = \frac{\tau_\infty - (q^{-\mathsf{N}_Q} \mathsf{A}_\infty + q^{\mathsf{N}_Q} \mathsf{A}_0)}{(Z^2 - 4) \prod_{k=0}^{p-1} [\tau_\infty - (q^k \mathsf{A}_\infty + q^{-k} \mathsf{A}_0)]} \det_p D_\tau^{(i+1)}(\lambda), \tag{112}$$

where $D_\tau^{(i)}(\lambda)$ is the $p \times p$ matrix obtained replacing the column $i$ by the column at the r.h.s. of (109). Let us now rewrite the system of equation (109) bringing the first element in the last one for the two column vectors,

$$\tilde{D}_\tau(\lambda) \begin{pmatrix} X_1(\lambda) \\ X_2(\lambda) \\ \vdots \\ X_{p-1}(\lambda) \\ X_0(\lambda) \end{pmatrix}_{p \times 1} = \left[\tau_\infty - (q^{-\mathsf{N}_Q} \mathsf{A}_\infty + q^{\mathsf{N}_Q} \mathsf{A}_0)\right] F(\lambda) \begin{pmatrix} \Lambda_1^2 - X^2 \\ \Lambda_2^2 - X^2 \\ \vdots \\ \Lambda_{p-1}^2 - X^2 \\ \Lambda_0^2 - X^2 \end{pmatrix}_{p \times 1}, \tag{113}$$

where it is easy to see that $\tilde{D}_\tau(\lambda) = D_\tau(\lambda q)$. Rescaling now the argument of the functions, we can rewrite it as it follows,

$$D_\tau(\lambda) \begin{pmatrix} X_1(\lambda/q) \\ X_2(\lambda/q) \\ \vdots \\ X_{p-1}(\lambda/q) \\ X_0(\lambda/q) \end{pmatrix}_{p \times 1} = \left[\tau_\infty - (q^{-\mathsf{N}_Q} \mathsf{A}_\infty + q^{\mathsf{N}_Q} \mathsf{A}_0)\right] F(\lambda) \begin{pmatrix} \Lambda_0^2 - X^2 \\ \Lambda_1^2 - X^2 \\ \vdots \\ \Lambda_{p-2}^2 - X^2 \\ \Lambda_{p-1}^2 - X^2 \end{pmatrix}_{p \times 1}, \tag{114}$$

so that it must hold

$$X_{i+1}(\lambda/q) = X_i(\lambda) \ \ \forall \lambda \in \mathbb{C} \backslash Z_{det_p D_\tau}, i \in \{0,...,p-1\}, \tag{115}$$

where we have used the notation $X_p(\lambda) \equiv X_0(\lambda)$, or equivalently,

$$X_a(\lambda) = X_0(\lambda q^a) \ \ \forall \lambda \in \mathbb{C} \backslash Z_{det_p D_\tau}, a \in \{1,...,p-1\}. \tag{116}$$

Let us observe now that, from their definition, $X_a(\lambda)$ are continuous functions of $\lambda$ so the above equation must be indeed satisfied for any value of $\lambda \in \mathbb{C}$. Moreover, from the identity

$$\det_p D_\tau^{(1)}(\lambda) = \det_p D_\tau^{(1)}(1/\lambda), \tag{117}$$

which we can prove by some simple exchange of rows and columns, and from the fact that

$$\forall i \in \{0...p-1\}, \ \lambda \to 1/\lambda \ \Rightarrow \ \Lambda_i \to \Lambda_{p-i}, \tag{118}$$

we get the symmetry

$$X_0(\lambda) = X_0(1/\lambda), \tag{119}$$

which together with the symmetry $X_0(\lambda) = X_0(-\lambda)$ implies that $X_0(\lambda)$ is a function of $\Lambda$.

By using this last result we can rewrite the first equation of the system (109) as it follows $\forall \lambda \in \mathbb{C}$,

$$\tau(\lambda)X_0(\lambda) - \text{A}(\lambda)X_0(\lambda/q) - \text{A}(1/\lambda)X_0(\lambda q) = $$
$$\left[\tau_\infty - (q^{-\text{N}_Q}\text{A}_\infty + q^{\text{N}_Q}\text{A}_0)\right]\left(\Lambda^2 - X^2\right)F(\lambda). \quad (120)$$

Let us now prove that $\det_p D_\tau^{(1)}(\lambda)$ is indeed a polynomial of degree $(p-1)\text{N}+2p$ in $\Lambda$. Note that in the following when we refer to a row $k \in \mathbb{Z}$ what we mean is the row $k' \in \{1, ..., p\}$ with $k' = k \mod p$. In the row $\bar{h} = (p+1)/2 + h$ of $D_\tau^{(1)}(\pm i^a q^{1/2-h}\lambda)$ at least one of the three non-zero elements is diverging under the limit $\lambda \to \pm 1, \pm i$. We can proceed as done in the previous theorem, we define the matrix $\bar{D}_{\tau,h}^{(1)}(\lambda)$ as the matrix with all the rows coinciding with those of $D_\tau^{(1)}(\lambda)$ except the row $h+1$, which is obtained by summing the row $h$ and $h+1$ of $D_\tau^{(1)}(\lambda)$ and dividing them by $((i^a q^{h-1/2}\lambda)^2 - 1/(i^a q^{h-1/2}\lambda)^2)$, and the row $\bar{h}$, obtained multiplying the row $\bar{h}$ of $D_\tau^{(1)}(\lambda)$ by $((i^a q^{h-1/2}\lambda)^2 - 1/(i^a q^{h-1/2}\lambda)^2)$. Clearly we have

$$\det_p \bar{D}_{\tau,h}^{(1)}(\lambda) = \det_p D_\tau^{(1)}(\lambda), \quad (121)$$

and the interesting point is that now all the rows of the matrix $D_{\tau,h}^{(1)}(\pm i^a q^{1/2-h}\lambda)$ are finite in the limits $\lambda \to \pm 1, \pm i$. We have that the nonzero elements of the rows $h$, $h+1$ and $\bar{h}$ of $\bar{D}_{\tau,h}^{(1)}(\pm i^a q^{1/2-h})$ reads

$$\left[\bar{D}_{\tau,h}^{(1)}(\pm i^a q^{1/2-h})\right]_{h+1,h-1} = -s_{a,\pm}(1-\delta_{h-1,1}) + \delta_{h-1,1}\omega_a, \quad (122)$$

$$\left[\bar{D}_{\tau,h}^{(1)}(\pm i^a q^{1/2-h})\right]_{h+1,h} = -r_{a,\pm}(1-\delta_{h,1}) + \delta_{h,1}\omega_a, \quad (123)$$

$$\left[\bar{D}_{\tau,h}^{(1)}(\pm i^a q^{1/2-h})\right]_{h+1,h+1} = r_{a,\pm}(1-\delta_{h+1,1}) + \delta_{h+1,1}\omega_a, \quad (124)$$

$$\left[\bar{D}_{\tau,h}^{(1)}(\pm i^a q^{1/2-h})\right]_{h+1,h+2} = s_{a,\pm}(1-\delta_{h+2,1}) + \delta_{h+2,1}\omega_a, \quad (125)$$

$$\left[\bar{D}_{\tau,h}^{(1)}(\pm i^a q^{1/2-h})\right]_{\bar{h},\bar{h}-1} = -\text{x}(\pm i^a)(-1)^a(1-\delta_{\bar{h}-1,1}), \quad (126)$$

$$\left[\bar{D}_{\tau,h}^{(1)}(\pm i^a q^{1/2-h})\right]_{\bar{h},\bar{h}+1} = \text{x}(\pm i^a)(-1)^a(1-\delta_{\bar{h}+1,1}), \quad (127)$$

$$\left[\bar{D}_{\tau,h}^{(1)}(\pm i^a q^{1/2-h})\right]_{h,h-1} = \tau(\pm i^a q^{1/2})(1-\delta_{h+1,1}), \quad (128)$$

$$\left[\bar{D}_{\tau,h}^{(1)}(\pm i^a q^{1/2-h})\right]_{h,h+1} = -\tau(\pm i^a q^{1/2})(1-\delta_{h+1,1}), \quad (129)$$

where we have defined

$$\omega_a = (-1)^a\left(q^2 - 1/q^2\right). \quad (130)$$

The remaining rows of $\bar{D}_{\tau,h}^{(1)}(\pm i^a q^{1/2-h})$ produce the tridiagonal part of this matrix. It is possible to prove than that for any $h \in \{0, ..., p-1\}$ the matrix $\bar{D}_{\tau,h}^{(1)}(\pm i^a q^{1/2-h})$ has linear dependent rows; so that $\det_p D_\tau^{(1)}(\pm i^a q^{1/2-h}) = 0$ and the following factorization holds,

$$\det_p D_\tau^{(1)}(\lambda) = \left(\lambda^{2p} - \frac{1}{\lambda^{2p}}\right)P_\tau(\lambda). \quad (131)$$

Here $P_\tau(\lambda)$ is a Laurent polynomial of degree $2(p-1)\text{N} + 2p$ in $\lambda$, with the following odd parity:

$$P_\tau(1/\lambda) = -P_\tau(\lambda), \quad (132)$$

being $\det_p D_\tau^{(1)}(\lambda)$ a polynomial of degree $(p-1)N + 2p$ in $\Lambda$. Here, we want to prove that in fact

$$\det_p D_\tau^{(1)}(\lambda) = \left(\lambda^{2p} - \frac{1}{\lambda^{2p}}\right)^2 \bar{Q}_\tau(\lambda), \tag{133}$$

where $\bar{Q}_\tau(\lambda)$ is a polynomial of degree $(p-1)N$ in $\Lambda$. In order to do so we write down the equation

$$\tau(\lambda)R_\tau(\lambda) = \mathrm{A}(\lambda)R_\tau(\lambda/q) + \mathrm{A}(1/\lambda)R_\tau(\lambda q)$$
$$+ \left(Z^2 - 4\right)\left(\Lambda^2 - X^2\right)\prod_{k=0}^{p-1}\left[\tau_\infty - (q^k \mathrm{A}_\infty + q^{-k}\mathrm{A}_0)\right]F(\lambda), \tag{134}$$

where for convenience we have denoted $R_\tau(\lambda) = \det_p D_\tau^{(1)}(\lambda)$, and we recall $Z = \lambda^{2p} + \frac{1}{\lambda^{2p}}$. The above equation is a direct consequence of the equation satisfied by $X_0(\lambda)$ and of the definition of this last function in terms of $\det_p D_\tau^{(1)}(\lambda)$. Now let us consider the following limit on the above equation $\lambda \to \pm i^a$ with $a \in \{0,1\}$:

$$\tau(\pm i^a)R_\tau(\pm i^a) = \frac{1}{\pm 2i^a}\frac{d\mathrm{X}}{d\lambda}(\pm i^a)(R_\tau(\pm i^a/q) - R_\tau(\pm i^a q))$$
$$+ \mathrm{X}(\pm i^a)\lim_{\lambda \to \pm i^a}\left[\frac{R_\tau(\lambda/q)}{\lambda^2 - 1/\lambda^2} - \frac{R_\tau(\lambda q)}{\lambda^2 - 1/\lambda^2}\right], \tag{135}$$

now by using the known identities,

$$R_\tau(\pm i^a) = R_\tau(\pm i^a/q) = R_\tau(\pm i^a q) = 0, \tag{136}$$
$$\frac{R_\tau(\lambda/q)}{\lambda^2 - 1/\lambda^2} = \frac{R_\tau(q/\lambda)}{\lambda^2 - 1/\lambda^2}, \tag{137}$$

we get

$$\lim_{\lambda \to \pm i^a}\frac{R_\tau(\lambda/q)}{\lambda^2 - 1/\lambda^2} = -\lim_{\lambda \to \pm i^a}\frac{R_\tau(\lambda q)}{\lambda^2 - 1/\lambda^2}, \tag{138}$$

and so being $\mathrm{X}(\pm i^a) \neq 0$

$$\lim_{\lambda \to \pm i^a}\frac{R_\tau(\lambda/q)}{\lambda^2 - 1/\lambda^2} = 0. \tag{139}$$

These results imply the identities

$$P_\tau(\pm i^a/q) = -P_\tau(\pm i^a q) = 0. \tag{140}$$

We can now write the functional equation for $P_\tau(\lambda)$,

$$\tau(\lambda)P_\tau(\lambda) = \mathrm{A}(\lambda)P_\tau(\lambda/q) + \mathrm{A}(1/\lambda)P_\tau(\lambda q)$$
$$+ \left(\lambda^{2p} - \frac{1}{\lambda^{2p}}\right)\left(\Lambda^2 - X^2\right)\prod_{k=0}^{p-1}\left[\tau_\infty - (q^k \mathrm{A}_\infty + q^{-k}\mathrm{A}_0)\right]F(\lambda). \tag{141}$$

Taking the limit $\lambda \to \pm i^a$ with $a \in \{0,1\}$, we obtain

$$\tau(\pm i^a)P_\tau(\pm i^a) = \frac{1}{\pm 2i^a}\frac{d\mathrm{X}}{d\lambda}(\pm i^a)(P_\tau(\pm i^a/q) - P_\tau(\pm i^a q))$$
$$+ \mathrm{X}(\pm i^a)\lim_{\lambda \to \pm i^a}\left[\frac{P_\tau(\lambda/q)}{\lambda^2 - 1/\lambda^2} - \frac{P_\tau(\lambda q)}{\lambda^2 - 1/\lambda^2}\right], \tag{142}$$

so that using the previous result (140) and the identity

$$\lim_{\lambda \to \pm i^a}\frac{P_\tau(\lambda/q)}{\lambda^2 - 1/\lambda^2} = \lim_{\lambda \to \pm i^a}\frac{P_\tau(\lambda q)}{\lambda^2 - 1/\lambda^2}, \tag{143}$$

we obtain

$$P_\tau(\pm i^a) = 0, \tag{144}$$

being $\tau(\pm i^a) \neq 0$. Let us now compute the functional equation for $P_\tau(\lambda)$ in the points $\lambda = \pm i^a q^\epsilon$ for $a \in \{0, 1\}$, $\epsilon \in \{-1, 1\}$, we obtain

$$\tau(\pm i^a q)P_\tau(\pm i^a q) = \mathrm{A}(\pm i^a q)P_\tau(\pm i^a) + \mathrm{A}(\pm i^a/q)P_\tau(\pm i^a q^2), \tag{145}$$
$$\tau(\pm i^a/q)P_\tau(\pm i^a/q) = \mathrm{A}(\pm i^a/q)P_\tau(\pm i^a/q^2) + \mathrm{A}(\pm i^a q)P_\tau(\pm i^a), \tag{146}$$

implying

$$P_\tau(\pm i^a/q^2) = -P_\tau(\pm i^a q^2) = 0, \tag{147}$$

being $\mathrm{A}(\pm i^a q^\epsilon) \neq 0$ for $a, \epsilon \in \{0, 1\}$. We can iterate these computations for $\lambda = \pm i^a q^{b\epsilon}$ for any $a \in \{0, 1\}$, $\epsilon \in \{-1, 1\}$ and $b \in \{2, ..., (p-3)/2\}$ obtaining that

$$P_\tau(\pm i^a/q^{2b}) = -P_\tau(\pm i^a q^{2b}) = 0, \text{ for any } b \in \{1, ..., (p-3)/2\}. \tag{148}$$

In the cases $\lambda = \pm i^a q^{\pm 1/2}$ as $\mathrm{A}(\pm i^a/q^{1/2}) = 0$ the functional equation for $P_\tau(\lambda)$ give us

$$\tau(\pm i^a q^{\pm 1/2})P_\tau(\pm i^a q^{\pm 1/2}) = \mathrm{A}(\pm i^a q^{1/2})P_\tau(\pm i^a q^{\mp 1/2}), \tag{149}$$

which being $P_\tau(\pm i^a q^{1/2}) = -P_\tau(\pm i^a q^{-1/2})$ and $\tau(\pm i^a q^{\pm 1/2}) = \mathrm{A}(\pm i^a q^{1/2}) \neq 0$ implies the identity

$$P_\tau(\pm i^a q^{1/2}) = -P_\tau(\pm i^a q^{-1/2}) = 0, \tag{150}$$

so that the factorization (133) is proven and we get that

$$X_0(\lambda) = \frac{\tau_\infty - (q^{-\mathsf{N}_Q}\mathrm{A}_\infty + q^{\mathsf{N}_Q}\mathrm{A}_0)}{\prod_{k=0}^{p-1}[\tau_\infty - (q^k \mathrm{A}_\infty + q^{-k}\mathrm{A}_0)]}\bar{Q}_\tau(\lambda), \tag{151}$$

is a polynomial of degree $\mathsf{N}_Q = (p-1)\mathsf{N}$ in $\Lambda$ which has the form (103). This follows by taking the asymptotic of its functional equation so that we can fix

$$Q(\lambda) \equiv X_0(\lambda), \tag{152}$$

hence giving a constructive proof of the existence of the polynomial $Q$-function solution of the equation (105). The fact that it is unique is shown observing that if $\hat{Q}(\lambda)$ is another polynomial solution then,

$$D_\tau(\lambda)\begin{pmatrix} Q(\lambda) - \hat{Q}(\lambda) \\ Q(\lambda q) - \hat{Q}(\lambda q) \\ \vdots \\ \vdots \\ Q(\lambda q^{p-1}) - \hat{Q}(\lambda q^{p-1}) \end{pmatrix}_{p \times 1} = \begin{pmatrix} 0 \\ 0 \\ \vdots \\ \vdots \\ 0 \end{pmatrix}_{p \times 1}, \tag{153}$$

from which it follows $Q(\lambda) \equiv \hat{Q}(\lambda)$ as $D_\tau(\lambda)$ is invertible for any $\lambda \in \mathbb{C} \backslash Z_{det_p D_\tau}$.

Finally, let us show that $Q(\lambda)$ satisfies the condition (106). By the definition (112), $Q(\lambda)$ is a continuous function of the boundary-bulk parameters, then it is enough to prove this statement for some value of these parameters to show that it holds for almost all the values of these parameters.

Let us impose the condition (322), where the ratio $\beta/\alpha$ is fixed by (51), then the following identities are satisfied:

$$\mathrm{A}(\zeta_a^{(0)}) = 0 \quad \forall a \in \{1, ..., \mathsf{N}\}, \tag{154}$$

and the SoV characterization of the transfer matrix spectrum holds for any value of the boundary-bulk parameters satisfying the inequalities (323)-(324). So in particular if we impose

$$\mu_{n_k,-} = 1/(q^{1+k}\mu_{a,+}) \quad \forall k \in \{1, ..., p-1\}, \tag{155}$$

for some $n_k \in \{1, ..., \mathsf{N}\}\backslash\{a\}$ once we have chosen any $a \in \{1, ..., \mathsf{N}\}$. Under these conditions it holds

$$\mathrm{A}(\zeta_a^{(k)}) = 0, \ \forall k \in \{1, ..., p-1\}, \tag{156}$$

and the SoV representation implies the following centrality condition

$$\prod_{k=0}^{p-1} \mathscr{T}(\zeta_a^{(k)}) = \prod_{k=0}^{p-1} \mathrm{A}(1/\zeta_a^{(k)}), \tag{157}$$

from which in particular follows

$$\prod_{k=0}^{p-1} \tau(\zeta_a^{(k)}) = \prod_{k=0}^{p-1} \mathrm{A}(1/\zeta_a^{(k)}). \tag{158}$$

Let us remark now that the r.h.s and the l.h.s of the above equation are continuous w.r.t. the boundary-bulk parameters so that the above identity holds also if we take the special limit $\mu_{a,-} \to q^{1-p}/\mu_{a,+}$ for which it holds $\mathrm{A}(1/\zeta_a^{(p-1)}) = 0$ and so we get

$$\exists! \bar{h} \in \{0, ..., p-1\} : \tau(\zeta_a^{(\bar{h})}) = 0. \tag{159}$$

By definition of the function $Q(\lambda)$ under these conditions and limit on the bulk parameters we get

$$Q(\zeta_a^{(\bar{h})}) \propto \det_p \begin{pmatrix} W_{a,\bar{h}} & -\mathrm{A}(1/\zeta_a^{(\bar{h})}) & 0 & \cdots & 0 & 0 \\ W_{a,\bar{h}+1} & \tau(\zeta_a^{(\bar{h}+1)}) & -\mathrm{A}(1/\zeta_a^{(\bar{h}+1)}) & 0 & \cdots & 0 \\ W_{a,\bar{h}+2} & 0 & \tau(\zeta_a^{(\bar{h}+2)}) & -\mathrm{A}(1/\zeta_a^{(\bar{h}+1)}) & & \vdots \\ \vdots & & \cdots & & & \vdots \\ W_{a,p-1} & 0 & \cdots 0 & \tau(\zeta_a^{(p-1)}) & 0\cdots & \vdots \\ \vdots & & & & \ddots & 0 \\ W_{a,\bar{h}} & \cdots & 0 & 0 & \tau(\zeta_a^{(\bar{h}-2)}) & -\mathrm{A}(1/\zeta_a^{(\bar{h}-2)}) \\ W_{a,\bar{h}} & 0 & \cdots & 0 & 0 & \tau(\zeta_a^{(\bar{h}-1)}) \end{pmatrix}, \tag{160}$$

where we have defined

$$W_{a,k} = \left((\zeta_a^{(k)})^2 + 1/(\zeta_a^{(k)})^2\right)^2 - X^2. \tag{161}$$

Now replacing the first row $R_1$ with the following linear combination of rows,

$$\bar{R}_1 = R_1 + \sum_{i=0}^{p-2-\bar{h}} \prod_{j=0}^{i} \frac{\mathrm{A}(1/\zeta_a^{(\bar{h}+j)})}{\tau(\zeta_a^{(\bar{h}+j+1)})} R_{2+i}, \tag{162}$$

we get

$$\bar{R}_1 = \begin{pmatrix} \bar{W}_{a,\bar{h}} & 0 & \cdots & 0 & 0 \end{pmatrix}_{1\times p}, \tag{163}$$

where

$$\bar{W}_{a,\bar{h}} = W_{a,\bar{h}} + \sum_{i=0}^{p-2-\bar{h}} \prod_{j=0}^{i} \frac{\mathrm{A}(1/\zeta_a^{(\bar{h}+j)})}{\tau(\zeta_a^{(\bar{h}+j+1)})} W_{a,\bar{h}+1+i}, \tag{164}$$

and so

$$Q(\zeta_a^{(\bar{h})}) = \bar{W}_{a,\bar{h}} \prod_{k \neq \bar{h}, k=0}^{p-1} \tau(\zeta_a^{(k)}) \neq 0, \tag{165}$$

for generic values of the boundary-bulk parameters. Indeed, as the $W_{a,\bar{h}+1+i}$ are functions only of the bulk parameter $\mu_{a,+}$ while the ratios $\mathrm{A}(1/\zeta_a^{(\bar{h}+j)})/\tau(\zeta_a^{(\bar{h}+j+1)})$ are functions of both the boundary and the bulk parameters then we can prove that $\bar{W}_{a,\bar{h}} \neq 0$. Explicitly we can compute the asymptotic of $\bar{W}_{a,\bar{h}}$ in the limit $\mu_{a,+} \to \infty$, by using the know asymptotic of the transfer matrix, therefore showing that it is non-zero for general values of boundary-bulk parameters. $\qquad \square$

In the previous theorem we have excluded the boundary-bulk one-constraint cases leading to an identically zero $\det D_\tau(\lambda)$ for any $\tau(\lambda) \in \Sigma_{\mathscr{T}}$, these specific cases are considered in the next theorem.

**Theorem 5.2.** *Let us assume that there exists $k \in \{0, ..., p-1\}$ such that it holds*

$$\tau_\infty = q^{-k}\mathrm{A}_\infty + q^k\mathrm{A}_0, \tag{166}$$

*then, for almost all the values of the boundary-bulk parameters, $\tau(\lambda) \in \Sigma_{\mathscr{T}}$ (the set of the eigenvalues of $\mathscr{T}(\lambda)$) if and only if $\tau(\lambda)$ is an entire function and there exists and is unique a polynomial $Q(\lambda)$ of the form (103) with $\mathrm{N}_Q \leq (p-1)(\mathrm{N}+1)$ and $\mathrm{N}_Q = k \bmod p$, satisfying the following homogeneous Baxter equation,*

$$\tau(\lambda)Q(\lambda) = \mathrm{A}(\lambda)Q(\lambda/q) + \mathrm{A}(1/\lambda)Q(\lambda q), \tag{167}$$

*and the conditions*

$$(Q(\zeta_a^{(0)}), ..., Q(\zeta_a^{(p-1)})) \neq (0, ..., 0) \quad \forall a \in \{1, ..., \mathrm{N}\}. \tag{168}$$

*Proof.* First let us assume that $\tau(\lambda)$ and $Q(\lambda)$ satisfies the homogeneous Baxter equation with $\tau(\lambda)$ entire function and $Q(\lambda)$ polynomial of the form (103) with $\mathrm{N}_Q \leq (p-1)(\mathrm{N}+1)$ and $\mathrm{N}_Q = k \bmod p$, then from this same equation it follows that $\tau(\lambda)$ is a polynomial of the form (56). Moreover, for any fixed $\lambda \in \mathbb{C}$ we can construct the following homogeneous system of equations:

$$D_\tau(\lambda) \begin{pmatrix} Q(\lambda) \\ Q(\lambda q) \\ \vdots \\ \vdots \\ Q(\lambda q^{p-1}) \end{pmatrix}_{p \times 1} = \begin{pmatrix} 0 \\ 0 \\ \vdots \\ \vdots \\ 0 \end{pmatrix}_{p \times 1}, \tag{169}$$

which is satisfied as a consequence of the Baxter equation. Finally, being $(Q(\lambda), ..., Q(\lambda q^{p-1}))$ non-zero for any $\lambda \in \mathbb{C}$, up to at most a finite number of values, we get

$$\det D_\tau(\lambda) = 0 \quad \forall \lambda \in \mathbb{C}, \tag{170}$$

so that Proposition 5.1 implies $\tau(\lambda) \in \Sigma_{\mathscr{T}}$.

To prove the reverse statement we use the results of the Lemma C.1 on the matrix $D_\tau(\lambda)$ and on its cofactors

$$\mathrm{C}_{i,j}(\lambda) = (-1)^{i+j}\det_{p-1} D_{\tau,i,j}(\lambda). \tag{171}$$

We take now $\tau(\lambda) \in \Sigma_{\mathscr{T}}$ from which it holds

$$\det D_\tau(\lambda) = 0 \quad \forall \lambda \in \mathbb{C}, \tag{172}$$

and so by Lemma C.1 it follows that $\text{rank} D_\tau(\lambda) = p - 1$ for any $\lambda \in \mathbb{C} \backslash K$, where $K$ is a finite set of complex numbers if not empty. Then the matrix composed of the cofactors of the matrix $D_\tau(\lambda)$ has rank 1 for any $\lambda \in \mathbb{C} \backslash K$. This just means the proportionality

$$V_i(\lambda) = A_{i,j}(\lambda) V_j(\lambda) \ \forall \lambda \in \mathbb{C} \backslash K, \forall i, j \in \{1, ..., p\}, \tag{173}$$

where we have defined

$$V_i(\lambda) \equiv (C_{i,1}(\lambda), C_{i,2}(\lambda), ..., C_{i,p}(\lambda)) \ \forall \lambda \in \mathbb{C} \backslash K, \forall i \in \{1, ..., p\}, \tag{174}$$

and $A_{i,j}(\lambda)$ are some functions such that

$$A_{i,j}(\lambda) \neq 0 \text{ and finite for any } \lambda \in \mathbb{C} \backslash \left\{ K \cup K_0 \cup K_i \cup K_j \right\}, \tag{175}$$

where $K_0$ is the set of the $p$-roots of unit and

$$K_a \equiv \{x \in \mathbb{C} : V_a(x) \equiv (0, ..., 0)\} \ \forall a \in \{1, ..., p\}, \tag{176}$$

such sets are finite if not empty, being the elements of the vectors $(\Lambda^p - X^p) V_i(\lambda)$ Laurent polynomials. The above identities in particular imply

$$A_{1,2}(\lambda) C_{1,1}(\lambda) C_{2,2}(\lambda) = A_{1,2}(\lambda) C_{1,2}(\lambda) C_{2,1}(\lambda) \ \forall \lambda \in \mathbb{C} \backslash K, \tag{177}$$

so that for any $\lambda \in \mathbb{C} \backslash \left\{ K \cup K_0 \cup K_i \cup K_j \right\}$ it holds

$$C_{1,1}(\lambda) C_{2,2}(\lambda) = C_{1,2}(\lambda) C_{2,1}(\lambda). \tag{178}$$

Hence it holds for any $\lambda \in \mathbb{C}$ using continutiy properties of the cofactors, being $\left\{ K \cup K_0 \cup K_i \cup K_j \right\}$ a finite set of values. Similarly, the fact that the vectorial condition $D(\lambda) V_1(\lambda) = \underline{0}$ holds true for any $\lambda \in \mathbb{C} \backslash K$ implies that it is indeed satisfied for any $\lambda \in \mathbb{C}$. Here, we write explicitly the first element of this vectorial condition,

$$\tau(\lambda) C_{1,1}(\lambda) = A(\lambda) C_{1,p}(\lambda) + A(1/\lambda) C_{1,2}(\lambda), \tag{179}$$

together with the rewriting of (178) by using the identity (355)

$$C_{1,1}(\lambda) C_{1,1}(\lambda q) = C_{1,2}(\lambda) C_{1,p}(q\lambda). \tag{180}$$

Once we recall that $C_{1,1}(\lambda)$, $C_{1,2}(\lambda)$ and $C_{1,p}(\lambda)$ are Laurent polynomial in $\lambda$ satisfying the factorizations (358), (359) and (360), respectively, it follows that the above two equations holds as well as if written in terms of the functions $\widehat{C}_{1,1}(\lambda)$, $\widehat{C}_{1,2}(\lambda)$ and $\widehat{C}_{1,p}(\lambda)$.

Similarly to what has been done in the Lemma 5 of the paper [128], we can show that the two above equations for $\widehat{C}_{1,1}(\lambda)$, $\widehat{C}_{1,2}(\lambda)$ and $\widehat{C}_{1,p}(\lambda)$ and their symmetry properties (357) imply that if $\widehat{C}_{1,1}(\lambda)$ has a common zero with $\widehat{C}_{1,2}(\lambda)$ then this is also a zero of $\widehat{C}_{1p}(\lambda)$ and also the inverse of such zero is a common zero of these polynomials. Moreover, the same statement holds exchanging $\widehat{C}_{1,2}(\lambda)$ with $\widehat{C}_{1,p}(\lambda)$. So we can denote with $c_{1,1}\overline{C}_{1,1}(\lambda)$, $c_{1,2}\overline{C}_{1,2}(\lambda)$ and $c_{1,p}\overline{C}_{1,p}(\lambda)$ the polynomials obtained simplifying the common factors in $\widehat{C}_{1,1}(\lambda)$, $\widehat{C}_{1,2}(\lambda)$ and $\widehat{C}_{1,p}(\lambda)$. Then, they have to satisfy the relations

$$\overline{C}_{1,p}(\lambda) = y_{1,1} \overline{C}_{1,1}(\lambda q^{-1}), \quad \overline{C}_{1,2}(\lambda) = y_{1,1}^{-1} \overline{C}_{1,1}(\lambda q) \tag{181}$$

and defined $x_{1,1} \equiv c_{1,1}/c_{1,2} = c_{1,p}/c_{1,1}$, we obtain the following Baxter equation in the polynomial $\overline{C}_{1,1}(\lambda)$:

$$t(\lambda) \overline{C}_{1,1}(\lambda) = \left( x_{1,1} y_{1,1} \right) A(\lambda) \overline{C}_{1,1}(\lambda q^{-1}) + \left( 1/(x_{1,1} y_{1,1}) \right) A(1/\lambda) \overline{C}_{1,1}(\lambda q), \tag{182}$$

and computing the above equation in $\lambda = q^{1/2}$ we get

$$t(q^{1/2})\overline{C}_{1,1}(q^{1/2}) = \left(x_{1,1}y_{1,1}\right) A(q^{1/2})\overline{C}_{1,1}(q^{-1/2}), \tag{183}$$

from which it follows $x_{1,1}y_{1,1} = 1$ once we recall that $C_{1,1}(q^{1/2}) \neq 0$ and that $\overline{C}_{1,1}(\lambda)$ is even under $\lambda \to 1/\lambda$. So, we can define

$$Q(\lambda) \equiv \overline{C}_{1,1}(\lambda), \tag{184}$$

a polynomial in $\Lambda$ of maximal degree $(p-1)(N+1)$, which satisfies the homogeneous Baxter equation as required. $\qquad\square$

Let us introduce now the following states:

$$\langle \beta, \omega | = \sum_{h_1,\dots,h_N=0}^{p-1} \prod_{a=1}^{N}\prod_{k_a=0}^{h_a-1} \frac{A(1/\zeta_a^{(k_a)})}{D(1/\zeta_a^{(k_a)})} \prod_{1 \leq b < a \leq N} (X_a^{(h_a)} - X_b^{(h_b)}) \langle \beta, h_1,\dots,h_N|, \tag{185}$$

$$|\beta, \bar{\omega}\rangle = \sum_{h_1,\dots,h_N=0}^{p-1} \prod_{1 \leq b < a \leq N} (X_a^{(h_a)} - X_b^{(h_b)})|\beta, h_1,\dots,h_N\rangle, \tag{186}$$

(see (60), (67) and (74)) and the following renormalization of the $\mathcal{B}_-$-operator family

$$\hat{\mathcal{B}}_-(\lambda|\beta) = \frac{\mathcal{B}_-(\lambda|\beta)T_\beta^2}{(\lambda^2/q - q/\lambda^2)B_-(\beta)}, \tag{187}$$

which is a degree $N$ polynomial in $\Lambda = \lambda^2 + \frac{1}{\lambda^2}$, and where $T_\beta$ is simply a shift on the gauge parameter $\beta$ (see (353)). As first remarked in the papers [73, 121], from the polynomial characterization of the $Q$-function and the SoV characterization it follows the Bethe-like rewriting of the transfer matrix eigenstates stated in the following[1]:

**Corollary 5.1.** *The left and right transfer matrix eigenstates associated to $\tau(\lambda) \in \Sigma_{\mathcal{T}}$ admit the following Bethe ansatz like representations,*

$$\langle \tau | = \langle \beta, \omega | \prod_{b=1}^{N_Q} \hat{\mathcal{B}}_-(\lambda_b|\beta), \quad |\tau\rangle = \prod_{b=1}^{N_Q} \hat{\mathcal{B}}_-(\lambda_b|\beta)|\beta, \bar{\omega}\rangle, \tag{188}$$

*where the $\lambda_b$ (fixed up the symmetry $\lambda_b \to -\lambda_b$, $\lambda_b \to 1/\lambda_b$) for $b \in \{1,\dots,N_Q\}$ are the zeros of $Q(\lambda)$ and we have imposed the condition (51) on the gauge parameters.*

*Proof.* These identities follow from the polynomiality of the $Q$-functions, which implies the following identity:

$$\prod_{a=1}^{N} q_{\tau,a}^{(h_a)} = \prod_{a=1}^{N} Q(\zeta_a^{(h)}) = \prod_{a=1}^{N}\prod_{b=1}^{N_Q}((\zeta_a^{(h)})^2 + 1/(\zeta_a^{(h)})^2 - \Lambda_b)$$

$$= (-1)^{N\,N_Q} \prod_{a=1}^{N}\prod_{b=1}^{N_Q}(\Lambda_b - ((\zeta_a^{(h)})^2 + 1/(\zeta_a^{(h)})^2)) = (-1)^{N\,N_Q} \prod_{b=1}^{N_Q} \hat{B}_{\mathbf{h}}(\lambda_b), \tag{189}$$

---

[1]One should remark that the logic that lead us to the ABA rewriting of the transfer matrix eigenstates is completely different from the one underling the algebraic Bethe ansatz. We get it by rewriting the original SoV form and this allows us to identity the non-trivial state that takes a role similar to a reference state. Note however that it has properties rather different from an ABA reference state as in general it is not an eigenstate of the transfer matrix! For simpler models, for which such a reference state can be naturally guessed, one can also follow the ABA logic i.e. to make an ansatz on the form of the ABA states and then to compute the action of the transfer matrix on these states deriving the Bethe equations by putting to zero the so-called unwanted terms. This is what it has been done in the paper [64] for the quantum spin 1/2 chains.

where the $\hat{B}_{\mathbf{h}}(\lambda_b)$ is the eigenvalue of the operator $\hat{\mathscr{B}}_-(\lambda|\beta)$ and the

$$\Lambda_b = \lambda_b^2 + 1/\lambda_b^2 \tag{190}$$

are the zeros of the $Q$-function as defined in (103). Now we have just to do the action of the monomial,

$$\prod_{b=1}^{\mathsf{N}_Q} \hat{\mathscr{B}}_-(\lambda_b|\beta) \tag{191}$$

on the right state (186) and use that by definition,

$$\prod_{b=1}^{\mathsf{N}_Q} \hat{\mathscr{B}}_-(\lambda_b|\beta)|\beta, h_1, ..., h_{\mathsf{N}}\rangle = |\beta, h_1, ..., h_{\mathsf{N}}\rangle \prod_{b=1}^{\mathsf{N}_Q} \hat{B}_{\mathbf{h}}(\lambda_b) \tag{192}$$

to prove that the vector in (188) coincides, up to the sign, with the vector (63) and so it is the corresponding transfer matrix eigenvector; similarly one shows that the covector in (188) coincides with the covector (65).

$\square$

# 6 Conclusions

In this second article we have shown how to implement the SoV method to characterize the transfer matrix spectrum for integrable models associated to the Bazhanov-Stroganov quantum Lax operator and to the most general integrable boundary conditions. For that purpose it was necessary to perform a gauge transformation so as to recast the problem in a form similar to the one studied in our first article, i.e., such that one of the boundary $K$-matrices becomes triangular after the gauge transformation. Let us stress that the separate basis was designed again as the (pseudo)-eigenvector basis of some gauged operator of the reflection algebra having simple spectrum. What remains to be done is the construction of integrable local cyclic Hamiltonian having appropriate boundary conditions and commuting with the boundary transfer matrices considered here. This amounts to use trace identities involving the fundamental $R$-matrix acting in the tensor product of two cyclic representations [143, 146, 147] and to construct the associated $K$-matrices, hence also acting in these cyclic representations. The reflection equations will have to be written for arbitrary choices (and mixing) of the spin-1/2 and cyclic representations. Correspondingly, there will be compatibility conditions between the different $K$-matrices acting in these two different representations. We will address this question in a forthcoming article [165].

# Acknowledgements

J. M. M. and G. N. are supported by CNRS and ENS de Lyon; B. P. is supported by ENS de Lyon and ENS Cachan.

# Appendices

## A Gauge transformed Yang-Baxter algebra

### A.1 Gauge transformed Yang-Baxter generators

For arbitrary complex parameters $\alpha$ and $\beta$ let us introduce the following two matrices:

$$G(\lambda|\alpha,\beta) \equiv \begin{pmatrix} 1/(\alpha\beta\lambda) & \beta/(\alpha\lambda) \\ 1 & 1 \end{pmatrix}, \quad \bar{G}(\lambda|\gamma) \equiv \begin{pmatrix} 1/(\lambda\gamma) & 0 \\ 1 & 1 \end{pmatrix}, \tag{193}$$

and their inverses,

$$G^{-1}(\lambda|\alpha,\beta) = \frac{\alpha\lambda}{\beta - 1/\beta} \begin{pmatrix} -1 & \beta/(\alpha\lambda) \\ 1 & -1/(\alpha\beta\lambda) \end{pmatrix}, \quad \bar{G}^{-1}(\lambda|\gamma) = \begin{pmatrix} \lambda\gamma & 0 \\ -\lambda\gamma & 1 \end{pmatrix}. \tag{194}$$

Now we can construct the gauge transformed bulk monodromy matrix (see (4)),

$$M(\lambda|\alpha,\beta,\gamma) = G^{-1}(\lambda q^{1/2}|\alpha,\beta) M(\lambda)\bar{G}(\lambda q^{1/2}|\gamma) = \begin{pmatrix} A(\lambda|\alpha,\beta,\gamma) & B(\lambda|\alpha,\beta) \\ C(\lambda|\alpha,\beta,\gamma) & D(\lambda|\alpha,\beta) \end{pmatrix}, \tag{195}$$

and, in a similar way, we can define (see (3))

$$\hat{M}(\lambda|\alpha,\beta,\gamma) = \bar{G}^{-1}(q^{1/2}/\lambda|\gamma) \hat{M}(\lambda)G(q^{1/2}/\lambda|\alpha,\beta) = \begin{pmatrix} \bar{A}(\lambda|\alpha,\beta,\gamma) & \bar{B}(\lambda|\alpha,\beta,\gamma) \\ \bar{C}(\lambda|\alpha,\beta,\gamma) & \bar{D}(\lambda|\alpha,\beta,\gamma) \end{pmatrix}. \tag{196}$$

The definition here chosen of these gauge transformations differ w.r.t. that used previously in the literature on one hand for the particular choice of the right transformation in $M(\lambda|\alpha,\beta,\gamma)$ and, on the other hand, as the parameters on the left and the right transformation are a priori independent. It is simple to prove by direct computations that

$$\bar{D}(\lambda|\alpha,\beta,\gamma) = f(\alpha,\beta,\gamma)A(1/\lambda|\alpha,\beta,\gamma), \quad \bar{B}(\lambda|\alpha,\beta,\gamma) = -f(\alpha,\beta,\gamma)B(1/\lambda|\alpha,\beta), \tag{197}$$

$$\bar{C}(\lambda|\alpha,\beta,\gamma) = -f(\alpha,\beta,\gamma)C(1/\lambda|\alpha,\beta,\gamma), \quad \bar{A}(\lambda|\alpha,\beta,\gamma) = f(\alpha,\beta,\gamma)D(1/\lambda|\alpha,\beta), \tag{198}$$

where

$$f(\alpha,\beta,\gamma) = (-1)^{\mathsf{N}} \frac{\gamma(1-\beta^2)}{\alpha\beta}. \tag{199}$$

Moreover, the identity

$$\det_q M(\lambda) = (-1)^{\mathsf{N}} M(\lambda q^{1/2})\hat{M}(q^{1/2}/\lambda), \tag{200}$$

and the corollaries

$$\begin{aligned} \det_q M(\lambda) &= (-1)^{\mathsf{N}} M(\lambda q^{1/2}|\alpha,\beta,\gamma)\hat{M}(q^{1/2}/\lambda|\alpha q,\beta,\gamma q) \\ &= (-1)^{\mathsf{N}} \hat{M}(q^{1/2}/\lambda|\alpha q,\beta,\gamma q)M(\lambda q^{1/2}|\alpha,\beta,\gamma), \end{aligned} \tag{201}$$

imply the following two equivalent expressions of the quantum determinant by the gauge transformed generators,

$$\det_q M(\lambda) = \frac{\gamma(1-q^2\beta^2)\left[A(\lambda q^{1/2}|\alpha,\beta,\gamma)D(\lambda/q^{1/2}|\alpha,\beta q) - B(\lambda q^{1/2}|\alpha,\beta)C(\lambda/q^{1/2}|\alpha,\beta q,\gamma q)\right]}{\alpha\beta} \tag{202}$$

$$= \frac{\gamma(q^2 - \beta^2)\left[D(\lambda q^{1/2}|\alpha,\beta)A(\lambda/q^{1/2}|\alpha,\beta/q,\gamma q) - C(\lambda q^{1/2}|\alpha,\beta,\gamma)B(\lambda/q^{1/2}|\alpha,\beta/q)\right]}{\alpha\beta},$$
(203)

plus other two equivalent rewriting. The gauge transformed Yang-Baxter generators are of special interest as they define a closed set of commutation relations,

$$A(\lambda|\alpha,\beta,\gamma)A(\mu|\alpha,\beta/q,\gamma/q) = A(\mu|\alpha,\beta,\gamma)A(\lambda|\alpha,\beta/q,\gamma/q),$$
(204)

$$B(\lambda|\alpha,\beta)B(\mu|\alpha,\beta/q) = B(\mu|\alpha,\beta)B(\lambda|\alpha,\beta/q),$$
(205)

$$C(\lambda|\alpha,\beta,\gamma)C(\mu|\alpha,\beta q,\gamma/q) = C(\mu|\alpha,\beta,\gamma)C(\lambda|\alpha,\beta q,\gamma/q),$$
(206)

$$D(\lambda|\alpha,\beta)D(\mu|\alpha,\beta/q) = D(\mu|\alpha,\beta)D(\lambda|\alpha,\beta/q),$$
(207)

$$A(\lambda|\alpha,\beta,\gamma)B(\mu|\alpha,\beta/q) = \frac{q(\lambda/\mu - \mu/\lambda)}{\lambda q/\mu - \mu/q\lambda}B(\mu|\alpha,\beta)A(\lambda|\alpha,\beta/q,\gamma q)$$
$$+ \frac{(q-1/q)\mu/\lambda}{\lambda q/\mu - \mu/q\lambda}A(\mu|\alpha,\beta,\gamma)B(\lambda|\alpha,\beta/q),$$
(208)

$$B(\lambda|\alpha,\beta)A(\mu|\alpha,\beta/q,\gamma q) = \frac{\lambda/\mu - \mu/\lambda}{q(\lambda q/\mu - \mu/q\lambda)}A(\mu|\alpha,\beta,\gamma)B(\lambda|\alpha,\beta/q)$$
$$+ \frac{(q-1/q)\lambda/\mu}{\lambda q/\mu - \mu/q\lambda}B(\mu|\alpha,\beta)A(\lambda|\alpha,\beta/q,\gamma q).$$
(209)

We can prove these commutation relations by direct computations using the properties of the gauge transformations and their action on the Yang-Baxter equation.

## A.2 Pseudo-reference state for the gauge transformed Yang-Baxter algebra

In the following, we want to study the conditions for which a nonzero state identically annihilated by the action of the operator family $A(\lambda|\alpha,\beta,\gamma)$ exists,

$$\langle\Omega,\alpha,\beta,\gamma|A(\lambda|\alpha,\beta,\gamma) = 0.$$
(210)

It is an easy consequence of the gauge transformed Yang-Baxter commutation relations that under the condition that this state exists and is unique then it is a pseudo-reference state for the gauge transformed Yang-Baxter algebra, i.e. it holds

$$\langle\Omega,\alpha,\beta,\gamma|A(\lambda|\alpha,\beta,\gamma) = 0, \quad \langle\Omega,\alpha,\beta,\gamma|B(\lambda|\alpha,\beta) = b(\lambda|\alpha,\beta)\langle\Omega,\alpha,\beta/q,\gamma q|, \quad (211)$$

$$\langle\Omega,\alpha,\beta/q,\gamma q|C(\lambda|\alpha,\beta q,\gamma q) = c(\lambda|\alpha,\beta q)\langle\Omega,\alpha,\beta,\gamma|, \quad \langle\Omega,\alpha,\beta,\gamma|D(\lambda|\alpha,\beta) \neq 0, \quad (212)$$

with

$$b(\lambda q^{1/2}|\alpha,\beta)c(\lambda q^{-1/2}|\alpha,\beta q) = -\det{}_q M(\lambda).$$
(213)

Here, we show that we can construct such a pseudo-reference state if and only if we impose at least $N + 1$ constraints on the bulk and gauge parameters.

Let us start our analysis looking to the local conditions to be imposed, in order to do so let us define the local gauge transformed bulk operators,

$$\begin{pmatrix} A_n(\lambda|\alpha,\beta,\gamma) & B_n(\lambda|\alpha,\beta) \\ C_n(\lambda|\alpha,\beta,\gamma) & D_n(\lambda|\alpha,\beta) \end{pmatrix}_0 = G_0^{-1}(\lambda q^{1/2}|\alpha,\beta)L_{0,n}(\lambda q^{-1/2})\bar{G}_0(\lambda q^{1/2}|\gamma),$$
(214)

and let us introduce the parameters

$$z_n^{(\epsilon_n,k_n)} = -q^{k_n+1/2}\left[\left(\frac{c_n^p + d_n^p}{a_n^p + b_n^p}\right)^{1/p}\frac{a_n}{c_n}\right]^{(1+\epsilon_n)/2}\frac{\alpha_n}{a_n},$$
(215)

$$w_n^{(\epsilon_n,k_n)} = -q^{1/2-k_n}\left[\left(\frac{a_n^p + b_n^p}{c_n^p + d_n^p}\right)^{1/p}\frac{d_n}{b_n}\right]^{(1+\epsilon_n)/2}\frac{\beta_n}{d_n},$$
(216)

where $n \in \{1, ..., \mathsf{N}\}$, $\epsilon_n = \pm 1$, $k_n \in \{0, ..., p-1\}$. Let us denote with

$$\langle h_n, n | v_n = q^{h_n} \langle h_n, n |, \quad v_n | h_n, n \rangle = | h_n, n \rangle q^{h_n}, \tag{217}$$

the left and right eigenbasis of the operators $v_n$.

**Lemma A.1.** *Let us assume that*

$$a_n^p + b_n^p \neq 0, \quad c_n^p + d_n^p \neq 0, \tag{218}$$

*then the non-zero left state annihilated by the local operator $A_n(\lambda | \alpha, \beta, \gamma)$ exists if and only if we impose the following two constraints on the gauge parameters:*

$$\beta / \alpha = w_n^{(\epsilon_n, k_n)}, \quad \gamma = z_n^{(\epsilon_n, k_n)}, \tag{219}$$

*for some fixed $\epsilon_n = \pm 1$ and $k_n \in \{0, ..., p-1\}$, moreover this state is uniquely defined by*

$$\langle \Omega_{n,\alpha,\beta,\gamma} | = \sum_{h_n=0}^{p-1} q^{h_n(k_n+1)} \left[ \prod_{r_n=1}^{h_n} \frac{a_n q^{r_n - 1/2} + b_n q^{1/2 - r_n}}{c_n q^{r_n - 1/2} + d_n q^{1/2 - r_n}} \left( \frac{c_n^p + d_n^p}{a_n^p + b_n^p} \right)^{1/p} \right]^{(1+\epsilon_n)/2} \langle h_n, n |. \tag{220}$$

*Similarly, if the condition (218) holds the non-zero right state annihilated by the local operator $A_n(\lambda | \alpha, \beta, \gamma)$ exists if and only if we impose the following two constraints on the gauge parameters:*

$$\beta / \alpha = w_n^{(\epsilon_n, k_n)}, \quad \gamma = z_n^{(\epsilon_n, k_n - 2)}, \tag{221}$$

*for some fixed $\epsilon_n = \pm 1$ and $k_n \in \{0, ..., p-1\}$, moreover this state is uniquely defined by*

$$| \Omega_{n,\alpha,\beta,\gamma} \rangle = \sum_{h_n=0}^{p-1} | h_n, n \rangle q^{-h_n k_n} \left[ \prod_{r_n=1}^{h_n} \frac{c_n q^{r_n - 1/2} + d_n q^{1/2 - r_n}}{a_n q^{r_n - 1/2} + b_n q^{1/2 - r_n}} \left( \frac{a_n^p + b_n^p}{c_n^p + d_n^p} \right)^{1/p} \right]^{(1+\epsilon_n)/2}. \tag{222}$$

*These are pseudo-eigenstates of the operator $B_n(\lambda | \alpha, \beta)$,*

$$\langle \Omega_{n,\alpha,\beta,\gamma} | B_n(\lambda | \alpha, \beta) = b_n(\lambda | \alpha, \beta) \langle \Omega_{n,\alpha,\beta/q,\gamma q} |, \tag{223}$$

$$B_n(\lambda | \alpha, \beta) | \Omega_{n,\alpha,\beta,\gamma} \rangle = q \, b_n(\lambda q | \alpha, \beta) | \Omega_{n,\alpha,\beta/q,\gamma q} \rangle, \tag{224}$$

*where*

$$b_n(\lambda | \alpha, \beta) = \frac{\beta^2 \gamma_n}{(1 - \beta^2) \mu_{n,\epsilon_n}} \left( \frac{\lambda}{q^{1/2} \mu_{n,\epsilon_n}} - \frac{q^{1/2} \mu_{n,\epsilon_n}}{\lambda} \right). \tag{225}$$

*Proof.* The lemma is proven by direct construction. Let us introduce a state

$$\langle \Omega_{n,\alpha,\beta,\gamma} | = \sum_{h=0}^{p-1} c_h(n, \alpha, \beta, \gamma) \langle h, n |, \tag{226}$$

and look for the conditions to be imposed on $c_h(n, \alpha, \beta, \gamma)$ in order to satisfy the equation

$$\langle \Omega_{n,\alpha,\beta,\gamma} | A_n(\lambda | \alpha, \beta, \gamma) = 0, \quad \forall \lambda \in \mathbb{C}. \tag{227}$$

By the definition of $A_n(\lambda | \alpha, \beta, \gamma)$ it is easy to verify that we have

$$\langle \Omega_{n,\alpha,\beta,\gamma} | A_n(\lambda | \alpha, \beta, \gamma) = \left( -\lambda \sum_{h=0}^{p-1} C_h^+ \langle h, n | + \frac{1}{\lambda} \sum_{h=0}^{p-1} C_h^- \langle h, n | \right) \frac{\beta^2}{1 - \beta^2}, \tag{228}$$

where

$$C_h^+ = q^{-1/2}c_h(\alpha_n q^h + (\beta\gamma/\alpha)\delta_n q^{-h}) + c_{h-1}\gamma q^{1/2}(a_n q^{h-1/2} + b_n q^{1/2-h}), \quad (229)$$

$$C_h^- = q^{1/2}c_h(\beta_n q^{-h} + (\beta\gamma/\alpha)\gamma_n q^h) + (\beta/\alpha)q^{-1/2}c_{h+1}(c_n q^{h+1/2} + d_n q^{-1/2-h}), \quad (230)$$

and we omit to write explicitly the dependence on $n, \alpha, \beta, \gamma$ in $c_h$ when it is not misleading. So that we get the following system of equations:

$$C_h^+ = 0, \quad C_h^- = 0 \quad \forall h \in \{0, ..., p-1\}. \quad (231)$$

As we have assumed that the bulk parameters are generic and satisfy (218), the equations $C_h^+ = 0$ fix the values of the ratios $E_h \equiv c_{h-1}/c_h$ and the equations $C_h^- = 0$ fix the value of ratios $F_h \equiv c_{h+1}/c_h$ for any $h \in \{0, ..., p-1\}$ and one has to impose the compatibility of these values,

$$E_h = 1/F_{h-1} \ \forall h \in \{0, ..., p-1\}, \quad (232)$$

together with the cyclicity condition

$$\prod_{h=0}^{p-1} E_h = 1. \quad (233)$$

Then it is easy to show that the only solution of this system of equation is obtained fixing the two gauge parameters by (219) which correspondingly fixes the form of the state (220).

Let us compute now the action of the operator $B_n(\lambda|\alpha,\beta)$ on this state; by definition it holds

$$B_n(\lambda|\alpha,\beta) = \frac{\beta^2 D_n(\lambda) - \alpha\beta\lambda q^{1/2}B_n}{\beta^2 - 1}, \quad (234)$$

so that

$$\langle\Omega_{n,\alpha,\beta,\gamma}|B_n(\lambda|\alpha,\beta)|h_n,n\rangle =$$
$$\frac{\beta^2}{1-\beta^2}\left\{\lambda\left[c_{h_n}\frac{q^{-1/2}\delta_n}{q^{h_n}} + \frac{\alpha}{\beta}c_{h_n-1}(a_n q^{h_n-1/2} + b_n q^{1/2-h_n})\right] - c_{h_n}\frac{q^{1/2}\gamma_n q^{h_n}}{\lambda}\right\}$$
$$= \frac{\beta^2 c_{h_n} q^{h_n}}{\beta^2 - 1}\left\{\frac{\lambda}{q^{1/2}}\frac{\alpha}{\beta\gamma}\alpha_n + \frac{q^{1/2}\gamma_n}{\lambda}\right\}, \quad (235)$$

where to get the third line we used the identity $C_{h_n}^+ = 0$. Now remarking that

$$c_{h_n}(n, \alpha, \beta, \gamma)q^{h_n} = c_{h_n}(n, \alpha, \beta/q, \gamma q), \quad (236)$$

as the effect of $q^{h_n}$ is to bring $k_n$ to $k_n + 1$ in the state $\langle\Omega_{n,\alpha,\beta,\gamma}|$, this, for the gauge choice (219), being equivalent to the above redefinitions of the gauge parameters. So that we get

$$\langle\Omega_{n,\alpha,\beta,\gamma}|B_n(\lambda|\alpha,\beta) = b_n(\lambda|\alpha,\beta)\langle\Omega_{n,\alpha,\beta/q,\gamma q}|, \quad (237)$$

and so

$$b_n(\lambda|\alpha,\beta) = \frac{\beta^2}{\beta^2 - 1}\left\{\frac{\lambda}{q^{1/2}}\frac{\alpha}{\beta\gamma}\alpha_n + \frac{q^{1/2}\gamma_n}{\lambda}\right\} \quad (238)$$

$$= \frac{\beta^2}{\beta^2 - 1}\begin{cases} \frac{q^{1/2}\gamma_n}{\lambda} + \frac{\lambda}{q^{1/2}}q^{-1}\frac{a_n d_n}{\beta_n} & \text{for } \epsilon_n = -1 \\ \frac{q^{1/2}\gamma_n}{\lambda} + \frac{\lambda}{q^{1/2}}q^{-1}\frac{c_n b_n}{\beta_n} & \text{for } \epsilon_n = 1 \end{cases}. \quad (239)$$

Similarly, one can prove our statements for the right state and the action on it of $B_n(\lambda|\alpha,\beta)$. $\quad\square$

Let us remark that if the condition (218) are not satisfied we can still derive the left and right local reference states imposing some case dependent condition on the gauge parameters; here for simplicity we have chosen to omit the description of these cases.

**Proposition A.1.** *Let us assume that for any $n \in \{1, ..., \mathsf{N}\}$ the conditions (218) is satisfied then the non-zero left state annihilated by the operator family $A(\lambda | \alpha, \beta, \gamma)$ exists if and only if we impose the following $\mathsf{N} + 1$ constraints on the bulk and gauge parameters*

$$\gamma = z_1^{(\epsilon_1, k_1)}, \quad \beta/\alpha = w_{\mathsf{N}}^{(\epsilon_{\mathsf{N}}, k_{\mathsf{N}})}, \quad w_n^{(\epsilon_n, k_n)} = 1/z_{n+1}^{(\epsilon_{n+1}, k_{n+1})} \quad \forall n \in \{1, ..., \mathsf{N} - 1\}, \tag{240}$$

*for fixed $\mathsf{N}$-tuples of $\epsilon_n = \pm 1$ and $k_n \in \{0, ..., p-1\}$, moreover it is uniquely defined by*

$$\langle \Omega, \alpha, \beta, \gamma | = \sum_{h_1, ..., h_{\mathsf{N}} = 0}^{p-1} \prod_{n=1}^{\mathsf{N}} q^{h_n(k_n+1)} \left[ \prod_{r_n=1}^{h_n} \frac{a_n q^{r_n - 1/2} + b_n q^{1/2 - r_n}}{c_n q^{r_n - 1/2} + d_n q^{1/2 - r_n}} \left( \frac{a_n^p + d_n^p}{a_n^p + b_n^p} \right)^{1/p} \right]^{(1+\epsilon_n)/2}$$

$$\bigotimes_{n=1}^{\mathsf{N}} \langle h_n, n |. \tag{241}$$

*Under the same condition the non-zero right state annihilated by the operator family $A(\lambda | \alpha, \beta, \gamma)$ exists if and only if we impose the following $\mathsf{N} + 1$ constraints on the bulk and gauge parameters,*

$$\gamma = z_1^{(\epsilon_1, k_1 - 2)}, \quad \beta/\alpha = w_{\mathsf{N}}^{(\epsilon_{\mathsf{N}}, k_{\mathsf{N}})}, \quad w_n^{(\epsilon_n, k_n)} = 1/z_{n+1}^{(\epsilon_{n+1}, k_{n+1} - 2)} \quad \forall n \in \{1, ..., \mathsf{N} - 1\}, \tag{242}$$

*for fixed $\mathsf{N}$-tuples of $\epsilon_n = \pm 1$ and $k_n \in \{0, ..., p-1\}$, moreover it is uniquely defined by*

$$|\Omega, \alpha, \beta, \gamma\rangle = \sum_{h_1, ..., h_{\mathsf{N}} = 0}^{p-1} \prod_{n=1}^{\mathsf{N}} q^{-h_n k_n} \left[ \prod_{r_n=1}^{h_n} \frac{c_n q^{r_n - 1/2} + d_n q^{1/2 - r_n}}{a_n q^{r_n - 1/2} + b_n q^{1/2 - r_n}} \left( \frac{a_n^p + b_n^p}{c_n^p + d_n^p} \right)^{1/p} \right]^{(1+\epsilon_n)/2}$$

$$\bigotimes_{n=1}^{\mathsf{N}} |h_n, n\rangle. \tag{243}$$

*These are pseudo-eigenstates of $B(\lambda | \alpha, \beta)$,*

$$\langle \Omega, \alpha, \beta, \gamma | B(\lambda | \alpha, \beta) = b(\lambda | \alpha, \beta) \langle \Omega, \alpha, \beta/q, \gamma q |, \tag{244}$$

$$B(\lambda | \alpha, \beta) | \Omega, \alpha, \beta, \gamma \rangle = | \Omega, \alpha, \beta q, \gamma/q \rangle q^{\mathsf{N}} b(\lambda q | \alpha, \beta), \tag{245}$$

*with*

$$b(\lambda | \alpha, \beta) = \prod_{n=1}^{\mathsf{N}} b_n(\lambda | \alpha, \beta). \tag{246}$$

*Proof.* The operator family $A(\lambda | \alpha, \beta, \gamma)$ is a degree $\mathsf{N}$ Laurent polynomial of the form

$$A(\lambda | \alpha, \beta, \gamma) = \sum_{n=0}^{\mathsf{N}} \lambda^{2n - \mathsf{N}} A_n(\alpha, \beta, \gamma), \tag{247}$$

where the $A_n(\alpha, \beta, \gamma)$ are operators, for example we write explicitly as

$$A_0(\alpha, \beta, \gamma) =$$

$$\frac{-\alpha \prod_{n=1}^{\mathsf{N}} \beta_n v_n^{-1} + \beta \gamma \prod_{n=1}^{\mathsf{N}} \gamma_n v_n + q^{-1/2} \beta \sum_{a=1}^{\mathsf{N}} \left( \prod_{n=a+1}^{\mathsf{N}} \beta_n v_n^{-1} \right) C_a \prod_{n=1}^{a-1} \gamma_n v_n}{(\beta - 1/\beta)\gamma}, \tag{248}$$

$$A_N(\alpha,\beta,\gamma) =$$
$$\frac{-\alpha\prod_{n=1}^{N}\alpha_n\nu_n + \beta\gamma\prod_{n=1}^{N}\delta_n\nu_n^{-1} - q^{1/2}\alpha\gamma\sum_{a=1}^{N}\left(\prod_{n=a+1}^{N}\alpha_n\nu_n\right)B_a\prod_{n=1}^{a-1}\delta_n\nu_n^{-1}}{(\beta-1/\beta)\gamma}. \quad (249)$$

For general values of the parameters these are invertible operators so that we have to impose at least $N+1$ constraints to have that their common kernel is at least one dimensional. We can find the set of constraints by using induction and decomposing $A(\lambda|\alpha,\beta,\gamma)$ in terms of gauged operators on two subchains one of $N-1$ sites and one of 1 site. The most general decomposition reads

$$A_{N,\ldots,1}(\lambda|\alpha,\beta,\gamma) = A_{N,\ldots,2}(\lambda|\alpha,\beta,x_1,y_1)A_1(\lambda|x_1,y_1,\gamma)$$
$$+ B_{N,\ldots,2}(\lambda|\alpha,\beta,x_1,y_1)C_1(\lambda|x_1,y_1,\gamma), \quad (250)$$

where we have defined

$$M(\lambda|\alpha,\beta,x,y) = G^{-1}(\lambda q^{1/2}|\alpha,\beta)M(\lambda)G(\lambda q^{1/2}|x,y)$$
$$= \left(\begin{array}{cc} A(\lambda|\alpha,\beta,x,y) & B(\lambda|\alpha,\beta,x,y) \\ C(\lambda|\alpha,\beta,x,y) & D(\lambda|\alpha,\beta,x,y) \end{array}\right), \quad (251)$$

and we have explicitly pointed out in the subscripts the quantum sites to which the operator are referred. The following identities holds:

$$A(\lambda|\alpha,\beta,x,y) = -A(\lambda|\alpha,\beta,xy), \quad (252)$$
$$B(\lambda|\alpha,\beta,x,y) = A(\lambda|\alpha,\beta,x/y), \quad (253)$$
$$C(\lambda|x,y,\gamma) = \frac{x^2y^2-1}{1-y^2}A(\lambda|1,1/xy,\gamma), \quad (254)$$
$$A(\lambda|x,y,\gamma) = \frac{x^2-y^2}{1-y^2}A(\lambda|1,y/x,\gamma), \quad (255)$$

from which it follows

$$A_{N,\ldots,1}(\lambda|\alpha,\beta,\gamma) = \frac{\left(y_1^2-x_1^2\right)A_{N,\ldots,2}(\lambda|\alpha,\beta,x_1y_1)A_1(\lambda|1,y_1/x_1,\gamma)}{1-y_1^2}$$
$$+ \frac{(x_1^2y_1^2-1)A_{N,\ldots,2}(\lambda|\alpha,\beta,x_1/y_1)A_1(\lambda|1,1/x_1y_1,\gamma)}{1-y_1^2}. \quad (256)$$

Then $A_{N,\ldots,1}(\lambda|\alpha,\beta,\gamma)$ admits a non-zero state annihilated by its action once we impose that it is true for $A_{N,\ldots,2}(\lambda|\alpha,\beta,x_1y_1)$ and $A_1(\lambda|1,1/x_1y_1,\gamma)$ or for $A_1(\lambda|1,y_1/x_1,\gamma)$ and $A_{N,\ldots,2}(\lambda|\alpha,\beta,x_1/y_1)$, and this state is given by the tensor product of the ones on the two subchains. As the parameters $x_1$ and $y_1$ are arbitrary in fact these two conditions are equivalents and so we can chose just one of them. So let us say we ask the second one and we repeat the same argument for $A_{N,\ldots,2}(\lambda|\alpha,\beta,x_1/y_1)$, i.e. $A_{N,\ldots,2}(\lambda|\alpha,\beta,x_1/y_1)$ admits such a state if $A_2(\lambda|1,y_2/x_2,x_1/y_1)$ and $A_{N,\ldots,3}(\lambda|\alpha,\beta,x_2/y_2)$ do. So on by induction we get that the existence condition is equivalent to the existence conditions for the following $N$ local operators:

$$A_n(\lambda|1,y_n/x_n,x_{n-1}/y_{n-1}) \text{ for any } n \in \{1,\ldots,N\}, \quad (257)$$

where we have denoted

$$y_N/x_N = \beta/\alpha, \quad x_0/y_0 = \gamma, \quad (258)$$

while the $y_n/x_n$ for any $n \in \{1, ..., N-1\}$ are free parameters to be used to satisfy the existence condition for the local operators $A_n(\lambda|1, y_n/x_n, x_{n-1}/y_{n-1})$. From the previous lemma for $A_n(\lambda|1, s_n, r_n)$, the existence condition is equivalent to

$$s_n = w_n^{(\epsilon_n)} \quad \text{and} \quad r_n = z_n^{(\epsilon_n)} \tag{259}$$

for any $\epsilon_n = \pm 1$ and the right state annihilated by $A_n(\lambda|1, s_n, r_n)$ reads

$$\langle \Omega_n, s_n, r_n | = \sum_{h_n=0}^{p-1} q^{h_n(k_n+1)} \left[ \prod_{k_n=1}^{h_n} \frac{a_n q^{k_n-1/2} + b_n q^{1/2-k_n}}{c_n q^{k_n-1/2} + d_n q^{1/2-k_n}} \left( \frac{c_n^p + d_n^p}{a_n^p + b_n^p} \right)^{1/p} \right]^{(1+\epsilon_n)/2} \langle h_n, n|. \tag{260}$$

From this it is clear that the existence conditions of such a state for $A_{N,...,1}(\lambda|\alpha, \beta, \gamma)$ coincides with the simultaneous existence for the N local operators (257) and that the state is just the tensor product of the states (260) so that our proposition is proven. Similarly, we can prove the statement for the right state and using the previous lemma we can prove our statement on the action of the operator $B(\lambda|\alpha, \beta)$ on these states. □

# B  Gauge transformed Reflection algebra

## B.1  Gauge transformed boundary operators

The gauged two-row monodromy matrix can be defined as it follows:

$$\mathcal{U}_-(\lambda|\alpha, \beta) \equiv \frac{q^{1/2}}{\lambda} G^{-1}(\lambda q^{1/2}|\alpha, \beta) \mathcal{U}_-(\lambda) G(q^{1/2}/\lambda|\alpha, \beta)$$

$$= \begin{pmatrix} \mathcal{A}_-(\lambda|\alpha, \beta q^2) & \mathcal{B}_-(\lambda|\alpha, \beta) \\ \mathcal{C}_-(\lambda|\alpha, \beta q^2) & \mathcal{D}_-(\lambda|\alpha, \beta) \end{pmatrix}. \tag{261}$$

Note that one can expand this last gauged monodromy matrix in terms of the gauged bulk ones. Moreover, $\mathcal{U}_-(\lambda|\alpha, \beta)$ does not depend on the internal gauge parameter $\gamma$, so we are free to chose it at will. The following decompositions hold:

$$\begin{pmatrix} \mathcal{A}_-(\lambda|\alpha, \beta q^2) \\ \mathcal{C}_-(\lambda|\alpha, \beta q^2) \end{pmatrix} = M(\lambda|\alpha, \beta, \gamma) \bar{K}_-(\lambda|\gamma) \begin{pmatrix} \bar{A}(\lambda|\alpha, \beta q, \gamma q) \\ \bar{C}(\lambda|\alpha, \beta q, \gamma q) \end{pmatrix}, \tag{262}$$

$$\begin{pmatrix} \mathcal{B}_-(\lambda|\alpha, \beta) \\ \mathcal{D}_-(\lambda|\alpha, \beta) \end{pmatrix} = M(\lambda|\alpha, \beta, \gamma) \bar{K}_-(\lambda|\gamma) \begin{pmatrix} \bar{B}(\lambda|\alpha, \beta/q, \gamma q) \\ \bar{D}(\lambda|\alpha, \beta/q, \gamma q) \end{pmatrix}, \tag{263}$$

where

$$\bar{K}_-(\lambda|\gamma) = \frac{q^{1/2}}{\lambda} \bar{G}^{-1}(\lambda q^{1/2}|\gamma) K_-(\lambda) \bar{G}(q^{1/2}/\lambda|\gamma q). \tag{264}$$

Explicitly, for $\mathcal{B}_-(\lambda|\alpha, \beta)$, it holds

$$\frac{\mathcal{B}_-(\lambda|\alpha, \beta)}{f(\alpha, \beta/q, \gamma q)}$$
$$= \bar{K}_-(\lambda|\gamma)_{12} A(\lambda|\alpha, \beta, \gamma) A(1/\lambda|\alpha, \beta/q, \gamma q) - \bar{K}_-(\lambda|\gamma)_{11} A(\lambda|\alpha, \beta, \gamma) B(1/\lambda|\alpha, \beta/q)$$
$$+ \bar{K}_-(\lambda|\gamma)_{21} B(\lambda|\alpha, \beta) B(1/\lambda|\alpha, \beta/q) - \bar{K}_-(\lambda|\gamma)_{22} B(\lambda|\alpha, \beta) A(1/\lambda|\alpha, \beta/q, \gamma q). \tag{265}$$

where

$$\bar{K}_-(\lambda|\gamma)_{11} = \frac{\lambda^2(\zeta_-/q + q^2\gamma\kappa_- e^{\tau_-}) - q^2\gamma\kappa_- e^{\tau_-}/(\zeta_-\lambda^2) - 1/\zeta_-}{\zeta_- - 1/\zeta_-}, \tag{266}$$

$$\bar{K}_-(\lambda|\gamma)_{22} = \frac{(\zeta_- + q\gamma\kappa_- e^{\tau_-})/\lambda^2 - \gamma\lambda^2 - 1/\zeta_-}{\zeta_- - 1/\zeta_-}, \tag{267}$$

$$\bar{K}_-(\lambda|\gamma)_{12} = \frac{q\gamma\kappa_- e^{\tau_-}(\lambda^2/q - q/\lambda^2)}{\zeta_- - 1/\zeta_-}, \tag{268}$$

$$\bar{K}_-(\lambda|\gamma)_{21} = \frac{(q/\lambda^2 - \lambda^2/q)[q\gamma e^{\tau_-}\kappa_- + \zeta_- - \kappa_-/(q\gamma e^{\tau_-})]}{\zeta_- - 1/\zeta_-}. \tag{269}$$

Then it holds

$$\bar{K}_-(\lambda|\gamma)_{21} \equiv 0, \ \ \forall \lambda \in \mathbb{C} \ \ \text{for } \gamma = \gamma_\epsilon \tag{270}$$

for $\epsilon = \pm$ and

$$\gamma_\epsilon = \frac{-\zeta_- + \epsilon\sqrt{\zeta_-^2 + 4\kappa_-^2}}{2qe^{\tau_-}\kappa_-}. \tag{271}$$

These gauge transformed boundary operators satisfies the following gauge deformed reflection algebra.

**Proposition B.1.** *The gauge transformed boundary operators satisfy the following commutation relations:*

$$\mathcal{B}_-(\lambda_2|\beta)\mathcal{B}_-(\lambda_1|\beta/q^2) = \mathcal{B}_-(\lambda_1|\beta)\mathcal{B}_-(\lambda_2|\beta/q^2), \tag{272}$$

$$
\begin{aligned}
\mathcal{A}_-(\lambda_2|\beta q^2)\mathcal{B}_-(\lambda_1|\beta) &= \frac{(\lambda_1 q/\lambda_2 - \lambda_2/q\lambda_1)(\lambda_1\lambda_2/q - q/\lambda_1\lambda_2)}{(\lambda_1/\lambda_2 - \lambda_2/\lambda_1)(\lambda_1\lambda_2 - 1/\lambda_1\lambda_2)}\mathcal{B}_-(\lambda_1|\beta)\mathcal{A}_-(\lambda_2|\beta) \\
&+ \frac{(\lambda_1\lambda_2/q - q/\lambda_1\lambda_2)(\lambda_1\beta/q\lambda_2 - \lambda_2 q/\beta\lambda_1)(q - 1/q)}{(\lambda_1/\lambda_2 - \lambda_2/\lambda_1)(\lambda_1\lambda_2 - 1/\lambda_1\lambda_2)(\beta/q - q/\beta)}\mathcal{B}_-(\lambda_2|\beta)\mathcal{A}_-(\lambda_1|\beta) \\
&+ \frac{(\lambda_1\lambda_2/\beta - \beta/\lambda_1\lambda_2)(q - 1/q)}{(\lambda_1\lambda_2 - 1/\lambda_1\lambda_2)(\beta/q - q/\beta)}\mathcal{B}_-(\lambda_2|\beta)\mathcal{D}_-(\lambda_1|\beta), \quad (273)
\end{aligned}
$$

$$
\begin{aligned}
\mathcal{B}_-(\lambda_1|\beta)\mathcal{D}_-(\lambda_2|\beta) &= \frac{(\lambda_1 q/\lambda_2 - \lambda_2/q\lambda_1)(\lambda_1\lambda_2/q - q/\lambda_1\lambda_2)}{(\lambda_1/\lambda_2 - \lambda_2/\lambda_1)(\lambda_1\lambda_2 - 1/\lambda_1\lambda_2)}\mathcal{D}_-(\lambda_2|\beta q^2)\mathcal{B}_-(\lambda_1|\beta) \\
&- \frac{(\lambda_1\lambda_2/q - q/\lambda_1\lambda_2)(\lambda_2\beta q/\lambda_1 - \lambda_1/\lambda_2\beta q)(q - 1/q)}{(\lambda_1/\lambda_2 - \lambda_2/\lambda_1)(\lambda_1\lambda_2 - 1/\lambda_1\lambda_2)(\beta q - 1/\beta q)}\mathcal{D}_-(\lambda_1|\beta q^2)\mathcal{B}_-(\lambda_2|\beta) \\
&- \frac{(\lambda_1\lambda_2\beta - 1/\lambda_1\lambda_2\beta)(q - 1/q)}{(\lambda_1\lambda_2 - 1/\lambda_1\lambda_2)(\beta q - 1/q\beta)}\mathcal{A}_-(\lambda_1|\beta q^2)\mathcal{B}_-(\lambda_2|\beta), \quad (274)
\end{aligned}
$$

*and*

$$
\begin{aligned}
\mathcal{A}_-(\lambda_1|\beta q^2)\mathcal{A}_-(\lambda_2|\beta q^2) &- \frac{(\lambda_1\lambda_2/\beta - 1/\lambda_1\lambda_2)(q - 1/q)}{(\lambda_1\lambda_2 - 1/\lambda_1\lambda_2)(\beta/q - q/\beta)}\mathcal{B}_-(\lambda_1|\beta)\mathcal{C}_-(\lambda_2|\beta q^2) = \\
\mathcal{A}_-(\lambda_2|\beta q^2)\mathcal{A}_-(\lambda_1|\beta q^2) &- \frac{(\lambda_1\lambda_2/\beta - 1/\lambda_1\lambda_2)(q - 1/q)}{(\lambda_1\lambda_2 - 1/\lambda_1\lambda_2)(\beta/q - q/\beta)}\mathcal{B}_-(\lambda_2|\beta)\mathcal{C}_-(\lambda_1|\beta q^2). \quad (275)
\end{aligned}
$$

*Similar commutation relations involving $\mathcal{C}_-(\lambda|\beta)$ can be written by using the following $\beta$-symmetries:*

$$\mathcal{B}_-(\lambda|\beta) = \mathcal{C}_-(\lambda|q^2/\beta), \quad \mathcal{A}_-(\lambda|\beta) = \mathcal{D}_-(\lambda|q^2/\beta). \tag{276}$$

*Moreover, these gauge transformed operators satisfy the following parity properties:*

$$\mathscr{A}_-(\lambda|\beta) = -\frac{(q-1/q)(\lambda^2 q/\beta - \beta/\lambda^2 q)}{(\beta/q^2 - q^2/\beta)(\lambda^2 - 1/\lambda^2)}\mathscr{D}_-(\lambda|\beta)+ \tag{277}$$

$$\frac{(\beta/q - q/\beta)(\lambda^2/q - q/\lambda^2)}{(\beta/q^2 - q^2/\beta)(\lambda^2 - 1/\lambda^2)}\mathscr{D}_-(1/\lambda|\beta), \tag{278}$$

$$\mathscr{D}_-(\lambda|\beta) = \frac{(q-1/q)(\lambda^2\beta/q - q/\lambda^2\beta)}{(\beta/q^2 - q^2/\beta)(\lambda^2 - 1/\lambda^2)}\mathscr{A}_-(\lambda|\beta)+ \tag{279}$$

$$\frac{(\beta/q - q/\beta)(\lambda^2/q - q/\lambda^2)}{(\beta - 1/\beta)(\lambda^2 - 1/\lambda^2)}\mathscr{A}_-(1/\lambda|\beta), \tag{280}$$

$$\mathscr{B}_-(1/\lambda|\beta) = -\frac{(\lambda^2 q - 1/q\lambda^2)}{(\lambda^2/q - q/\lambda^2)}\mathscr{B}_-(\lambda|\beta)\,, \quad \mathscr{C}_-(1/\lambda|\beta) = -\frac{(\lambda^2 q - 1/q\lambda^2)}{(\lambda^2/q - q/\lambda^2)}\mathscr{C}_-(\lambda|\beta). \tag{281}$$

*Proof.* Both the commutation relations and the parity properties here presented coincide with those derived in [80] for the case of the XXZ spin 1/2 quantum chain with general integrable boundaries. This is the case as they are clearly representation independent. Here we are just writing them in a Laurent polynomial form instead of a trigonometric form. □

## B.2 Representation of the gauge transformed Reflection algebra

In the bulk of the paper we have anticipated that for almost all the values of the boundary, bulk and gauge parameters the operator family $\mathscr{B}_-(\lambda|\beta)$ is pseudo-diagonalizable. We will show this statement in the last subsection of this appendix, but for now we want to write explicitly the representation of the other gauge transformed boundary operator families in the left and right basis formed out of the pseudo-eigenstates of $\mathscr{B}_-(\lambda|\beta)$.

**Theorem B.1.** *The action of the reflection algebra generator $\mathscr{A}_-(\lambda|\beta q^2)$ on the generic state $\langle\beta, \mathbf{h}|$ is given by the following expression:*

$$\langle\beta, \mathbf{h}|\mathscr{A}_-(\lambda|\beta q^2) =$$

$$\sum_{a=1}^{2N} \frac{\zeta_a^{(h_a)}(\lambda^2/q - q/\lambda^2)(\lambda\zeta_a^{(h_a)} - 1/(\lambda\zeta_a^{(h_a)}))A_-(\zeta_a^{(h_a)})}{\lambda\left((\zeta_a^{(h_a)})^2/q - q/(\zeta_a^{(h_a)})^2\right)\left((\zeta_a^{(h_a)})^2 - 1/(\zeta_a^{(h_a)})^2\right)} \prod_{\substack{b=1 \\ b\neq a \bmod N}}^{N} \frac{\Lambda - X_b^{(h_b)}}{X_a^{(h_a)} - X_b^{(h_b)}}$$

$$\times \langle\beta, \mathbf{h}|T_a^{-\varphi_a} + (-1)^N \frac{q^{1/2}}{2\lambda}det_q M(1)(\frac{\lambda}{q^{1/2}} + \frac{q^{1/2}}{\lambda})\prod_{b=1}^{N}\frac{\Lambda - X_b^{(h_b)}}{X - X_b^{(h_b)}}\langle\beta, \mathbf{h}|$$

$$+ (-1)^{N+1}\frac{iq^{1/2}}{2\lambda}\frac{\zeta_- + 1/\zeta_-}{\zeta_- - 1/\zeta_-}det_q M(i)(\frac{\lambda}{q^{1/2}} - \frac{q^{1/2}}{\lambda})\prod_{b=1}^{N}\frac{\Lambda - X_b^{(h_b)}}{X + X_b^{(h_b)}}\langle\beta, \mathbf{h}|$$

$$+ q\left(\lambda^2/q - q/\lambda^2\right)\prod_{b=1}^{N}(\Lambda - X_b^{(h_b)})\langle\beta, \mathbf{h}|\mathscr{A}_-^{\infty}(\beta q^2), \tag{282}$$

*where*

$$\langle\beta,\mathbf{h}|\mathscr{A}_-^\infty(\beta q^2) = \frac{1}{q(1-\beta^2)}\left[\sum_{a=1}^{2N}\frac{\prod_{\substack{b=1\\b\neq a\ mod N}}^N(X_a^{(h_a)}-X_b^{(h_b)})^{-1}\mathsf{A}_-(\zeta_a^{(h_a)})\langle\beta,\mathbf{h}|T_a^{-\varphi_a}}{\left((\zeta_a^{(h_a)})^2/q-q/(\zeta_a^{(h_a)})^2\right)\left((\zeta_a^{(h_a)})^2-1/(\zeta_a^{(h_a)})^2\right)}\right.$$

$$\left.+(-1)^N\left(\frac{1}{2}det_qM(1)\prod_{b=1}^N\frac{1}{X-X_b^{(h_b)}}+\frac{i}{2}\frac{\zeta_-+1/\zeta_-}{\zeta_--1/\zeta_-}det_qM(i)\prod_{b=1}^N\frac{1}{X+X_b^{(h_b)}}\right)\langle\beta,\mathbf{h}|\right]$$

$$+\frac{\kappa_-}{q(\zeta_--1/\zeta_-)}\left(\frac{\prod_{a=1}^N\gamma_a\delta_a}{q\alpha e^{\tau_-}}-q\alpha e^{\tau_-}\prod_{a=1}^N\alpha_a\beta_a\right)\langle\beta,\mathbf{h}|,$$

*and*

$$\langle\beta,h_1,...,h_a,...,h_N|T_a^\pm = \langle\beta,h_1,...,h_a\pm1,...,h_N|, \tag{283}$$

*once the parameter $\alpha$ has been fixed by* (51).

*Proof.* The following interpolation formula:

$$\langle\beta,\mathbf{h}|\mathscr{A}_-(\lambda|\beta q^2) = \sum_{a=1}^{2N}\frac{\zeta_a^{(h_a)}\left(\lambda^2/q-q/\lambda^2\right)\mathsf{A}_-(\zeta_a^{(h_a)})}{\lambda\left((\zeta_a^{(h_a)})^2/q-q/(\zeta_a^{(h_a)})^2\right)}\prod_{\substack{b=1\\b\neq a}}^{2N}\frac{\lambda/\zeta_b^{(h_b)}-\zeta_b^{(h_b)}/\lambda}{\zeta_a^{(h_a)}/\zeta_b^{(h_b)}-\zeta_b^{(h_b)}/\zeta_a^{(h_a)}}$$

$$\times\langle\beta,\mathbf{h}|T_a^{-\varphi_a}+(-1)^N\frac{q^{1/2}}{2\lambda}det_qM(1)(\frac{\lambda}{q^{1/2}}+\frac{q^{1/2}}{\lambda})\prod_{b=1}^N\frac{\lambda/\zeta_b^{(h_b)}-\zeta_b^{(h_b)}/\lambda}{q^{1/2}/\zeta_b^{(h_b)}-\zeta_b^{(h_b)}/q^{1/2}}\langle\beta,\mathbf{h}|$$

$$+(-1)^{N+1}\frac{iq^{1/2}}{2\lambda}\frac{\zeta_-+1/\zeta_-}{\zeta_--1/\zeta_-}det_qM(i)(\frac{\lambda}{q^{1/2}}-\frac{q^{1/2}}{\lambda})\prod_{b=1}^N\frac{\lambda/\zeta_b^{(h_b)}-\zeta_b^{(h_b)}/\lambda}{q^{1/2}/\zeta_b^{(h_b)}-\zeta_b^{(h_b)}/q^{1/2}}\langle\beta,\mathbf{h}|$$

$$+q\left(\lambda^2/q-q/\lambda^2\right)\prod_{b=1}^N(\Lambda-X_b^{(h_b)})\langle\beta,\mathbf{h}|\mathscr{A}_-^\infty(\beta q^2),$$

where we have defined

$$\mathscr{A}_-^{\infty,0}(\beta q^2) = \lim_{\lambda\to\infty,0}\lambda^{\mp2(N+1)}\mathscr{A}_-(\lambda|\beta q^2), \tag{284}$$

is a direct consequence of the functional dependence with respect to $\lambda$,

$$\mathscr{A}_-(\lambda|\beta) = \sum_{a=0}^{2N+2}\lambda^{2(a-(N+1))}\mathscr{A}_a(\beta) \tag{285}$$

and of the identities

$$\mathscr{U}_-(q^{1/2}) = (-1)^N det_qM(1)\,I_0, \quad \mathscr{U}_-(iq^{1/2}) = -i\frac{\zeta_-+1/\zeta_-}{\zeta_--1/\zeta_-}det_qM(i)\,\sigma_0^z, \tag{286}$$

which are representation independent. Instead the asymptotic operators $\mathscr{A}_-^{\infty,0}(\beta q^2)$ depend on the representation and we can compute them observing that using the definition (26) of $\mathscr{A}_-(\lambda|\beta q^2)$ it holds

$$\mathscr{A}_-^\infty(\beta q^2) = \left[-\mathscr{A}_-^\infty/\beta q^{1/2}-\alpha q\mathscr{B}_-^\infty+\mathscr{C}_-^\infty/\alpha q\right]/(\beta-1/\beta), \tag{287}$$

$$\mathscr{A}_-^0(\beta q^2) = \left[\beta q^{1/2}\mathscr{D}_-^0-\alpha q\mathscr{B}_-^0+\mathscr{C}_-^0/\alpha q\right]/(\beta-1/\beta), \tag{288}$$

where the $\mathscr{A}_-^{\infty,0}$, $\mathscr{D}_-^{\infty,0}$, $\mathscr{B}_-^{\infty,0}$ and $\mathscr{C}_-^{\infty,0}$ are the asymptotic limits of the ungauged elements of $\mathscr{U}_-(\lambda)$. The identities

$$\mathscr{A}_-^{\infty,0} = q^{\mp 1}\mathscr{D}_-^{0,\infty}, \quad \mathscr{B}_-^0 = -q^2\mathscr{B}_-^{\infty}, \quad \mathscr{C}_-^0 = -q^2\mathscr{C}_-^{\infty}, \tag{289}$$

following from (278)-(281), and

$$\mathscr{B}_-^{\infty} = \frac{\kappa_- e^{\tau_-}\prod_{a=1}^{N}\alpha_a\beta_a}{q(\zeta_- - 1/\zeta_-)}, \quad \mathscr{C}_-^{\infty} = \frac{\kappa_- e^{-\tau_-}\prod_{a=1}^{N}\gamma_a\delta_a}{q(\zeta_- - 1/\zeta_-)}, \tag{290}$$

imply the main identity,

$$\beta q\mathscr{A}_-^{\infty}(\beta q^2) + \mathscr{A}_-^0(\beta q^2)/(\beta q) = \frac{\kappa_-(\beta - 1/\beta)}{\zeta_- - 1/\zeta_-}\left[\frac{\prod_{a=1}^{N}\gamma_a\delta_a}{q\alpha e^{\tau_-}} - q\alpha e^{\tau_-}\prod_{a=1}^{N}\alpha_a\beta_a\right]. \tag{291}$$

This identity allows to compute these asymptotic operators once we use the interpolation formula to write $\mathscr{A}_-^0(\beta q^2)$ in terms of $\mathscr{A}_-^{\infty}(\beta q^2)$ as it follows,

$$\langle\beta,\mathbf{h}|\mathscr{A}_-^0(\beta q^2) = \sum_{a=1}^{2N}\frac{q\mathsf{A}_-(\zeta_a^{(h_a)})}{\left((\zeta_a^{(h_a)})^2/q - q/(\zeta_a^{(h_a)})^2\right)\left((\zeta_a^{(h_a)})^2 - 1/(\zeta_a^{(h_a)})^2\right)}\prod_{\substack{b=1\\b\neq a\,\mathrm{mod}N}}^{N}\frac{1}{X_a^{(h_a)} - X_b^{(h_b)}}$$

$$\times\,\langle\beta,\mathbf{h}|T_a^{-\varphi_a} + (-1)^N\frac{q}{2}\det_q M(1)\prod_{b=1}^{N}\frac{1}{X - X_b^{(h_b)}}\langle\beta,\mathbf{h}|$$

$$+ (-1)^N\frac{iq}{2}\frac{\zeta_- + 1/\zeta_-}{\zeta_- - 1/\zeta_-}\det_q M(i)\prod_{b=1}^{N}\frac{1}{X + X_b^{(h_b)}}\langle\beta,\mathbf{h}|$$

$$-\,q^2\langle\beta,\mathbf{h}|\mathscr{A}_-^{\infty}(\beta q^2). \tag{292}$$

$\square$

Similarly, the following theorem characterizes the right SoV representation of the gauged cyclic reflection algebra:

**Theorem B.2.** *The action of the reflection algebra generators $\mathscr{D}_-(\lambda|\beta)$ on the generic state $|\beta,\mathbf{h}\rangle$, can be written as it follows,*

$$\mathscr{D}_-(\lambda|\beta)|\beta,\mathbf{h}\rangle = \sum_{a=1}^{2N}T_a^{-\varphi_a}|\beta,\mathbf{h}\rangle\frac{\zeta_a^{(h_a)}\left(\lambda^2/q - q/\lambda^2\right)(\lambda\zeta_a^{(h_a)} - 1/(\lambda\zeta_a^{(h_a)}))\mathsf{D}_-(\zeta_a^{(h_a)})}{\lambda\left((\zeta_a^{(h_a)})^2/q - q/(\zeta_a^{(h_a)})^2\right)\left((\zeta_a^{(h_a)})^2 - 1/(\zeta_a^{(h_a)})^2\right)}\cdot$$

$$\cdot\prod_{\substack{b=1\\b\neq a\,\mathrm{mod}N}}^{N}\frac{\Lambda - X_b^{(h_b)}}{X_a^{(h_a)} - X_b^{(h_b)}} + |\beta,\mathbf{h}\rangle(-1)^N\frac{q^{1/2}}{2\lambda}\det_q M(1)(\frac{\lambda}{q^{1/2}} + \frac{q^{1/2}}{\lambda})\prod_{b=1}^{N}\frac{\Lambda - X_b^{(h_b)}}{X - X_b^{(h_b)}}$$

$$+ |\beta,\mathbf{h}\rangle(-1)^N\frac{iq^{1/2}}{2\lambda}\frac{\zeta_- + 1/\zeta_-}{\zeta_- - 1/\zeta_-}\det_q M(i)(\frac{\lambda}{q^{1/2}} - \frac{q^{1/2}}{\lambda})\prod_{b=1}^{N}\frac{\Lambda - X_b^{(h_b)}}{X + X_b^{(h_b)}}$$

$$+ q\left(\lambda^2/q - q/\lambda^2\right)\prod_{b=1}^{N}(\Lambda - X_b^{(h_b)})\mathscr{D}_-^{\infty}(\beta)|\beta,\mathbf{h}\rangle, \tag{293}$$

*where*

$$\mathscr{D}_-^\infty(\beta) =$$

$$\frac{\beta}{q(\beta-1/\beta)}\left[\sum_{a=1}^{2N}\frac{D_-(\zeta_a^{(h_a)})T_a^{-\varphi_a}|\beta,\mathbf{h}\rangle}{\left((\zeta_a^{(h_a)})^2/q-q/(\zeta_a^{(h_a)})^2\right)\left((\zeta_a^{(h_a)})^2-1/(\zeta_a^{(h_a)})^2\right)}\prod_{\substack{b=1\\b\neq a\,modN}}^{N}\frac{1}{X_a^{(h_a)}-X_b^{(h_b)}}\right.$$

$$\left.+(-1)^N\left(\frac{1}{2}det_qM(1)\prod_{b=1}^{N}\frac{1}{X-X_b^{(h_b)}}-\frac{i}{2}\frac{\zeta_-+1/\zeta_-}{\zeta_--1/\zeta_-}det_qM(i)\prod_{b=1}^{N}\frac{1}{X+X_b^{(h_b)}}\right)|\beta,\mathbf{h}\rangle\right]$$

$$+\frac{\kappa_-[q\alpha e^{\tau_-}\prod_{a=1}^{N}\alpha_a\beta_a-\frac{\prod_{a=1}^{N}\gamma_a\delta_a}{q\alpha e^{\tau_-}}]}{q(\zeta_--1/\zeta_-)}|\beta,\mathbf{h}\rangle,$$

*and*

$$T_a^\pm|\beta,h_1,...,h_a,...,h_N\rangle=|\beta,h_1,...,h_a\pm 1,...,h_N\rangle. \tag{294}$$

*Proof.* The following interpolation formula is derived as in the previous theorem using the polynomiality of the operator family $\mathscr{D}_-(\lambda|\beta)$:

$$\mathscr{D}_-(\lambda|\beta)=\sum_{a=0}^{2N+2}\lambda^{(2a-(2N+2))}\mathscr{D}_a(\beta), \tag{295}$$

we have just to compute the asymptotic operator $\mathscr{D}_-^\infty(\beta)$. The following identities,

$$\mathscr{D}_-^\infty(\beta) = \left[\beta\mathscr{A}_-^\infty/q^{1/2}+\alpha q\mathscr{B}_-^\infty-\mathscr{C}_-^\infty/\alpha q\right]/(\beta-1/\beta), \tag{296}$$

$$\mathscr{D}_-^0(\beta) = -\left[q^{3/2}\mathscr{A}_-^\infty/\beta+q^2(\alpha q\mathscr{B}_-^\infty-\mathscr{C}_-^\infty/\alpha q)\right]/(\beta-1/\beta) \tag{297}$$

trivially follows by the definition of the operator family $\mathscr{D}_-(\lambda|\beta)$, from which we get

$$q\mathscr{D}_-^\infty(\beta)/\beta+\beta\mathscr{D}_-^0(\beta)/q=\frac{\kappa_-(\beta-1/\beta)}{\zeta_--1/\zeta_-}\left[\frac{\prod_{a=1}^{N}\gamma_a\delta_a}{q\alpha e^{\tau_-}}-q\alpha e^{\tau_-}\prod_{a=1}^{N}\alpha_a\beta_a\right], \tag{298}$$

while from the interpolation formula we get

$$\mathscr{D}_-^0(\beta)=\sum_{a=1}^{2N}\frac{qD_-(\zeta_a^{(h_a)})T_a^{-\varphi_a}|\beta,\mathbf{h}\rangle}{\left((\zeta_a^{(h_a)})^2/q-q/(\zeta_a^{(h_a)})^2\right)\left((\zeta_a^{(h_a)})^2-1/(\zeta_a^{(h_a)})^2\right)}\prod_{\substack{b=1\\b\neq a\,modN}}^{N}\frac{1}{X_a^{(h_a)}-X_b^{(h_b)}}$$

$$+(-1)^N\left(\frac{q}{2}det_qM(1)\prod_{b=1}^{N}\frac{1}{X-X_b^{(h_b)}}-\frac{iq}{2}\frac{\zeta_-+1/\zeta_-}{\zeta_--1/\zeta_-}det_qM(i)\prod_{b=1}^{N}\frac{1}{X+X_b^{(h_b)}}\right)|\beta,\mathbf{h}\rangle$$

$$-q^2\mathscr{D}_-^\infty(\beta)|\beta,\mathbf{h}\rangle. \tag{299}$$

from which the statement of the theorem follows easily. $\square$

## B.3 SoV spectral decomposition of the identity

The Theorem 3.1 states the pseudo-diagonalizability of $\mathscr{B}_-(\lambda|\beta)$ for almost all the values of the boundary-bulk-gauge parameters, so that for almost all the values of these parameters the left and right states $\langle\beta,\mathbf{h}|$ and $|\beta,\mathbf{k}\rangle$ are well defined nonzero left and right states describing a left and right basis in the space of the representation.

We can now defines the following $p^N \times p^N$ matrices $U^{(L,\beta)}$ and $U^{(R,\beta q^2)}$ defining the change of basis from the original left and right basis,

$$\underline{\langle \mathbf{h}|} \equiv \otimes_{n=1}^N \langle h_n, n| \quad \text{and} \quad \underline{|\mathbf{h}\rangle} \equiv \otimes_{n=1}^N |h_n, n\rangle, \tag{300}$$

composed by $\nu_n$-eigenstates, to the left and right pseudo-eigenbasis of $\mathscr{B}_-(\lambda|\beta)$,

$$\langle \beta, \mathbf{h}| = \underline{\langle \mathbf{h}|} U^{(L,\beta)} = \sum_{i=1}^{p^N} U^{(L,\beta)}_{\varkappa(\mathbf{h}),i} \underline{\langle \varkappa^{-1}(i)|} \quad \text{and} \quad |\beta q^2, \mathbf{h}\rangle = U^{(R,\beta q^2)} \underline{|\mathbf{h}\rangle} = \sum_{i=1}^{p^N} U^{(R,\beta q^2)}_{i,\varkappa(\mathbf{h})} \underline{|\varkappa^{-1}(i)\rangle}, \tag{301}$$

where $\varkappa$ is an isomorphism between the sets $\{0, ..., p-1\}^N$ and $\{1, ..., p^N\}$ defined by

$$\varkappa : \mathbf{h} \in \{0, ..., p-1\}^N \to \varkappa(\mathbf{h}) \equiv 1 + \sum_{a=1}^N p^{(a-1)} h_a \in \{1, ..., p^N\}. \tag{302}$$

It follows from the pseudo-diagonalizability of $\mathscr{B}_-(\lambda|\beta)$ that the $p^N \times p^N$ square matrices $U^{(L,\beta)}$ and $U^{(R,\beta q^2)}$ are invertible matrices for which it holds

$$U^{(L,\beta)} \mathscr{B}_-(\lambda|\beta) = \Delta_{\mathscr{B}_-}(\lambda|\beta) U^{(L,\beta)}, \quad \mathscr{B}_-(\lambda|\beta) U^{(R,\beta q^2)} = U^{(R,\beta q^2)} \Delta_{\mathscr{B}_-}(\lambda|\beta), \tag{303}$$

where $\Delta_{\mathscr{B}_-}(\lambda|\beta)$ is the $p^N \times p^N$ diagonal matrix defined by

$$\left(\Delta_{\mathscr{B}_-}(\lambda|\beta)\right)_{i,j} \equiv \delta_{i,j} \mathrm{B}_{\varkappa^{-1}(i)}(\lambda|\beta) \quad \forall i,j \in \{1, ..., p^N\}. \tag{304}$$

We can prove that it holds, with the same notation as in Theorem 3.1:

**Proposition B.2.** *For almost all the values of the boundary-bulk-gauge parameters it holds*

$$\langle \Omega_\beta | \Omega_{\beta q^2} \rangle \neq 0, \tag{305}$$

*so that fixed the normalization factor*

$$\mathrm{N}_\beta = \left( \prod_{1 \leq b < a \leq N} \left( X_a^{(p-1)} - X_b^{(p-1)} \right) \langle \Omega_\beta | \Omega_{\beta q^2} \rangle \right)^{1/2}, \tag{306}$$

*in the left and right pseudo-eigenstates, then the $p^N \times p^N$ matrix $M \equiv U^{(L,\beta)} U^{(R,\beta q^2)}$ is the following invertible diagonal matrix:*

$$M_{\varkappa(\mathbf{h})\varkappa(\mathbf{k})} = \langle \beta, \mathbf{h}|\beta q^2, \mathbf{k}\rangle = \prod_{1 \leq a \leq N} \delta_{h_a, k_a} \prod_{1 \leq b < a \leq N} \frac{1}{X_a^{(h_a)} - X_b^{(h_b)}}, \tag{307}$$

*from which the following spectral decomposition of the identity $\mathbb{I}$ follows*

$$\mathbb{I} \equiv \sum_{h_1, ..., h_N = 0}^{p-1} \prod_{1 \leq b < a \leq N} (X_a^{(h_a)} - X_a^{(h_a)}) |\beta q^2, h_1, ..., h_N\rangle \langle \beta, h_1, ..., h_N|. \tag{308}$$

*Proof.* The following identity holds:

$$\mathrm{B}_{\mathbf{h}}(\lambda|\beta) \langle \beta/q^2, \mathbf{h}|\beta, \mathbf{k}\rangle = \langle \beta, \mathbf{h}| \mathscr{B}_-(\lambda|\beta)|\beta, \mathbf{k}\rangle = \mathrm{B}_{\mathbf{k}}(\lambda|\beta) \langle \beta, \mathbf{h}|\beta q^2, \mathbf{k}\rangle. \tag{309}$$

From it follows the fact that the matrix $M$ is diagonal,

$$\langle \beta/q^2, \mathbf{h}|\beta, \mathbf{k}\rangle = \langle \beta, \mathbf{h}|\beta q^2, \mathbf{k}\rangle = 0 \quad \forall \mathbf{h} \neq \mathbf{k} \in \{0, ..., p-1\}^N, \tag{310}$$

as there exists at least a $n \in \{1, ..., \mathsf{N}\}$ such that $h_n \neq k_n$ and then

$$\mathbf{B_h}(\zeta_n^{(k_n)}|\beta) \neq 0, \quad \mathbf{B_k}(\zeta_n^{(k_n)}|\beta) = 0. \tag{311}$$

Moreover, independently from the choice of the nonzero normalization factor $\mathsf{N}_\beta$, the Theorem 3.1 implies that the matrices $U^{(L,\beta)}$ and $U^{(R,\beta q^2)}$ are invertible for almost all the values of the boundary-bulk-gauge parameters so that the same must be true for the diagonal matrix $M$, i.e. it must holds

$$M_{\varkappa(\mathbf{h})\varkappa(\mathbf{h})} = \langle \mathbf{h}, \beta | \beta q^2, \mathbf{h} \rangle \neq 0 \quad \forall \mathbf{h} \in \{0, ..., p-1\}^{\mathsf{N}}, \tag{312}$$

which implies (305) for $\mathbf{p-1} \equiv (p-1, ..., p-1)$ being

$$M_{\varkappa(\mathbf{p-1})\varkappa(\mathbf{p-1})} = \frac{\langle \Omega_\beta | \Omega_{\beta q^2} \rangle}{\mathsf{N}_\beta^2}, \tag{313}$$

and we can define the normalization factor according to (306). The computation now of the remaining diagonal matrix elements $M_{\varkappa(\mathbf{h})\varkappa(\mathbf{h})}$ for $\mathbf{h} \neq \mathbf{p-1}$ can be done in a standard way by computing the matrix elements

$$\theta_{a,h_a}(\beta) \equiv \langle \beta, h_1, ..., h_a, ..., h_{\mathsf{N}} | \mathscr{A}_-(\zeta_a^{(h_a+1)}|\beta q^2) | \beta q^2, h_1, ..., h_a + 1, ..., h_{\mathsf{N}} \rangle, \tag{314}$$

where $h_a \in \{0, ..., p-1\}$, $a \in \{1, ..., \mathsf{N}\}$. Using the left action of the operator $\mathscr{A}_-(\zeta_a^{(h_a+1)}|\beta q^2)$ we get

$$\theta_{a,h_a}(\beta) = \left[ \frac{1}{q\left(\zeta_a^{(h_a)}\right)^2} - \frac{\left(\zeta_a^{(h_a)2}q - 1/q\zeta_a^{(h_a)2}\right)}{\beta(\beta - 1/\beta)} \right] \prod_{\substack{b=1 \\ b \neq a}}^{\mathsf{N}} \frac{X_a^{(h_a+1)} - X_b^{(h_b)}}{X_a^{(h_a)} - X_b^{(h_b)}}$$

$$\times \frac{\mathsf{A}_-(1/\zeta_a^{(h_a)})(q-1/q)}{\left((\zeta_a^{(h_a)})^2 - 1/(\zeta_a^{(h_a)})^2\right)} \langle \beta, h_1, ..., h_a+1, ..., h_{\mathsf{N}} | \beta, h_1, ..., h_a+1, ..., h_{\mathsf{N}} \rangle \tag{315}$$

$$= \frac{\mathsf{A}_-(1/\zeta_a^{(h_a)})(1/q-q)((\zeta_a^{(h_a)})^2 q/\beta - \beta/q(\zeta_a^{(h_a)})^2)}{\left((\zeta_a^{(h_a)})^2 - 1/(\zeta_a^{(h_a)})^2\right)(\beta - 1/\beta)} \prod_{\substack{b=1 \\ b \neq a}}^{\mathsf{N}} \frac{X_a^{(h_a+1)} - X_b^{(h_b)}}{X_a^{(h_a)} - X_b^{(h_b)}}$$

$$\times \langle \beta, h_1, ..., h_a+1, ..., h_{\mathsf{N}} | \beta, h_1, ..., h_a+1, ..., h_{\mathsf{N}} \rangle, \tag{316}$$

while using the decomposition (278) and the fact that

$$\langle \beta, h_1, ..., h_a, ..., h_{\mathsf{N}} | \mathscr{D}_-(1/\zeta_a^{(h_a+1)}|\beta q^2) | \beta q^2, h_1, ..., h_a + 1, ..., h_{\mathsf{N}} \rangle = 0, \tag{317}$$

it holds

$$\theta_{a,h_a}(\beta) = \frac{\mathsf{A}_-(1/\zeta_a^{(h_a)})(1/q-q)((\zeta_a^{(h_a)})^2 q/\beta - \beta/q(\zeta_a^{(h_a)})^2)k_a^{(h_a+1)}}{\left((\zeta_a^{(h_a+1)})^2 - 1/(\zeta_a^{(h_a+1)})^2\right)(\beta - 1/\beta)}$$

$$\times \langle \beta, h_1, ..., h_a, ..., h_{\mathsf{N}} | \beta q^2, h_1, ..., h_a, ..., h_{\mathsf{N}} \rangle. \tag{318}$$

These results lead to the identity

$$\frac{\langle \beta, h_1, ..., h_a+1, ..., h_{\mathsf{N}} | \beta q^2, h_1, ..., h_a+1, ..., h_{\mathsf{N}} \rangle}{\langle \beta, h_1, ..., h_a, ..., h_{\mathsf{N}} | \beta q^2, h_1, ..., h_a, ..., h_{\mathsf{N}} \rangle} = \prod_{\substack{b=1 \\ b \neq a}}^{\mathsf{N}} \frac{X_a^{(h_a)} - X_b^{(h_b)}}{X_a^{(h_a+1)} - X_b^{(h_b)}}, \tag{319}$$

from which one can prove

$$\frac{\langle \beta, h_1, ..., h_a, ..., h_N | \beta q^2, h_1, ..., h_a, ..., h_N \rangle}{\langle \beta, p-1, ..., p-1 | \beta q^2, p-1, ..., p-1 \rangle} = \prod_{1 \le b < a \le N} \frac{X_a^{(p-1)} - X_b^{(p-1)}}{X_a^{(h_a)} - X_b^{(h_b)}}. \tag{320}$$

This proves the proposition being by our choice of normalization,

$$\langle \beta, p-1, ..., p-1 | \beta q^2, p-1, ..., p-1 \rangle = \prod_{1 \le b < a \le N} \frac{1}{X_a^{(p-1)} - X_b^{(p-1)}}. \tag{321}$$

$$\square$$

## B.4 Proof of pseudo diagonalizability and simplicity of $\mathcal{B}_-(\lambda|\beta)$

We prove the pseudo-digonalizability and pseudo-simplicity of $\mathcal{B}_-(\lambda|\beta)$ in two steps. We first consider some special representation for which such statement is proven by direct computation then we use this result to prove our statement for general representations.

### B.4.1 Pseudo diagonalizability and simplicity of $\mathcal{B}_-(\lambda|\beta)$: special representations

The following theorem holds:

**Theorem B.3.** *Let us assume that the conditions on the bulk-gauge parameters*

$$\beta = \alpha w_N^{(\epsilon_N, k_N)}, \quad z_{n+1}^{(\epsilon_{n+1}, k_{n+1})} = 1/w_n^{(\epsilon_n, k_n)} \quad \forall n \in \{1, ..., N-1\}, \tag{322}$$

*are satisfied for fixed $N$-tuples of $\epsilon_n = \pm 1$ and $k_n \in \{0, ..., p-1\}$ and that the conditions* (218) *hold together with the following ones:*

$$\left(z_1^{(\epsilon_1, k_1)}\right)^p \neq (-\zeta_- + \epsilon_0 \sqrt{\zeta_-^2 + 4\kappa_-^2})^p / (2qe^{\tau_-}\kappa_-)^p, \tag{323}$$

*and*

$$\mu_{n,\epsilon_n}^{2p} \neq \pm 1, \quad \mu_{n,\epsilon_n}^{2p} \neq \alpha_-^{2p\epsilon}, \quad \mu_{n,\epsilon_n}^{2p} \neq -\beta_-^{2p\epsilon}, \quad \mu_{n,+}^{2p} \neq \mu_{m,-}^{2p\epsilon}, \quad \mu_{n,\epsilon_n}^p \neq \mu_{m,\epsilon_n}^p, \tag{324}$$

*for any $\epsilon_0 = \pm 1$ and $n, m \in \{1, ..., N\}$, then the operator family $\mathcal{B}_-(\lambda|\beta)$ has simple pseudo-spectrum characterized by*

$$B_{-,n}(\beta) = \mu_{n,\epsilon_n} q^{1/2} \ \forall n \in \{1, ..., N\}, \ i.e. \ independent \ w.r.t. \ \beta, \tag{325}$$

$$B_-(\beta) = f(\alpha, \beta/q, z_1^{(\epsilon_1, k_1+1)}) \left(\frac{\beta^4/q^2}{(1-\beta^2)(1-(\beta/q)^2)}\right)^N (-1)^N \prod_{n=1}^N \frac{\gamma_n^2}{\mu_{n,\epsilon_n}^2}$$

$$\times \left[z_1^{(\epsilon_1, k_1+1)} e^{\tau_-}\kappa_- + \zeta_- - \kappa_-/(z_1^{(\epsilon_1, k_1+1)} e^{\tau_-})\right] / (\zeta_- - 1/\zeta_-), \tag{326}$$

*and the left pseudo-eigenbasis characterized by the formulae* (34) *by fixing*

$$\langle \Omega_\beta | = \sum_{h_1, ..., h_N = 0}^{p-1} \bigotimes_{n=1}^N q^{h_n(k_n+1)} \left[\prod_{k_n=1}^{h_n} \frac{a_n q^{k_n - 1/2} + b_n q^{1/2 - k_n}}{c_n q^{k_n - 1/2} + d_n q^{1/2 - k_n}} \left(\frac{c_n^p + d_n^p}{a_n^p + b_n^p}\right)^{1/p}\right]^{(1+\epsilon_n)/2} \langle h_n, n|. \tag{327}$$

*Similarly, let us assume that the conditions* (218) *and* (323)-(324) *are satisfied together with*

$$\beta = \alpha w_N^{(\epsilon_N, k_N)}, \quad z_{n+1}^{(\epsilon_{n+1}, k_{n+1} - 2)} = 1/w_n^{(\epsilon_n, k_n)} \quad \forall n \in \{1, ..., N-1\}, \tag{328}$$

*for fixed* N-*tuples of* $\epsilon_n = \pm 1$ *and* $k_n \in \{0, ..., p-1\}$, *then the operator family* $\mathscr{B}_-(\lambda|\beta)$ *has simple pseudo-spectrum characterized by fixing*

$$\text{B}_{-,n}(\beta) = \mu_{n,\epsilon_n} q^{-1/2} \; \forall n \in \{1, ..., \text{N}\}, \; i.e. \; independent \; w.r.t. \; \beta, \tag{329}$$

$$\text{B}_-(\beta) = f(\alpha, \beta/q, z_1^{(\epsilon_1, k_1 - 1)}) \left( \frac{\beta^4}{(1-\beta^2)(1-(\beta/q)^2)} \right)^{\text{N}} (-1)^{\text{N}} \prod_{n=1}^{\text{N}} \frac{\gamma_n^2}{\mu_{n,\epsilon_n}^2}$$

$$\times \left[ z_1^{(\epsilon_1, k_1 - 1)} e^{\tau_-} \kappa_- + \zeta_- - \kappa_- / (z_1^{(\epsilon_1, k_1 - 1)} e^{\tau_-}) \right] / (\zeta_- - 1/\zeta_-), \tag{330}$$

*and right pseudo-eigenbasis characterized by the formulae* (35) *by fixing*

$$|\Omega_\beta\rangle = \prod_{n=1}^{\text{N}} \prod_{r_n=1}^{p-1} D_-(\text{B}_{-,n}(\beta)/q^{r_n}|\beta)|\bar{\Omega}_\beta\rangle, \tag{331}$$

*with*

$$|\bar{\Omega}_\beta\rangle = \left( \frac{\beta}{q} \right)^{\text{N}} \sum_{h_1, ..., h_N = 0}^{p-1} \prod_{n=1}^{\text{N}} q^{-h_n k_n} \left[ \prod_{r_n=1}^{h_n} \frac{c_n q^{r_n - 1/2} + d_n q^{1/2 - r_n}}{a_n q^{r_n - 1/2} + b_n q^{1/2 - r_n}} \left( \frac{a_n^p + b_n^p}{c_n^p + d_n^p} \right)^{1/p} \right]^{(1+\epsilon_n)/2}$$

$$\bigotimes_{n=1}^{\text{N}} |h_n, n\rangle. \tag{332}$$

*Proof.* The conditions (322) and the choice of internal gauge parameter

$$\gamma = z_1^{(\epsilon_1, k_1)}, \tag{333}$$

imply that the states (327) and (332) are annihilated by $A(\lambda|\alpha, \beta, \gamma)$ and $A(1/\lambda|\alpha, \beta/q, \gamma q)$ respectively as the following identifications hold:

$$\langle\Omega_\beta| = \langle\Omega, \alpha, \beta, \gamma|, \quad |\bar{\Omega}_\beta\rangle = |\Omega, \alpha, \beta/q, \gamma q\rangle \left( \frac{\beta}{q} \right)^{\text{N}}, \tag{334}$$

moreover, it holds

$$\langle\Omega_\beta|B(\lambda|\alpha, \beta) = b(\lambda|\alpha, \beta)\langle\Omega, \alpha, \beta/q, \gamma q|, \tag{335}$$

$$B(\lambda|\alpha, \beta/q)|\bar{\Omega}_\beta\rangle = |\Omega, \alpha, \beta, \gamma\rangle\beta^{\text{N}} b(\lambda q|\alpha, \beta/q), \tag{336}$$

so that it holds

$$\langle\Omega_\beta|\mathscr{B}_-(\lambda|\alpha, \beta) = f(\alpha, \beta/q, \gamma q)\bar{K}_-(\lambda|\gamma)_{21}\langle\Omega_\beta|B(\lambda|\alpha, \beta)B(1/\lambda|\alpha, \beta/q), \tag{337}$$

$$\mathscr{B}_-(\lambda|\alpha, \beta)|\bar{\Omega}_\beta\rangle = B(\lambda|\alpha, \beta)B(1/\lambda|\alpha, \beta/q)|\bar{\Omega}_\beta\rangle f(\alpha, \beta/q, \gamma q)\bar{K}_-(\lambda|\gamma)_{21}, \tag{338}$$

and consequently

$$\langle\Omega_\beta|\mathscr{B}_-(\lambda|\alpha, \beta) = f(\alpha, \beta/q, \gamma q)\bar{K}_-(\lambda|\gamma)_{21} b(\lambda|\alpha, \beta)b(1/\lambda|\alpha, \beta/q)\langle\Omega_{\beta/q^2}|, \tag{339}$$

$$\mathscr{B}_-(\lambda|\alpha, \beta)|\bar{\Omega}_\beta\rangle = |\bar{\Omega}_{\beta q^2}\rangle b(\lambda q|\alpha, \beta)b(q/\lambda|\alpha, \beta/q)f(\alpha, \beta/q, \gamma q)\bar{K}_-(\lambda|\gamma)_{21}, \tag{340}$$

so that for the pseudo-eigenvalue it holds

$$\text{B}_0(\lambda|\beta) = f(\alpha, \beta/q, \gamma q)\bar{K}_-(\lambda|\gamma)_{21} b(\lambda|\alpha, \beta)b(1/\lambda|\alpha, \beta/q), \tag{341}$$

$$\text{B}_1(\lambda|\beta) = f(\alpha, \beta/q, \gamma q)\bar{K}_-(\lambda|\gamma)_{21} b(\lambda q|\alpha, \beta)b(q/(\lambda)|\alpha, \beta/q), \tag{342}$$

respectively on the left and the right. This fixes the values of the $\text{B}_-(\beta)$ and $\text{B}_{-,a}(\beta)$ to those stated in this theorem. Note that the condition (323) implies that

$$\text{B}_-(\beta/q^{2a}) \neq 0 \quad \forall a \in \{0, ..., p-1\}. \tag{343}$$

Let us now prove that the states (34) and (35) are all nonzero states. The reasoning is done explicitly only for the left case as for the right one we can proceed similarly. We know by construction that the state $\langle \Omega_\beta |$ is nonzero so let us assume by induction that the same is true for the state $\langle \beta, \mathbf{h}^{(0)} | = \langle \beta, h_1^{(0)}, ..., h_N^{(0)} |$ with $h_j^{(0)} \in \{0, ..., p-2\}$ and let us show that $\langle \beta, \mathbf{h}_j^{(0)} | = \langle \beta, h_1^{(0)}, ..., h_j^{(0)}+1, ..., h_N^{(0)} |$ is nonzero. We have that

$$\langle \beta, \mathbf{h}_j^{(0)} | \mathscr{A}_-(\zeta_j^{(h_j^{(0)}+1)} | \beta q^2) = \text{A}_-(\zeta_j^{(h_j^{(0)}+1)}) \langle \beta, \mathbf{h}^{(0)} | \neq \underline{0} \quad \forall j \in \{1, ..., N\}, \tag{344}$$

so that $\langle \beta, \mathbf{h}_j^{(0)} |$ is nonzero. Using this we can prove that all the states $\langle \beta, h_1^{(0)}+x_1, ..., h_N^{(0)}+x_N |$ with $x_j \in \{0, 1\}$ for any $j \in \{1, ..., N\}$ are nonzero, which just proves the validity of the induction. Note that the same statements hold if we substitute the given value of $\beta$ fixed in (328) with any value $\beta/q^{2a}$ for any $a \in \{1, ..., p-1\}$; i.e. we have that $\langle \Omega_{\beta/q^{2a}} |$ is nonzero and from that we prove similarly the induction.

Let us now prove that the sets of left and right states define respectively a left and a right basis of the linear space of the representation. Let us consider the linear combination to zero of the left states

$$\underline{0} = \sum_{\mathbf{k} \in Z_p^N} c_{\mathbf{k}} \langle \beta, \mathbf{k} |, \tag{345}$$

and let us act on it with the following product of operator:

$$\begin{aligned}
\mathscr{B}_{-,\mathbf{h}}(\beta) &\equiv \mathscr{B}_-(\zeta_1^{(0)} | \beta) \mathscr{B}_-(\zeta_1^{(1)} | \beta/q^2) \cdots \mathscr{B}_-(\zeta_1^{(p-1)} | \beta/q^{2(p-2)}) \\
&\times \mathscr{B}_-(\zeta_2^{(0)} | \beta/q^{2(p-1)}) \mathscr{B}_-(\zeta_2^{(1)} | \beta/q^{2p}) \cdots \mathscr{B}_-(\zeta_2^{(p-1)} | \beta/q^{4(p-1)-2}) \\
&\cdots \times \mathscr{B}_-(\zeta_N^{(0)} | \beta/q^{2(N-1)(p-1)}) \cdots \mathscr{B}_-(\zeta_2^{(p-1)} | \beta/q^{2N(p-1)-2}), 
\end{aligned} \tag{346}$$

where the generic monomial in it,

$$\mathscr{B}_-(\zeta_m^{(0)} | \beta/q^{2(m-1)(p-1)}) \mathscr{B}_-(\zeta_2^{(1)} | \beta/q^{2(m-1)(p-1)+2}) \cdots \mathscr{B}_-(\zeta_2^{(p-1)} | \beta/q^{2m(p-1)-2}), \tag{347}$$

contains only the $p-1$ arguments $\zeta_m^{(k_m)}$ with $k_m \in \{0, ..., p-1\} \backslash \{h_n\}$ and $\mathbf{h} \equiv \{h_1, ..., h_N\}$ is a generic element of $Z_p^N$. Then, it easily to understand that it holds

$$\begin{aligned}
\underline{0} &= \left( \sum_{\mathbf{k} \in Z_p^N} c_{\mathbf{k}} \langle \beta, \mathbf{k} | \right) \mathscr{B}_{-,\mathbf{h}}(\beta) = c_{\mathbf{h}} \langle \beta, \mathbf{h} | \mathscr{B}_{-,\mathbf{h}}(\beta) \\
&= c_{\mathbf{h}} \prod_{n=1}^{N} \prod_{k_n \in \{0, ..., p-1\} \backslash \{h_n\}} \text{B}_{\mathbf{h}}(\zeta_n^{(k_n)} | \beta/q^{2(n-1)(p-1)+k_n'}) \langle \beta/q^{2N(p-1)}, \mathbf{h} |, 
\end{aligned} \tag{348}$$

where $k_n' = k_n$ if $k_n' < h_n$ and $k_n' = k_n - 1$ if $h_n < k_n$. Now the simplicity of the pseudo-spectrum of $\mathscr{B}_-(\lambda | \beta)$ implies that

$$\prod_{n=1}^{N} \prod_{k_n \in \{0, ..., p-1\} \backslash \{h_n\}} \text{B}_{\mathbf{h}}(\zeta_n^{(k_n)} | \beta/q^{2(n-1)(p-1)+k_n'}) \neq 0, \tag{349}$$

from which we derive

$$c_{\mathbf{h}} = 0, \tag{350}$$

having already proven

$$\langle \beta / q^{2N(p-1)}, \mathbf{h}| \neq \underline{0}. \tag{351}$$

The generality of the chosen $\mathbf{h} \in Z_p^N$ implies that the linear combination to zero (345) is satisfied if and only if (350) holds for any $\mathbf{h} \in Z_p^N$, that is the left pseudo-eigenstates $\langle \beta, \mathbf{k}|$ are a left basis. $\qquad\square$

Note that in the bulk of the paper we have chosen to present the construction of the SoV-basis starting from a state $|\Omega_\beta\rangle$ associated to the pseudo-eigenvalue $\mathrm{B}_0(\lambda|\beta)$ just to simplify the simultaneous presentation of the left and right basis; in fact, we can construct the right basis also starting from the state $|\bar{\Omega}_\beta\rangle$ associated to $\mathrm{B}_1(\lambda|\beta)$, which is the state constructed directly here for the considered special representations.

### B.4.2  Pseudo diagonalizability and simplicity of $\mathscr{B}_-(\lambda|\beta)$: general representations

In this section we prove the Theorem 3.1 stating the pseudo diagonalizability and simplicity of the operator family $\mathscr{B}_-(\lambda|\beta)$ for almost all the values of the boundary-bulk-gauge parameters. Let us first prove the following lemma:

**Lemma B.1.** *There exists at least one left and one right pseudo-eigenstate $|\Omega_\beta\rangle$ and $\langle \Omega_\beta|$ of the one parameter family of pseudo-commuting operators $\mathscr{B}_-(\lambda|\beta)$ satisfying the condition (30) with pseudo-eigenvalue $\mathrm{B}_0(\lambda|\beta)$ satisfying the conditions (32) and (33).*

*Proof.* The operator family $\mathscr{B}_-(\lambda|\beta)$ admits the following representation:

$$\mathscr{B}_-(\lambda|\beta) = (\frac{\lambda^2}{q} - \frac{q}{\lambda^2}) \sum_{a=0}^{N} \Lambda^a \bar{\mathscr{B}}_{-,a}(\beta) T_\beta^{-2}, \tag{352}$$

where the following commutation relations holds

$$\bar{\mathscr{B}}_{-,a}(\beta) T_\beta = T_\beta \bar{\mathscr{B}}_{-,a}(\beta q), \quad \left[\bar{\mathscr{B}}_{-,a}(\beta), \bar{\mathscr{B}}_{-,b}(\beta)\right] = 0 \quad \forall a, b \in \{1, ..., N\}, \tag{353}$$

as a consequence of the commutation relations (272). The result of the previous section implies that for some special choice of the boundary-bulk-gauge parameters all the operators $\bar{\mathscr{B}}_{-,a,\beta}$ are invertible as $\mathscr{B}_-(\lambda|\beta)$ is pseudo-diagonalizable and it admits the following representation:

$$\mathscr{B}_-(\lambda|\beta) = \mathrm{B}_-(\beta) \left(\frac{\lambda^2}{q} - \frac{q}{\lambda^2}\right) \prod_{a=1}^{N} \left(\frac{\lambda}{\mathscr{B}_{-,a}(\beta)} - \frac{\mathscr{B}_{-,a}(\beta)}{\lambda}\right) \left(\lambda \mathscr{B}_{-,a}(\beta) - \frac{1}{\lambda \mathscr{B}_{-,a}(\beta)}\right) T_\beta^{-2}, \tag{354}$$

where the $\mathscr{B}_{-,a}(\beta)$ are commuting and invertible operators. Then the fact that this operators depend continuously on these parameters implies that this statement is true for almost any values of these parameters. This also implies that for almost all the value of the boundary-bulk-gauge parameters we can use the above representation for $\mathscr{B}_-(\lambda|\beta)$.

We can now recall that, thanks to the result of the Lemma A.1 of our previous paper, we can always find a nonzero simultaneous eigenstate of commuting operators such as the $\mathscr{B}_{-,a}(\beta)$ for any $a \in \{1, ..., N\}$. This is a pseudo-eigenstate of the operator family $\mathscr{B}_-(\lambda|\beta)$.

Now, for the same set of representations considered in the previous section we know that the pseudo-eigenvalues of $\mathscr{B}_-(\lambda|\beta)$ satisfy the conditions (32) and (33). Then, we can use once again the continuity argument to argue that the eigenvalues on the common eigenstate still satisfy (32) and (33). $\qquad\square$

We can now prove the Theorem 3.1, by using the results of the previous sections.

*Proof of Theorem 3.1.* The proof of the pseudo-diagonalizability of $\mathscr{B}_-(\lambda|\beta)$ is a direct consequence of the previous lemma. Indeed, under the conditions (32) and (33) we can prove that all the left and right states are well defined and nonzero states which are pseudo-eigenstates of $\mathscr{B}_-(\lambda|\beta)$ associated to different pseudo-eigenvalues as a consequence of the gauge transformed commutation relations. The proof of the fact that the states (34) and (35) are all nonzero is done reproducing the argument presented in the proof of Theorem B.3.

The statements about the spectral decomposition of the identity of the theorem have been already given in Proposition B.2. $\qquad\square$

# C Properties of cofactor

In this appendix we prove a lemma giving the main properties of the *cofactors* of the matrix $D_\tau(\lambda)$.

**Lemma C.1.** *The matrix $D_\tau(\lambda)$ has at least rank $p-1$ for any $\lambda \in \mathbb{C}$, up to at most a finite number of values. The following symmetries,*

$$C_{i+h,j+h}(\lambda) = C_{i,j}(\lambda q^h) \quad \forall i,j,h \in \{1,...,p\}, \tag{355}$$

$$C_{i,j}(-\lambda) = C_{i,j}(\lambda) \quad \forall i,j \in \{1,...,p\}, \tag{356}$$

$$C_{1,1}(1/\lambda) = C_{1,1}(\lambda), \ C_{1,2}(1/\lambda) = C_{1,p}(\lambda), \tag{357}$$

*hold. Moreover, the cofactors $C_{1,1}(\lambda)$, $C_{1,2}(\lambda)$ and $C_{1,p}(\lambda)$ are polynomials in $\lambda$ of maximal degree $(p-1)(2N+4)$ which admit the following decomposition:*

$$C_{1,1}(\lambda) = \widehat{C}_{1,1}(\lambda)\left(\lambda^2 - \frac{1}{\lambda^2}\right)^2 \prod_{\substack{k=1 \\ k \neq (p-1)/2}}^{p-2} \left(\lambda^2 q^{1+2k} - \frac{1}{\lambda^2 q^{1+2k}}\right), \tag{358}$$

*where $\widehat{C}_{1,1}(\lambda)$ is a polynomial in $\Lambda$ of degree $(p-1)(N+1)$, and*

$$C_{1,2}(\lambda) = \widehat{C}_{1,2}(\lambda)\left(\lambda^2 - \frac{1}{\lambda^2}\right)^2 \prod_{\substack{k=1 \\ k \neq (p-1)/2}}^{p-2} \left(\lambda^2 q^{1+2k} - \frac{1}{\lambda^2 q^{1+2k}}\right), \tag{359}$$

$$C_{1,p}(\lambda) = \widehat{C}_{1,p}(\lambda)\left(\lambda^2 - \frac{1}{\lambda^2}\right)^2 \prod_{\substack{k=1 \\ k \neq (p-1)/2}}^{p-2} \left(\lambda^2 q^{1+2k} - \frac{1}{\lambda^2 q^{1+2k}}\right), \tag{360}$$

*where $\widehat{C}_{1,2}(\lambda)$ and $\widehat{C}_{1,p}(\lambda)$ are polynomials of maximal degree $2(p-1)(N+1)$ in $\lambda$.*

*Proof.* Let us remark that independently from the explicit form of $\tau(\lambda)$ the following identities hold:

$$C_{1,p}(q^{1/2-p}/\mu_{a,+}) = \prod_{j=1}^{p-1} A(\mu_{a,+}q^j) \neq 0 \quad \forall a \in \{1,...,N\}, \tag{361}$$

so that $\overline{C}_{1,p}(\lambda)$ is a non-zero polynomial in $\lambda$ which implies the statement on the rank of $D_\tau(\lambda)$. The proof of the above symmetry properties is standard we just need to make some exchange of rows and columns to bring the matrix in the determinant defining the cofactor in the l.h.s into the matrix defining the cofactor in the r.h.s..

Let us show our statement on the form of $C_{1,1}(\lambda)$. In order to do so we have to prove that $C_{1,1}(\lambda)$ is finite in the points[2] $\lambda = \pm i^a q^h$ for any $h \in \{1, ..., p-1\}$. More precisely, in the line $p-h$ there is at least one element of the matrix $M_{1,1}(\lambda)$ associated to $C_{1,1}(\lambda)$ which is diverging in the limit $\lambda \to \pm i^a q^h$. Here, we have to distinguish three cases. For the case $h \neq (p \pm 1)/2$, we can proceed as done in the bulk of the paper. We can define the matrix $M_{1,1}^{(h)}(\lambda)$ as the matrix with all the rows coinciding with those of $M_{1,1}(\lambda)$ except the row $(p+1)/2-h$, which is obtained by summing the row $(p-1)/2-h$ and $(p+1)/2-h$ of $M_{1,1}(\lambda)$ and dividing them by $((\lambda/q^h)^2 - (q^h/\lambda)^2)$, and the row $p-h$, obtained multiplying the row $p-h$ of $M_{1,1}^{(h)}(\lambda)$ by $((\lambda/q^h)^2 - (q^h/\lambda)^2)$. Clearly it holds $\det_{p-1} M_{1,1}^{(h)}(\lambda) = (-1)^{i+j} C_{1,1}(\lambda)$ and all the rows of the matrix $M_{1,1}^{(h)}(\pm i^a q^h \lambda)$ are finite in the limits $\lambda \to 1$ and so the same is true for their determinants. In fact, it is possible to show that these lines are linear dependents in each one of the matrices $M_{1,1}^{(h)}(\pm i^a q^h)$, so that

$$\det_{p-1} M_{1,1}^{(h)}(\pm i^a q^h) = 0 \quad \forall h \in \{1, ..., p\} \setminus \{0, (p \pm 1)/2\}. \tag{362}$$

In the remaining cases, if $h = (p \pm 1)/2$ then the row $(p \pm 1)/2p - h = p \mod(p)$ is not contained in $M_{1,1}(\pm i^a q^h)$ so that we cannot remove here the divergence as we have done before. However, we can proceed differently, let us explain it in the case $h = (p+1)/2$ as in the other case we can proceed similarly. In the last row of $M_{1,1}(\pm i^a q^{(p+1)/2} \lambda)$ under the limit $\lambda \to 1$ the last element tend to $\tau(i^a q^{1/2})$, finite nonzero value, and the next to last tend to $A(\pm i^a q^{-1/2}) = 0$, all the others on this row are zero. So that $C_{1,1}(\pm i^a q^{(p-1)/2})$ is finite iff $\det_{p-2} D_{(1,p),(1,p)}(\pm i^a q^{(p+1)/2})$ is finite. This is shown using the following expansion of the determinant:

$$\det_{p-2} D_{(1,p),(1,p)}(\pm i^a q^{(p+1)/2} \lambda) = \tau(\lambda) \det_{p-3} D_{\tau,(1,(p+1)/2,p),(1,(p+1)/2,p)}(\pm i^a q^{(p+1)/2} \lambda)$$
$$+ \frac{x(\lambda) \det_{p-3} D_{\tau,(1,(p+1)/2,p),(1,(p+1)/2-1,p)}(\pm i^a q^{(p+1)/2} \lambda)}{\lambda^2 - 1/\lambda^2}$$
$$- \frac{x(1/\lambda) \det_{p-3} D_{\tau,(1,(p+1)/2,p),(1,(p+1)/2+1,p)}(\pm i^a q^{(p+1)/2} \lambda)}{\lambda^2 - 1/\lambda^2}, \tag{363}$$

and the identity

$$\det_{p-3} D_{\tau,(1,(p+1)/2,p),(1,(p+1)/2-1,p)}(\pm i^a q^{(p+1)/2}) =$$
$$\det_{p-3} D_{\tau,(1,(p+1)/2,p),(1,(p+1)/2-1,p)}(\pm i^a q^{(p+1)/2}). \tag{364}$$

Finally, let us remark that in the case $h = 0$ the lines $(p-1)/2$ and $(p+1)/2$ of $M_{1,1}(\pm i^a)$ are one the opposite of the other so that $\det_{p-1} M_{1,1}(\pm i^a) = 0$. We can so define the matrix $M_{1,1}^{(0)}(\lambda)$ as the matrix with all the rows coinciding with those of $M_{1,1}(\lambda)$ except the row $(p+1)/2$, which is obtained by summing the row $(p-1)/2$ and $(p+1)/2$ of $M_{1,1}(\lambda)$ and dividing them by $(\lambda^2 - 1/\lambda^2)$, this matrix has finite elements on the row $(p+1)/2$ also in the limit $\lambda \to \pm i^a$. Similarly to the previous cases one can show that the rows of $M_{1,1}^{(0)}(\pm i^a)$ are linear dependent so that it holds also

$$\lim_{\lambda \to \pm i^a} \frac{\det_{p-1} M_{1,1}(\pm i^a \lambda)}{\lambda^2 - 1/\lambda^2} = (-1)^a \det_{p-1} M_{1,1}^{(0)}(\pm i^a) = 0, \tag{365}$$

from which our statement on the form of $C_{1,1}(\lambda)$ follows. Similarly, we can prove our statement on $C_{1,p}(\lambda)$. $\qquad \square$

---

[2] As for $h = 0$ the matrix $M_{1,1}(\pm i^a)$ does not contain any singular elements.

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
