# Peer review of "Transfer matrix spectrum for cyclic representations of the 6-vertex reflection algebra II"

_SciPost Physics, doi:SciPost Phys. 5, 026 (2018)_

## Round 1 · Referee Report · Anonymous (Referee 1) · 2018-4-16

Strengths

1- This is a very detailed paper II, following up paper I, with fully general integrable boundary conditions. Thus it is an essential companion to paper I.

Weaknesses

1- Even though this is an important paper in a difficult subject, deserving publication, it may not get many citations, being outside the scope of most researchers in the field.

2- The model derived from the original studies of integrable chiral Potts models in the 1980s, which were introduced in part to get to parafermions. No mention of parafermions is made in this project.

Report

In my opinion, this should be published as it is. Even though I have not verified all the equations in detail, from what I have seen the paper appears sound.

Requested changes

1- An optional small addition is to address possible connections with parafermions, if the authors have something useful to say about that. It would make the paper more interesting to a wider audience.

  • validity: high
  • significance: high
  • originality: high
  • clarity: high
  • formatting: excellent
  • grammar: excellent

Author:  Jean Michel Maillet  on 2018-06-29  [id 281]

(in reply to Report 1 on 2018-04-16)
Category:
remark
answer to question

There have been indeed in the recent years several quite interesting developments, in particular by P. Fendley, R. J. Baxter and H. Au-Yang and J. H. H. Perk, on the role of parafermions in the context of Chiral Potts and $\tau_2$ models. In the present article, dealing with the complete spectrum resolution of the transfer matrices of these models in the presence of general boundaries through SoV method, we do not see yet the role of these variables. Therefore, we do not have any meaningful comment to add at this stage on the role of parafermions. However, in a forthcoming article, we will show how to derive new integrable local Hamiltonians with non trivial boundary conditions associated to these transfer matrices. They are different from the Hamiltonian considered by R. J. Baxter and it would certainly be of interest to investigate the possible role of parafermions for these new integrable Hamiltonians.

---

## Round 1 · Referee Report · Anonymous (Referee 2) · 2018-4-20

Strengths

1- The authors gives the SoV spectrum for the most general transfert matrix associated to the Bazhanov-Stroganov Lax operator in the cyclic representations.

2-This paper generalizes the result of [1], also published in SciPost, and thus deserve to be published in the same journal

3-All proofs are well detailed

Weaknesses

1- Due to the complexity of the notations and the lake of references to the definition of the symbols when they are used make the paper quite hard to read, even for a specialist.

Report

I have some remarks that could allows the authors to improve the discussion in the paper and give more informations for the reader :

  • Did the S.O.V could allows to reconstruct the result of [60] starting from an other Q Baxter ? Did the author can say a bit more about a possible (or not) comparison with this result.

  • The reconstruction of SOV states into the Bethe ansatz like representation is performed in the case with a constraint (corollary 5.1). Did the author can do the same for the case without a constraint ? In the case of the open XXZ spin-1/2 chain the connection between SOV states and Bethe Vector was done in [53]

  • Did the action of the transfert matrix on the Bethe ansatz like representation of the state have a off-shell action with wanted and unwanted terms ?

Requested changes

1- all symbols should be define such that the reader can known what they are: e. g. the parameters in (2.7) , p, p' are integer, B_{-,a}(\beta),...

2- I encourage the authors to put more references of the definitions of symbols if the are not define in the same page or section to make the paper more friendly with the reader.

3- I think that there is a typo in (5.25)

4- A proof (even if it is quite trivial) for the corollary (5.1) should be added

  • validity: high
  • significance: good
  • originality: good
  • clarity: good
  • formatting: good
  • grammar: perfect

Author:  Jean Michel Maillet  on 2018-06-29  [id 282]

(in reply to Report 2 on 2018-04-20)
Category:
remark
answer to question

We first would like to express our thanks to the referee for the attentive reading, the useful help given in finding some typos, the suggestions and the interest shown in our paper. Let us already mention that to carry out the legitimate requirements of clarifications of symbols definition pointed out by the referee, we have added all along the version 2 of our manuscript several recalling to the original definitions whenever needed with the hope that it is now a bit more reader friendly. A proof of corollary (5.1) has also been added. Let us now answer the questions raised by the referee in the report.

  1. The referees 2 and 3 both asked for a more detailed explanation of the relation between our functional equation characterization of the spectrum and the ansatz proposed in [60]. This is a point that indeed is worth clarifying and we thank the referees for their question.

Let us start mentioning the first crucial difference between our approach and the one presented in [60]. Our functional equation is proven to be equivalent to the complete spectrum characterization derived in the SoV approach. It defines also the selection rule (5.27) on its solutions producing true eigenvalues (and construction of the corresponding eigenstates) of the transfer matrix. The functional equation in [60] has been proposed using only fusion relations and truncation identities. This is in fact an ansatz closely related to the analytical Bethe ansatz first proposed long ago by N. Reshetikhin. Hence the results of [60] enable to say that all the transfer matrix eigenvalues should be solutions of the equation (5.9) of [60] but do not allow to make the reverse statement. In particular in [60] there is no access to some selection rule on these solutions to identify those that really correspond to transfer matrix eigenvalues. The main origin for this lack of information in [60] is the lack of eigenstates construction. Our SoV approach indeed allows us to identify the correct solutions and to introduce the selection rule (5.27).

The comparison of the two functional equations and the explicit verification that the proposal of [60] is correct (once our selection rule (5.27) is introduced) is unfortunately not a simple task as the functional equation given in [60] is not completely explicit. Indeed, the function F, entering in the inhomogeneous term of equation (5.9) of [60], is characterized in this article in (5.12) in terms of the function A and D defined, respectively, in (4.15) and (4.16). The formulae (4.15) and (4.16) contain the average values (3.10) and (3.11) of the bulk monodromy matrices. For general cyclic representations these average functions are not given in [60]. Indeed, they are characterized by recursion using (3.10) and (3.11) and (3.12) and (3.13) of [60] but these recursions are not resolved explicitly in [60]. A possible way to verify if the functional equation of [60] is valid is to check if it interpolates the SoV characterization of the spectrum. This requires the computation of the inhomogeneous term F of the functional equation in terms of the spectrum of the separate variables, an a priori complicated task in the [60] formulation, which we have instead accomplished starting from our SoV approach.

Moreover, this explicit computation of the inhomogeneous term F, that we have carried out, allows us to identify the Laurent polynomial ($Z^2-4$) of degree $4p$ in our equation (5.55) and to show that it can be factored out from each term of our T-Q equation. This elimination of redundant zeros in the functional equation, proven in our Theorem 5.1, also explains why our Q Laurent polynomial are of $4p$ smaller degree with respect to the one appearing in the proposal of [60].

We have modified accordingly our manuscript by introducing some of these comments that indeed should be enlightening for the reader. In particular, we have modified the sentence at the end of our introduction, where we refer to the paper [60], and we have added some more technical comments after our Theorem 5.1.

  1. About the relation of our SoV construction (and properties) of separate states and eigenstates with the algebraic Bethe ansatz (or its modified version) Bethe states and off-shell Bethe states, some clarifications are indeed needed.

The Corollary 5.1 holds in general (both unconstrained and constrained cases), i.e. both if the Q Laurent polynomial is solution of the inhomogeneous or of the homogeneous Baxter like T-Q equations. The condition (3.26) only fix the gauge parameter to a convenient value while it leaves completely general the boundary parameters.

Once the transfer matrix eigenstates are derived in the SoV representation their rewriting in a Bethe ansatz like form holds true as soon as the Q-function is a (Laurent) polynomial. This statement is model independent (i.e. it holds for any representation associated to Yang-Baxter, Reflection algebra, etc.) just the “polynomiality” of Q is required and it is just one-line proof. We refer to the papers [61,104] as this rewriting was first observed there.

In the SoV approach we do not need to compute the action of the transfer matrix on states of Bethe ansatz type, indeed the transfer matrix action is completely characterized directly in the SoV basis. However, it is clear that for off-shell Bethe ansatz vectors the action of the transfer matrix produces unwanted terms, as in the SoV representation these vectors aren't transfer matrix eigenvectors. I is also clear that these unwanted terms annihilate on-shell since in this case these vectors are transfer matrix eigenvectors, as we have proven in the SoV representation.

The logic to compute the action of the transfer matrix on Bethe type vectors belongs to the algebraic Bethe ansatz and/or modified algebraic Bethe ansatz approaches. In this case the Bethe equations are obtained just identifying the conditions to annihilate the unwanted terms. For example, this is the work done in [53] for the open XXZ spin-1/2 chain.

As a concluding remark, we can comment that if desired in the SoV representation one can explicitly compute the action of the transfer matrix on a generic off-shell Bethe ansatz vector. Indeed, one can show that such type of vectors are nothing else than those that we call separate vectors. Then, by the SoV representation of the generators of the reflection algebra, we can compute the action of the transfer matrix on them proving that this produce linear combinations of new separate states, i.e. a linear combination of off-shell Bethe ansatz vectors which will contain the wanted and unwanted terms.

In version 2 of our manuscript, we have added the proof of the rewriting in Corollary 5.1 and referred to the paper [104], where it was first argued that this type of proof is model independent. Finally, we have added also a comment on the derivation given in [56] by the different logic of the modified Bethe ansatz for the XXZ spin-1/2 chain.

---

## Round 1 · Referee Report · Anonymous (Referee 3) · 2018-4-23

Strengths

1- the most general solutions of the reflection eq. for this model are considered

2- results are given in discrete form of the SoV basis and in terms of Baxter type functional eqs

3- scalar products of 'separate' states are computed as first steps towards computation of correlation functions

Weaknesses

1- the number of parameters and functions introduced along the way makes it difficult to follow the arguments.

Report

After gauging the 6-vertex cyclic reflection algebra the authors can study the spectrum of the most general boundary transfer matrices following the lines of their previous paper [1]. Compared to the earlier work the role of the gauge parameter is considered.

The analysis is sound and the results will be valuable to experts interested in this class of systems.

It would be interesting to better understand the appearance of different inhomogeneous Baxter-type functional equations -- all of them should be equivalent to the SoV-formulation. This, however, is beyond the scope of the paper.

Requested changes

1- In their remark concerning the comparison with [60] the authors note that the Q-function there has "$4p$ more zeroes". The parameter $p$, however, is introduced only later in (2.10)

2- References should be checked, e.g. Ref. [60] should contain the issue (November) in which the paper can be found.

  • validity: top
  • significance: good
  • originality: high
  • clarity: ok
  • formatting: good
  • grammar: good

Author:  Jean Michel Maillet  on 2018-06-29  [id 283]

(in reply to Report 3 on 2018-04-23)

We would like first to express our thanks to the referee for the attentive reading, the useful help given in finding some typos, the suggestions and the interest shown in our paper. Requested changes have been implemented in the version 2 of our manuscript. In particular please note that we have provided href links for all references whenever possible.

Concerning the interesting question about the comparison with the results of [60], we refer to our answer to the similar question raised by referee 2. The corresponding comments have been included in version 2 of the manuscript at the end of our introduction, where we refer to the paper [60], and we have also added some more technical comments after our Theorem 5.1.

---

## Round 2 · Author Response

List of changes
1. To carry out the legitimate requirements of clarifications of symbols definition pointed out by the referees, we have added all along the manuscript several recalling to the original definitions of symbols whenever needed (essentially whenever the definition was given several pages ago). Also href links have been provided whenever they exist.
2. Concerning the comparison of our results with the ones given in [60], we have modified the sentence at the end of our introduction, where we refer to the paper [60], to give a more detailed comparison, and we have added some more technical comments about this just after our Theorem 5.1.
3. Concerning the comparison of the SoV approach with the algebraic Bethe ansatz one, in particular concerning the role of separate states with respect to off-shell Bethe states, we have added the proof of the rewriting in Corollary 5.1 and referred to the papers [61] and [104], where it was first argued that this type of proof is model independent. Finally, we have added also a comment on the derivation given in [56] by the different logic of the modified Bethe ansatz for the XXZ spin-1/2 chain.

---

## Round 2 · List of Changes

1. To carry out the legitimate requirements of clarifications of symbols definition pointed out by the referees, we have added all along the manuscript several recalling to the original definitions of symbols whenever needed (essentially whenever the definition was given several pages ago). Also href links have been provided whenever they exist.
2. Concerning the comparison of our results with the ones given in [60], we have modified the sentence at the end of our introduction, where we refer to the paper [60], to give a more detailed comparison, and we have added some more technical comments about this just after our Theorem 5.1.
3. Concerning the comparison of the SoV approach with the algebraic Bethe ansatz one, in particular concerning the role of separate states with respect to off-shell Bethe states, we have added the proof of the rewriting in Corollary 5.1 and referred to the papers [61] and [104], where it was first argued that this type of proof is model independent. Finally, we have added also a comment on the derivation given in [56] by the different logic of the modified Bethe ansatz for the XXZ spin-1/2 chain.

---

## Editorial Decision

published